# Evaluating LLM Understanding via Structured Tabular Decision Simulations

## Abstract

Large language models (LLMs) often achieve impressive predictive accuracy, yet correctness alone does not imply genuine *understanding*. True LLM understanding, analogous to human expertise, requires making consistent, well-founded decisions across multiple instances and diverse domains, relying on relevant and domain-grounded decision factors. We introduce **Structured Tabular Decision Simulations (STaDS)**, a suite of expert-like decision settings that evaluate LLMs as if they were professionals undertaking structured decision "exams". In this context, **understanding** is defined as the ability to identify and rely on the correct **decision factors**, features that determine outcomes within a domain. STaDS jointly assesses understanding through: (i) question and instruction comprehension, (ii) knowledge-based prediction, and (iii) reliance on relevant decision factors. By analyzing 9 frontier LLMs across 15 diverse decision settings, we find that (a) most models struggle to achieve consistently strong accuracy across diverse domains; (b) models can be *accurate yet globally unfaithful*, and there are frequent mismatches between stated rationales and factors driving predictions. Our findings highlight the need for global-level understanding evaluation protocols and advocate for novel frameworks that go beyond accuracy to enhance LLMs' understanding ability.

## 1 Introduction

Large language models (LLMs) are increasingly deployed as **surrogate professionals** due to their strong predictive performance, acting as physicians for medical triage, analysts for financial risk assessment, or policy advisors for legislative decisions Abd-Alrazaq et al. (2023); Brown et al. (2020); Dong et al. (2022); Zhao et al. (2023). In such applications, users expect models to reason and make decisions with the reliability of domain experts. Yet current evaluations overwhelmingly focus on surface metrics like accuracy or task completion. What is largely missing is an assessment of the model's *understanding ability*: its internal competence to grasp and apply the principles that govern a decision task, going *beyond making a single correct prediction.* Understanding in this context is cognitive rather than purely behavioral, which makes it challenging to measure explicitly.

While recent work has begun to ask whether LLMs "reason faithfully", especially through chain-of-thought (CoT) rationales, such analyses mostly operate at the level of a single problem instance: does the explanation accompanying an answer reflect the steps that actually produced it? (Wei et al., 2022; Lewkowycz et al., 2022; Barez et al., 2025; Yu et al., 2024) These studies expose important failures of *local* reasoning faithfulness, but they do not tell us whether a model behaves like a reliable expert across many decisions in a domain (Jacovi & Goldberg, 2020; Arcuschin et al., 2025). In this work, we extend the concept of faithfulness from local reasoning traces to *global* decision faithfulness, exploring whether the model's predictions across multiple cases consistently rely on a governing rule, driven by meaningful decision factors, rather than superficial correlations (see our motivation in Fig. 1).

**This limitation motivates a shift in perspective.** We propose evaluating the **understanding ability** of LLMs as a distinct axis of competence by simulating human-like decision settings, through the lens of interpretability and explainability.

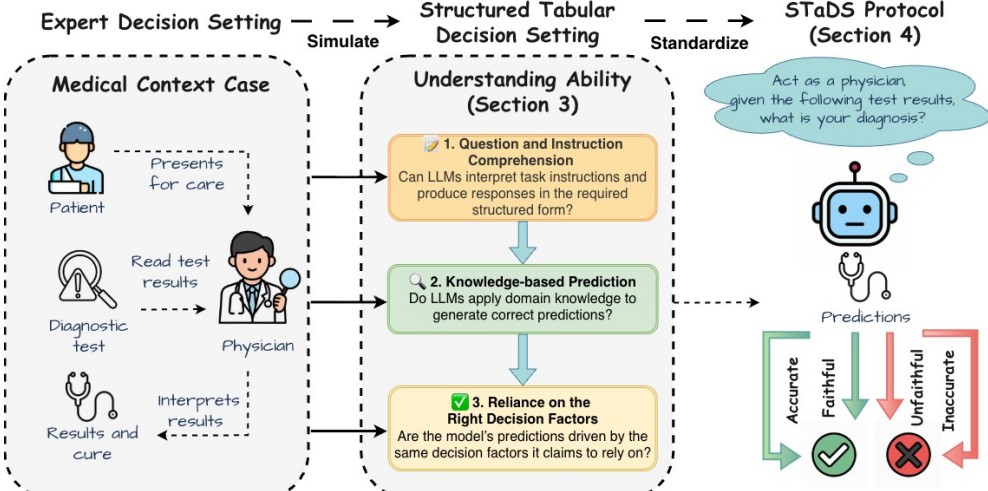

Figure 1: This diagram illustrates how the STaDS protocol simulates expert decision-making processes in structured tabular decision settings. The protocol evaluates LLMs' understanding ability through three key dimensions: (1) Question and Instruction Comprehension, assessing task interpretation and output adherence; (2) Knowledge-based Prediction, evaluating the model's application of domain knowledge for accurate predictions; and (3) Reliance on the Right Decision Factors, determining whether predictions align with the factors the model claims to rely on. The diagram depicts how these dimensions together form a principled basis for understanding and evaluating LLMs.

> By understanding ability, we refer to the capacity of an LLM to capture the underlying concepts of a task, generalize across diverse instances, and base its decisions on conceptually meaningful features (Mayer, 1989; Bereiter, 2005).

Human experts exemplify this ability: they demonstrate understanding not merely by producing a correct answer once, but by consistently applying underlying domain concepts across diverse cases. For instance, a physician is expected to base diagnoses on established medical knowledge rather than incidental correlations. Similarly, a reliable LLM should leverage its pre-learned knowledge representations to generate well-founded, conceptually grounded decisions.

To probe this dimension, we introduce **Structured Tabular Decision Simulations (STaDS)** protocol: a systematic evaluation framework that casts expert-like decision problems into tabular form (details see Sec. 3.2), enabling *controlled* assessment of both predictive performance and explanation faithfulness. Unlike reasoning, which emphasizes step-by-step justifications where models may make errors on intermediate steps (Turpin et al., 2023), tabular simulations focus on an end-to-end evaluation on whether LLMs capture the underlying decision rules or policies that govern outcomes (*interpretability*). Moreover, tabular features and labels are unambiguous and semantically well-defined: each corresponds to a clearly grounded concept (e.g., disease diagnosis, loan default, voting preference) that has direct meaning in the application domain. This stands in contrast to *free-text rationales*, which may admit multiple plausible interpretations, or *image-based tasks*, where concepts are often loosely bounded and harder to specify, and typically require extensive labeling, human validation, and additional grounding efforts to ensure *explainability* (Kim, 2024; Zarlenga et al., 2023; Li et al., 2023). In this way, the setting enables quantitative assessment of understanding ability.

**Research Questions and Contributions.** Ultimately, we target two central research questions:

> **RQ1:** *To what extent do LLMs demonstrate understanding ability by generalizing across diverse tabular decision settings, beyond surface-level correlations?*

> **RQ2:** *To what extent do LLMs demonstrate globally faithful behavior, consistently identifying the decision factors that govern outcomes within a domain?*

To answer these questions, we present a unified evaluation framework and empirical study of LLMs' understanding ability. Our primary contributions are as follows:

- **STaDS Protocol:** We propose the **Structured Tabular Decision Simulation (STaDS)** protocol as a comprehensive evaluation of LLMs' understanding ability through structured, expert-like decision settings. STaDS operationalizes understanding along three behavioral proxies: (i) *Question and instruction comprehension*: the ability to interpret task instructions and follow structured output specifications; (ii) *Knowledge-based prediction*: the capacity to apply intrinsic and in-context knowledge to produce accurate predictions; and (iii) *Reliance on the right decision factors*: the degree to which a model's predictions are driven by the same decision factors it claims to rely on. Unlike conventional evaluation frameworks, STaDS enables an end-to-end assessment that minimizes prompt bias and avoids errors from unfaithful reasoning steps. It provides a structured, reproducible, and extensible setting for LLMs, bridging concepts of interpretability (internal mechanisms) and explainability (post-hoc justifications) from eXplainable AI (XAI) to LLM evaluation (Bender et al., 2021).

- **STaDS Metrics and Benchmarks.** Building on these three axes, we define targeted metric suites for each: (1) **Comprehension Fidelity**: captured by Len-F1, UnkLbl%, and the format-related component of Penalized Accuracy to quantify instruction adherence; (2) **Predictive Competence**: measured through zero/few-shot Accuracy, Macro-F1, and overall Penalized Accuracy to assess knowledge grounding; and (3) **Decision Faithfulness**: evaluated via LAO-based feature reliance, Self-Decision Faithfulness, and SelfAtt@k to determine whether stated and actual decision factors align at the global level. To support systematic evaluation, we release a curated suite of *15* real-world tabular datasets spanning healthcare, finance, and public policy, summing up to *160k* tasks, where each task represents a real-world decision environment. This is accompanied by *standardized instruction templates* designed to minimize prompt bias and promote reproducibility.

- **Empirical Insights.** We conduct a large-scale study of *9* state-of-the-art LLMs, including advanced closed-source models (GPT, Gemini (Achiam et al., 2023; Team et al., 2023)) and leading open-source models (LLaMA, Mistral, DeepSeek, Qwen, Gemma), across all benchmarks (Dubey et al., 2024; DeepSeek, 2024; Yang et al., 2025; Team et al., 2025). Our analysis reveals a spectrum of behaviors, from models that are *neither accurate nor faithful*, to those that are *accurate but unfaithful*, and a small subset that achieves *both accuracy and faithfulness*, highlighting persistent challenges in building understanding ability in LLMs. The STaDS dataset and code are available in the Supplementary Material.

## 2 Background & Related Work

We first differentiate the notion of *understanding* in LLMs from existing terms:

**Explainability & Interpretability for LLMs.** Explainability involves providing *human-understandable justifications* for model decisions, typically through post-hoc methods linking inputs to outputs (Adadi & Berrada, 2018; Li et al., 2022). Interpretability, conversely, emphasizes *transparent internal mechanisms* such as weights and attention interactions, making the model's reasoning intrinsically comprehensible Das & Rad (2020); Ali et al. (2023). In the LLM setting, explainability has been pursued through prompt-based rationales (Liu et al., 2023), contrastive saliency for token-level influence (Min et al., 2023), and gradient-free feature attribution methods for structured tasks (Sui et al., 2024). While these approaches generate plausible explanations, they often fail to guarantee *faithfulness*, i.e., alignment with the actual causal drivers of model predictions (Jacovi & Goldberg, 2020; Agarwal et al., 2022). Evaluation practices also remain fragmented, relying on human judgments or narrow benchmarks. STaDS bridges these gaps by adapting concepts from XAI (eXplainable AI) to LLMs in a structured and model-agnostic way. *External* feature attributions are evaluated via LAO to capture human-understandable justifications, which are then compared to internally *stated reasons* for agreement.

**Reasoning & Unfaithfulness.** Chain-of-thought prompting is widely used to evaluate logical reasoning in free-form text (Wang et al., 2024), where advanced models generate step-by-step intermediate reasoning. However, such reasoning can be unreliable: models may produce errors in intermediate steps or exploit latent shortcuts that yield correct answers for the wrong reasons. Recent studies further highlight evidence of *unfaithfulness* in both thinking and non-thinking frontier models, showing that they sometimes provide correct or incorrect answers accompanied by fabricated (Implicit Post-Hoc Rationalization) or illogical justifications (Unfaithful Illogical Shortcuts) (Barez et al., 2025; Lanham et al., 2023; Arcuschin et al., 2025). We distinguish this reasoning-level unfaithfulness from the *behavioral inconsistency* targeted in STaDS, which captures differences between what a model claims and how it actually decides. Rather than examining intermediate reasoning steps, STaDS evaluates whether a model's *global feature reliance* aligns with its stated attributions, shifting the focus from local reasoning chains to end-to-end decision faithfulness.

**Step-level Reasoning & Global Attribution Reasoning.** Arcuschin et al. (2025) evaluates *Unfaithful Illogical Shortcuts* through three steps: answer correctness, step criticality, and step unfaithfulness. Their analysis centers on intermediate reasoning steps that accumulate toward a **single** decision, which makes models vulnerable to biases or fabricated justifications at the **step level**. In contrast, STaDS deliberately avoids over-reliance on step-level reasoning. Instead, it emphasizes a holistic perspective: evaluating whether a model's **overall attribution ranking**, elicited via self-attribution, faithfully reflects the features governing a **set** of decisions. We acknowledge that self-stated attribution rankings may not perfectly reflect an LLM's global attribution knowledge.

**Tabluar Relevant Tasks.** A growing body of work develops table-centric models tailored for structured data, including TAPAS (Herzig et al., 2020), TURL (Deng et al., 2022), TableLlama (Zhang et al., 2023), and TabPFN (Hollmann et al., 2025). These models support tasks such as entity linking, column annotation, and fact extraction (Deng et al., 2022; Zhang et al., 2023), and more broadly span table interpretation, augmentation, question answering, fact verification, and dialogue generation. While these methods advance training efficiency and correctness across diverse applications, they often rely on specialized architectures restricted to particular table formats or tasks (Sui et al., 2024). Specifically, they mainly target table semantics and QA. By contrast, STaDS treats tabular data as a *probing decision simulation setting*, where explicit columns enable controlled ablations, systematic perturbations, and unambiguous attribution scoring. Crucially, STaDS is **not intended as** another table-specific architecture, but rather as a complementary evaluation framework that leverages tabular data to probe whether LLMs behave like it states, beyond merely producing correct answers [1].

**In-Context Learning (ICL).** ICL enables LLMs to solve new tasks by *learning* from demonstrations provided in prompts without parameter updates, rather than focus on understanding Dong et al. (2022); Brown et al. (2020). Follow-up work probes sensitivities for understanding ICL by measuring influence factors, such as demonstration selection, order, and formatting Wang et al. (2023); Wei et al. (2023); Akyürek et al. (2022); Min et al. (2022), which can be improved by our framework. These efforts highlight that ICL competence is fragile, but they rarely interrogate whether predictions are faithful to underlying reasoning. Moreover, ICL generally evaluates the likelihood of a **single** candidate answer, whereas STaDS systematically queries a **sequence** of labels, offering a more comprehensive measure of understanding.

**Benchmark Landscape.** Existing benchmark suites typically assess competence (e.g., GLUE, MMLU) (Wang et al., 2018; Hendrycks et al., 2020), reasoning (e.g., GSM8K, DROP) (Cobbe et al., 2021; Dua et al., 2019), or explanation plausibility (e.g., ERASER, e-SNLI) (DeYoung et al., 2019; Camburu et al., 2018) in isolation. However, none addresses the interplay among understanding, interpretability, and explainability *within* an controllable decision simulation setting. Standard LLM reasoning benchmarks evaluate instance-level reasoning traces or decisions, but we aim to evaluate global attribution consistency across many cases. XAI benchmarks (e.g., OpenXAI (Agarwal et al., 2022)) focus on faithfulness of explanations for fixed models, but STaDS fills this gap by integrating real-world tabular datasets while jointly evaluating predictive competence, attributional faithfulness, and explanation agreement at a global level.

---

[1] Recent table-specific models such as TableLlama, built on LLaMA-2 (7B), can handle contexts of up to 8K tokens, yet remain unable to process the longer tabular inputs considered in this work.

# 3 STaDS Protocol: Formalizing Understanding with Tabular Decision Simulations

We introduce the STaDS protocol as a systematic evaluation framework for evaluating the understanding ability of LLMs.

## 3.1 What is Understanding?

Understanding is a broad and abstract notion. It has been described as *"a cognitive process in which concepts are used to model an abstract or physical object, establishing a relation between the knower and the object of understanding by Mayer"*. Understanding implies abilities and dispositions with respect to an object of knowledge that are sufficient to support intelligent behavior (Bereiter, 2005). We adopt this concept and refer understanding to as:

> **The model's internal competence to grasp and apply the underlying concepts and principles that govern a decision task.**

We characterize understanding as a multidimensional capacity from cognitive science and we therefore decompose understanding into three dimensions:

1. **Question and instruction comprehension**: This dimension is the ability to correctly **interpret a task**: to recognize what is being requested, identify the goal state, and determine the appropriate form of response. In cognitive terms, this is often described as constructing a situation model or problem representation "*what is going on in this task, and what is being asked of me?*" Chi et al. (2014). Successful comprehension requires mapping linguistic instructions to an internal representation of required actions, not just decoding words. In human learners, failure at this stage (misreading the question, misunderstanding constraints) is considered a failure of understanding even before any attempt at problem solving Chi et al. (1981). Interpreting and adhering to instructions is therefore evidence of understanding at the level of task framing: the LLMs (knower) demonstrates it knows what problem it is solving and what form a valid answer should take.

2. **Knowledge-based prediction**: This dimension is the capacity to apply relevant prior knowledge to produce correct inferences or decisions in a new context. This corresponds to what is classically called transfer in cognitive science, the ability to take learned knowledge or principles from one setting and apply them appropriately in another Bransford et al. (2000). Transfer is widely treated as a defining marker of genuine understanding, because it shows that performance is not tied to rote pattern matching or memorized responses but instead reflects grasp of underlying relationships and principles. Under this view, producing accurate, generalizable predictions across varied instances is behavioral evidence that the system can align internal knowledge with the current decision context productively, and use that alignment productively, known as a *hallmark of deep rather than superficial understanding* Chi et al. (2014).

3. **Reliance on the right decision factors**: This dimension is the extent to which the model's decisions are driven by the same task-relevant factors it identifies as important. Expert performance is not only accurate; it is principled. Experts are able to justify their decisions by referencing structurally relevant features of the situation, and those justifications reflect the same internal criteria they actually used to make the decision Chi et al. (1981). Novices, by contrast, often give explanations that are either post hoc, superficial, or anchored to salient but non-diagnostic surface cues. The alignment between (i) what an LLM claims matters and (ii) what actually drives its choices is therefore treated as evidence of decision faithfulness: it indicates that the LLM is guided by appropriate conceptual features of the problem space, rather than by opportunistic heuristics or randomness.

Taken together, these three dimensions mirror how human understanding is evaluated in cognitive science and educational assessment. A competent human decision-maker is expected to demonstrate *all* and we adopt the same structure when assessing LLM understanding.

### 3.1.1 What are Decision Factors?

In structured decision settings, decision factors are *the explicit, semantically grounded variables that influence outcomes within a domain.* Each factor corresponds to a domain-grounded explanatory variable, such as age, income, or tumor size, that represents part of the evidence a competent decision-maker would consider. Decision factors are thus the building blocks of rational decision-making: *they encode the domain's causal or normative structure and collectively define the reasoning space in which expertise operates.* Hence, evaluating understanding in STaDS means probing whether the model correctly identifies, prioritizes, and relies on the appropriate decision factors, mirroring how human experts justify their judgments with causally meaningful reasoning.

### 3.2 Why Tabular Decision Simulations?

Tabular decision simulations provide a principled setting for evaluating LLMs' understanding ability. Their design offers several advantages over other data formats:

1. **Instance-level structure.** Each row corresponds to a complete, self-contained decision instance: the set of feature values in that row provides all the information required to determine an outcome. This framing mirrors how experts make case-by-case judgments in real-world domains. Crucially, it prevents models from exploiting dataset-level artifacts or spurious correlations, since each prediction must be grounded in the features of a single case. Unlike tasks in vision or other multimodal reasoning, which often require additional perceptual processing or contextualization, tabular data provides a uniquely "decision-ready" input format. Each row presents a well-bounded context, with explicit and interpretable features, and an associated ground-truth label.

2. **Global-level faithfulness.** Because all attributes are explicitly named, defined, and consistently shared across rows, tabular data naturally support analysis of *global feature importance*, an established goal in XAI tasks Samek et al. (2017); Ali et al. (2023). This **distinguishes** tabular simulations from faithfulness evaluation in conventional reasoning tasks, where faithfulness is typically examined at the level of (i) individual decisions with (ii) intermediate reasoning steps. In contrast, tabular simulations enable evaluation of whether a model's decision-making aligns with coherent, domain-wide patterns of feature reliance, providing a bridge between local prediction accuracy and global reasoning consistency.

3. **Clear decision setting.** While many evaluation tasks adopt binary questions (e.g., yes/no in open-ended text or image-based object detection), such questions are typically *constructed* for the benchmark (Li et al., 2023; Arcuschin et al., 2025). In contrast, tabular data naturally encode decision outcomes as binary or multi-class classification labels, which can be directly transformed into answers without additional design. This framing avoids the biases inherent in multiple-choice formats, where prior work shows that models can flip answers in up to 36% of cases (Arcuschin et al., 2025). While our present focus is on classification, extending the protocol to regression tasks represents a natural direction for future work.

4. **Explainable end-to-end evaluation.** Because both features and labels carry explicit, domain-grounded meanings (e.g., age, income, or medical indicators), they are directly interpretable to humans without requiring additional segmentation or concept mapping. Tabular simulations capture the entire reasoning pipeline end-to-end, from structured inputs to predicted outputs, without relying on access to internal parameters or querying intermediate reasoning steps, thereby avoiding extraneous sources of bias. This design ensures that comprehension, grounding, application, and output fidelity are jointly assessed within a unified and realistic decision setting.

5. **Generality with systematic probing.** Tabular data appear across nearly every real-world domain, from healthcare and finance to science and policy, making them a natural substrate for evaluating whether LLMs can act like domain experts. Their structured format also enables systematic perturbations, such as varying the number of rows, masking attributes, or ablating features, which provide direct tests of whether predictions truly depend on the claimed decision factors. Together,

this breadth and manipulability establish tabular simulations as both widely applicable and quantitatively rigorous for assessing attributional faithfulness.

In this way, tabular decision simulations provide a distinct, structured, and reproducible environment where understanding ability can be quantitatively assessed against explainable, domain-relevant ground truth.

### 3.3 STaDS Protocol Tasks

In this section, we operationalize the process of understanding and the relationship between LLMs and structured decision tasks (the "object of understanding") through the dimensions introduced in Section 3.1. These dimensions are assessed through **observable behavioral indicators**, which are identified through violations of task specifications. These violations reveal different aspects of a model's understanding ability.

**Behavioral Indicators for Understanding**   Violations of the output specification provide diagnostic signals for different aspects of understanding ability:

1. **Question and instruction comprehension violations**: producing the wrong number of predictions, misaligned outputs, or irrelevant text indicates that the model has misinterpreted task instructions or failed to identify the required outputs. These violations assess whether the model has correctly **comprehended the task** by interpreting and following instructions properly. Misunderstandings at this level suggest that the model has not fully grasped what is being asked.

2. **Knowledge-based prediction violations**: generating labels in the correct format but with consistently low accuracy, or producing invalid labels outside $\mathcal{Y}$, suggests that the model has failed to ground prior knowledge in the domain or apply it effectively to the task. These violations reflect whether the model has correctly **applied learned knowledge** to generate correct and consistent predictions. Failures here indicate the model's inability to ground its predictions in the relevant domain knowledge.

3. **Reliance on the right decision factors violations**: providing self-claimed feature rankings inconsistent with actual feature reliance reflects that models produces unfaithful rationales despite having task-relevant knowledge. These violations measure, **decision faithfulness**, whether the model is reasoning based on the correct, relevant factors. A model with low decision faithfulness may make accurate predictions but fail to justify them using the right reasoning or decision criteria.

**STaDS Evaluation Tasks**   STaDS evaluates understanding ability through the three dimensions of understanding defined above, each assessed through distinct tasks:

1. **Comprehension Fidelity**: This task evaluates whether the model can correctly interpret task instructions and adhere to output specifications. It probes whether the model understands the task by checking whether it produces the correct number of predictions, follows the specified output format, and provides relevant responses. Misalignments or format violations here assess the model's **question and instruction comprehension ability**.

2. **Predictive Competence.** This task measures whether the model can generate accurate predictions based on its learned knowledge. It involves producing predictions for masked rows under zero-shot and few-shot settings, testing whether the model can effectively apply domain knowledge and generalize across different instances. Violations of this task (e.g., generating invalid or incorrect labels) assess the model's ability to apply relevant knowledge to make **correct predictions**.

3. **Decision faithfulness.** After generating predictions, the model is asked to provide a feature-importance ranking (self-attribution). This ranking is compared with behavioral attributions obtained through systematic perturbations, such as Leave-Any-Out (LAO) analysis. This task probes whether the model's stated rationale aligns with its actual feature reliance when making predictions, testing its **decision faithfulness**. Misalignments indicate that the model's explanation does not reflect the actual decision factors that influenced its predictions.

## 4 STaDS Metrics: Quantifying Comprehension, Competence, and Faithfulness

To rigorously evaluate understanding, STaDS introduces a suite of metrics that quantify all dimensions discussed above. We first formalize the input and output.

**Input formulation.** A tabular decision task is represented as $\mathcal{D} = \{(x_i, y_i)\}_{i=1}^{N}$, where each $x_i \in \mathbb{R}^d$ is a feature vector and $y_i \in \mathcal{Y}$ is the corresponding label. To evaluate the model, $\mathcal{D}$ is rendered into a structured prompt $\mathcal{C} = (I, T, S_k)$ at inference time, consisting of:

- $I$: a natural language instruction specifying the professional role, task, and an attribute glossary mapping features to domain concepts;

- $T$: a textual rendering of the structured table, where target labels are masked as `class=?`;

- $S_k = \{(x^{(j)}, y^{(j)})\}_{j=1}^{k}$: an optional set of $k$ in-prompt demonstrations.

**Output specification.** The LLM $f_\theta$ is required to output predictions for the masked rows.

$$\hat{\mathbf{y}} = (\hat{y}_1, \ldots, \hat{y}_{n_{\mathrm{p}}}) = f_\theta(C),$$

where $n_p$ is the number of predictions generated. Outputs are expected to follow strict formatting rules (e.g., exact number of predictions, valid label set, no additional text). These constraints ensure that performance reflects genuine task comprehension and not prompt-formatting artifacts.

**Other Notations.** We evaluate model predictions against the ground-truth labels available for each task. Let $n_p$ denote the number of predictions produced by the model and $n_g$ the number of ground-truth labels available for evaluation. To enable fair comparison[2], we define the *aligned evaluation length* as:

$$n_a = \min(n_p, n_g),$$

so that only the first $n_a$ prediction–label pairs $(\hat{y}_i, y_i)$ are considered. We define the following sets:

- **Valid label set ($\mathcal{Y}_{\mathbf{valid}}$):** the complete set of permissible label values specified by the task (e.g., $0, 1$ for binary classification, or $0, 1, 2$ for 3-class). Any $\hat{y}_i \notin \mathcal{Y}_{\mathrm{valid}}$ is treated as an invalid prediction.

- **Ground-truth label set ($\mathcal{Y}_{\mathbf{gt}}$):** the set of unique ground-truth labels among the aligned pairs $n_a$.

$$\mathcal{Y}_{\mathrm{gt}} = \{y_i : 1 \le i \le n_a\}$$

- **Predicted label set($\hat{\mathcal{Y}}$):** : the set of valid predicted labels among the aligned pairs $n_a$.

$$\hat{\mathcal{Y}} = \{\hat{y}_i : \hat{y}_i \in \mathcal{Y}_{\mathrm{valid}}, 1 \le i \le n_a\}$$

**Zero/Few Shot Settings.** STaDS explicitly considers performance in both zero-shot and few-shot settings, since the gap between them reflects the extent to which models rely on intrinsic knowledge versus in-context adaptation.

- **Zero-shot ($k = 0$):** The model receives prompt $\mathcal{C}$ with no demonstrations and must predict all rows whose labels are masked as `class=?`. This setting tests whether the model can ground its predictions directly in the instructions and table structure, reflecting its intrinsic knowledge grounding Chen et al. (2023). In other words, does the model already know enough about the decision domain to perform competently without examples?

- **Few-shot ($k > 0$):** The model receives $\mathcal{C}$ with $k$ labelled demonstrations injected into the prompt. This probes whether the model can align row-level features with labels when given exemplars, i.e., whether it can perform in-context learning in a manner analogous to how humans adapt to case-based examples Petroni et al. (2019).

---

[2]LLMs may produce incorrect number of predictions even if specified.

**General Metrics.** We adopt conventional classification metrics Accuracy (Acc), Macro-F1, and Label-Set Jaccard (Set-Jacc) for reference. These capture baseline task performance and provide a point of comparison to existing tabular classification benchmarks.

For each $c \in \mathcal{Y}_{\mathrm{gt}}$, compute precision, recall, and $F1_c$ on the aligned set.

$$\mathrm{Acc} \;=\; \frac{1}{n_a}\sum_{i=1}^{n_a} \mathbb{1}[\hat{y}_i = y_i], \quad \mathrm{Macro\text{-}F1} \;=\; \frac{1}{|\mathcal{Y}_{\mathrm{gt}}|} \sum_{c \in \mathcal{Y}_{\mathrm{gt}}} F1_c.$$

The label-set Jaccard is given by:

$$\mathrm{Set\text{-}Jacc} \;=\; \frac{|\hat{\mathcal{Y}} \cap \mathcal{Y}_{\mathrm{gt}}|}{|\hat{\mathcal{Y}} \cup \mathcal{Y}_{\mathrm{gt}}|}.$$

**Comprehension Fidelity Metrics.** We explicitly penalize *over/under-production* as well as *unknown labels*, as these indicate a failure to understand and follow task instructions.

- **Length F1 (Len-F1)**: Len-F1 measures output-length fidelity, where the model produces incorrect number of expected predictions. Let $P_L = n_a/n_p$ (set 0 if $n_p = 0$) denote precision with respect to output length, and $R_L = n_a/n_g$ (set 0 if $n_g = 0$) denote recall with respect to the number of ground-truth labels. Then,

$$\mathrm{Len\text{-}F1} \;=\; \begin{cases} \dfrac{2 P_L R_L}{P_L + R_L}, & \text{if } P_L + R_L > 0, \\ 0, & \text{otherwise.} \end{cases} \tag{1}$$

- **Unknown-Label Rate (UnkLbl%)** UnkLbl% refers to the fraction of all produced predictions outside the valid label set:

$$\mathrm{UnkLbl\%} \;=\; \frac{1}{n_p}\sum_{i=1}^{n_p} \mathbb{1}[\hat{y}_i \notin \mathcal{Y}_{\mathrm{valid}}] \quad (0 \text{ if } n_p = 0). \tag{2}$$

**Predictive Competence Metrics.** We integrates correctness, output-length fidelity, and label validity into a single measure of predictive quality as **Penalized Accuracy (PenAcc)**:

$$\mathrm{PenAcc} \;=\; \mathrm{Acc} - (\alpha \times (1 - \mathrm{Len\text{-}F1}) + \beta \times \mathrm{UnkLbl\%}), \tag{3}$$

where $\alpha, \beta > 0$ are penalty weights. Unless otherwise stated, we set the default values to $\alpha = 0.5$ and $\beta = 0.5$. Because both $(1 - \mathrm{Len\text{-}F1})$ and $\mathrm{UnkLbl\%}$ lie in $[0,1]$, PenAcc is always bounded by $[0, \mathrm{Acc}]$.

**Remark.** This metric suite allows us to rigorously assess (i) whether the output length matches the required prediction count (Len-F1); (ii) whether all predicted labels are valid (UnkLbl%), and and (iii) whether the predictions themselves are correct (Acc). PenAcc consolidates these requirements into a single indicator of predictive competence. An ideal model achieves Acc = PenAcc; any non-zero penalty ($\Delta_{\mathrm{acc}} = \mathrm{Acc} - \mathrm{PenAcc} > 0$) indicates a format violation. Consequently, PenAcc serves as a concise, end-to-end indicator of whether an LLM both *understands* the prompt specification and *solves* the prediction task.

**Decision Faithfulness Metrics.** Decision faithfulness is evaluated by comparing a model's *stated reasons* for its predictions with its *actual feature reliance*. Following principles from XAI, we employ two complementary attribution methods:

- **Self-claimed Attribution (Interpretability).** Given the same context, the model is prompted to produce a ranking $\pi_{\mathrm{self}}$ over the $m$ features, indicating which attributes it *believes* were most influential for its decision. This refers to the model's own understanding of the features that influenced its decision.

- **Leave-Any-Out (LAO) Attribution (Explainability).** For each feature $j \in \{1, \ldots, m\}$, we re-evaluate the model under identical prompting while *ablating* that feature from all rows (Koh & Liang, 2017):

$$\Delta_j \;=\; \mathrm{Perf}(\mathcal{D}) \;-\; \mathrm{Perf}\big(\mathcal{D} \setminus x_{[:,j]}\big),$$

where Perf is the same predictive metric used for predictive competence (e.g., Accuracy or F1). This yields an attribution vector $\mathbf{\Delta} = (\Delta_1, \ldots, \Delta_m)$ and an induced ranking $\pi_{\mathrm{LAO}}$, where larger $\Delta_j$ indicates a stronger reliance on feature $j$[3]. This provides a post-hoc justification for the model's decision based on the impact of removing specific features.

---

[3]Ablation can target individual features or pre-defined feature *groups* to capture higher-order interactions.

Comparing $\pi_{\text{self}}$ and $\pi_{\text{LAO}}$ helps assess the extent to which the model's stated explanations (self-claimed attribution) faithfully reflect its actual decision-making reliance (behavioral attribution). We then formalize the following metrics for global faithfulness.

- **Self-Attribution Recall (SelfAtt@ $k$)**: This metric measures how well the model's self-reported important features cover the ground-truth feature set. Specifically, given the ground-truth feature set $\mathcal{S}_m$ and $\text{Top}_k(\pi_{\text{self}})$ as the first $k$ distinct features in $\pi_{\text{self}}$, then $\text{SELFATT@}k$ is defined as:

$$\text{SELFATT@}k = \frac{|\mathcal{S}_m \cap \text{Top}_k(\pi_{\text{self}})|}{|\mathcal{S}_m|}, \quad k = |\mathcal{S}_m| \text{ by default.}$$

- **Self-Faith: Global decision faithfulness ($\rho$)**: This metric measures the agreement between the model's self-claimed attribution ranking and its actual reliance on features. Specifically, we calculate Spearman's rank correlation coefficient ($\rho$) between the behavioral ranking derived from LAO scores ($\pi_{\text{LAO}}$) and the self-claimed ranking ($\pi_{\text{self}}$). A higher agreement between these rankings indicates how faithfully the model reports its own decision rule.

$$\text{Spearman's } \rho = 1 - \frac{6}{m(m^2 - 1)} \sum_{i=1}^{m} (r_i - s_i)^2,$$

where $r_i$ and $s_i$ denote the respective ranks of feature $i$ in $\pi_{\text{self}}$ and $\pi_{\text{LAO}}$. $p$-values for $\rho$ (and $\tau$) can be obtained through permutation tests.

- **LAO Magnitude ($\sigma_{\textbf{LAO}}$)**: This metric captures the dispersion of the model's behavioral reliance across features, reflecting the concentration and interpretability of its decision rationale. It is computed as the standard deviation of the LAO performance changes across all features:

$$\sigma_{\text{LAO}} = \sqrt{\frac{1}{m-1} \sum_{j=1}^{m} (\Delta_j - \bar{\Delta})^2}, \quad \bar{\Delta} = \frac{1}{m} \sum_{j=1}^{m} \Delta_j,$$

where $\boldsymbol{\Delta} = (\Delta_1, \ldots, \Delta_m)$. A small $\sigma_{\text{LAO}}$ indicates that feature effects are evenly distributed, suggesting reliance on many weak signals. A large $\sigma_{\text{LAO}}$ indicates *sparse and concentrated reliance*, often regarded as more interpretable and human-comprehensible in XAI literature.

## 4.1 Integrated View

Taken together, these three key dimensions, **Comprehension Fidelity**, **Predictive Competence** and **Decision Faithfulness**, allow us to map the model's behavior into distinct regions of understanding. These dimensions jointly reveal how well the model performs the task and whether it does so for the right understanding.

- **Accurate & Faithful:** Reliable expert-like decision-maker, producing correct predictions grounded in faithful decision factors.
- **Accurate & Unfaithful:** Correct predictions are made, but the model's stated rationale is misleading, suggesting that while the predictions are accurate, the global rule behind them is flawed from the true decision factors.
- **Inaccurate & Faithful:** Grounded but honest decision-maker, whose reasoning aligns with true decision factors, but fails to generalize or make correct predictions.
- **Inaccurate & Unfaithful:** A failure of both competence and faithfulness, where the model neither understands the task nor applies the correct global rule.

## 5 STaDS Benchmark & Experimental Setup

We next describe the datasets, models, prompting strategy (e.g., table formatting, instruction, and question), LAO-attribution protocol, and implementation details used to instantiate the evaluation. The overview is illustrated in Fig. 2.

**Benchmark datasets.** We employ 15 tabular datasets spanning diverse domains (see professional roles in Table 1), encompassing binary and multi-class classification settings as well as both balanced and imbalanced scenarios. Sample

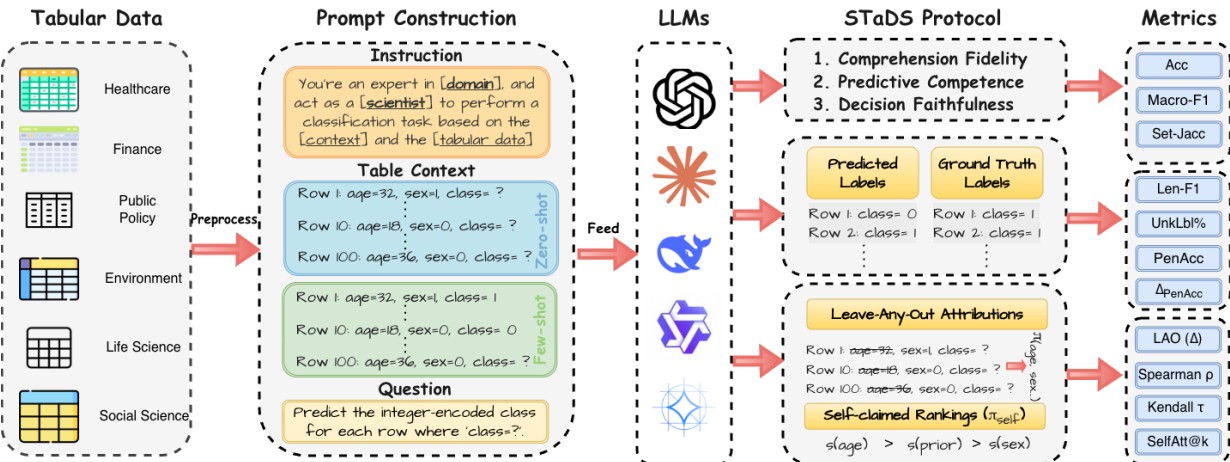

Figure 2: The diagram illustrates the process flow from tabular data preprocessing across various domains to prompt construction for LLMs. The model is tasked with predicting labels based on structured input rows and is evaluated along three key dimensions of understanding. *Question and Instruction Comprehension* is measured by Len-F1, UnkLbl%, and $\Delta_{\text{PenAcc}}$, *Predictive Competence* is quantified through Accuracy, Macro-F1, and PenAcc, and *Reliance on the Right Decision Factors* is measured by SelfAtt@k, Self-Decision Faithfulness and LAO Magnitude.

| Dataset | Sample ($N$) | Feature ($m$) | Task | Acting Role |
|---|---|---|---|---|
| Adult Income (Yeh & Lien, 2009) | 32561 | 14 | Binary | labour–economist |
| Breast Cancer (Xin et al., 2022) | 277 | 9 | Binary | clinical oncologist |
| Car Evaluation (Asuncion et al., 2007) | 1728 | 6 | Multi | automotive specialist |
| COMPAS (Jordan & Freiburger, 2015) | 6172 | 25 | Binary | criminal risk-assessment analyst |
| Congressional Voting (con, 1987) | 232 | 16 | Binary | legislative political scientist |
| Gaussian Synthetic (Agarwal et al., 2022) | 5000 | 20 | Binary | applied statistician |
| German Credit (Asuncion et al., 2007) | 1000 | 20 | Binary | bank credit risk analyst |
| Give Me Some Credit (Freshcorn, 2022) | 102209 | 10 | Binary | consumer credit risk analyst |
| Framingham Heart (World Health Organization, 2021) | 3658 | 15 | Binary | cardiovascular epidemiology analyst |
| HELOC (FICO) (Holter et al., 2018) | 9871 | 23 | Binary | home-equity lending risk analyst |
| Iris (Unwin & Kleinman, 2021) | 150 | 4 | Multi | botanical data analyst |
| MONK 1 / 2 / 3 (Thrun, 1991) | 432 | 6 | Binary | data analyst |
| Pima Indians Diabetes (Smith et al., 1988) | 768 | 8 | Binary | diabetes researcher |

Table 1: Summary of the tabular datasets, listing sample size ($N$), feature count ($m$), task type (binary/multi-class classification), and the corresponding professional roles adopted by the model in each prompt to ensure domain-relevant reasoning.

sizes range from 150 to 100,000 (tokens up to 128k), and the number of features varies from 4 to 25. For the few-shot setting, each dataset is stratified into 80% *train* (used for in-prompt demonstrations) and 20% *test* (hidden rows for evaluation).

**LLMs & Hardware.** We evaluate 9 LLMs. Open-source models: including Llama3-8B-Instruct (Llama3-8B) and Llama3-3B (Dubey et al., 2024), Mistral-7B-Instruct-v0.3 (Mistral-7B) (Jiang et al., 2023), DeepSeek-Llama-8B (DeepSeek-Llama-8B) (DeepSeek, 2024), Qwen3-8B (Yang et al., 2025), and Gemma-1B/4B-it (Gemma-1B/4B) (Team et al., 2025) are run on 8×NVIDIA RTX 3090 cluster, GH200, and 8×A100 cluster. Commercial baselines Gemini-2.5-Pro and GPT-4.1-mini are accessed via their public APIs. All models are decoded with a unified configuration (temperature 0.2, top-p 1.0, new maximum tokens 8192).

## 5.1 Prompt Construction

Inspired by the structured prompting strategy of Zhang et al. (2023), we design a **single, deterministic template** composed of four consecutive blocks for STaDS, as shown below:

| Dataset | Tokens (100 rows) | Zero-shot | | Best Z Model | Few-shot | | Best F Model |
|---|---|---|---|---|---|---|---|
| | | Acc | Macro-F1 | | Acc | Macro-F1 | |
| Adult Income | 10K | 0.700 | 0.688 | Gemini-2.5-Pro | 0.737 | 0.708 | Gemini-2.5-Pro |
| Breast Cancer | 17K | 0.729 | 0.524 | GPT-4.1-mini | 0.732 | 0.525 | GPT-4.1-mini |
| Car Evaluation | 16K | 0.419 | 0.243 | Gemini-2.5-Pro | 0.600 | 0.616 | Gemini-2.5-Pro |
| Compas | 39K | 0.816 | 0.810 | Gemini-2.5-Pro | 0.716 | 0.714 | Gemini-2.5-Pro |
| Congression Votes | 36K | 0.534 | 0.348 | Llama3-3B | 0.638 | 0.636 | Gemini-2.5-Pro |
| Synthetic | 48K | 0.550 | 0.448 | Gemini-2.5-Pro | 0.880 | 0.873 | GPT-4.1-mini |
| German Credit | 13K | 0.660 | 0.616 | GPT-4.1-mini | 0.889 | 0.862 | Gemini-2.5-Pro |
| Give Me Some Credit | 14K | 0.830 | 0.832 | DeepSeek-Llama-8B | 0.917 | 0.916 | GPT-4.1-mini |
| Heart Disease | 12K | 0.640 | 0.614 | GPT-4.1-mini | 0.700 | 0.697 | GPT-4.1-mini |
| HELOC | 25K | 0.670 | 0.670 | Gemini-2.5-Pro | 0.885 | 0.883 | Gemma-4B |
| Iris | 7K | 0.787 | 0.787 | Gemini-2.5-Pro | 1.000 | 1.000 | Gemini-2.5-Pro |
| Monk_1 | 18K | 0.620 | 0.613 | Gemini-2.5-Pro | 0.759 | 0.758 | Qwen3-8B |
| Monk_2 | 18K | 0.674 | 0.409 | DeepSeek-Llama-8B | 0.713 | 0.601 | Qwen3-8B |
| Monk_3 | 18K | 0.579 | 0.596 | Gemini-2.5-Pro | 0.644 | 0.642 | Gemini-2.5-Pro |
| Pima | 31K | 0.758 | 0.784 | Gemini-2.5-Pro | 0.820 | 0.814 | Gemini-2.5-Pro |

Table 2: Prediction results on tabular benchmarks. We report the best accuracy and macro-F1 across models for zero-shot and few-shot settings, along with the model achieving the best score and the approximate number of tokens (for 100 examples).

```
Below is an instruction that describes a task, paired with an input table that provides
further context.
Write a response that appropriately completes the request.
⟨Instruction⟩
⟨Input⟩
⟨Question⟩
⟨Response⟩
```

### 5.1.1 Instruction Template.

Each prompt begins with a concise, self-contained instruction specifying *who* the model should act as and *what* task it is to perform. Specifically, we include the following fields: ⟨DATASET⟩, ⟨ROLE⟩, ⟨TASK TYPE⟩, ⟨TARGET ENCODING⟩, and an ⟨ATTRIBUTE GLOSSARY⟩. Below are descriptions:

⟨DATASET⟩ The name of the benchmark dataset. *Example:* `Breast Cancer`.

⟨ROLE⟩ The professional identity that the model is instructed to assume, chosen to reflect domain expertise. *Example:* `clinical oncologist`.

⟨TASK TYPE⟩ The prediction setting (e.g., binary or multi-class classification) together with its domain-specific description. *Example:* `binary classification - predicting whether a breast cancer patient will experience recurrence or not`.

⟨TARGET ENCODING⟩ The mapping between integer-coded labels and their semantic meanings. This ensures the model outputs strictly integer predictions while preserving human interpretability. *Example:* `{0: no-recurrence-events, 1: recurrence-events}`.

⟨ATTRIBUTE GLOSSARY⟩ A glossary listing each input feature, its semantic description, and categorical encodings (if applicable). This grounds the tabular features in explicit domain knowledge. *Example (Breast Cancer dataset, partial):*

- `age`: Age group of the patient {0: 10–19, 1: 20–29, . . . , 8: 90–99}
- `menopause`: Menopausal status {0: lt40, 1: ge40, 2: premeno}
- `tumor-size`: Tumor size intervals in mm {0: 0–4, 1: 5–9, . . . , 11: 55–59}
- `node-caps`: Capsular invasion {0: no, 1: yes}
- `irradiat`: Radiation therapy received {0: no, 1: yes}

The instruction typically opens with:

```
Act as a professional ⟨ROLE⟩,
Your task is to perform ⟨TASK TYPE⟩, predicting whether ...  or not.
⟨TARGET ENCODING⟩ ⟨ATTRIBUTE GLOSSARY⟩
For every row where "class=?", predict its integer target, relying solely on your
pre-trained knowledge.
Return one integer per row, in the exact same order as the rows appear.
The number of predictions must equal the number of rows with "class=?".
```

### 5.1.2   Tabular Input.

The dataset is rendered as plain text, with one row per line:



`Row Num:  attribute_1 = 2, attribute_2 = 0,..., class = ?`



All categorical variables are pre-encoded as integers, and rows requiring prediction are marked with `class=?`. For *LAO experiments*, the specified feature(s) are physically removed from each row, leaving all other attributes unchanged.

### 5.1.3   Question.

The task is restated in a single sentence, explicitly specifying *the exact number $N$* of unknown rows for clarity:

> *Predict the integer-encoded class for the $N$ rows where `class=?`. Output* exactly *$N$ predictions in the same order.*

The length requirement links directly to the LEN-F1 metric.

### 5.1.4   Output Format.

The model is instructed to return a list of integer labels. For instance,



`[0, 2, 1, ..., 3]`



i.e., a Python-style list of $N$ comma-separated integers and *no additional text.* Any deviation triggers the label penalties in Sec. 4.

**Remark 5.1** *Despite explicit instructions, some LLMs often generate additional explanations or non-standard formatting. To ensure fair evaluation, we post-process all raw outputs using GPT-4.1-MINI to obtain clean, standardized predictions.*

**Self-attribution protocol.**   To elicit the self-attribution ranking ($\pi_{\text{self}}$), we provide the full table and ask the model to order all input features by their importance for predicting the target variable.

Each prompt explicitly enforces a strict output format: a single comma-separated line listing all valid feature names in descending order of importance, with the target label `class` excluded. This reduces ambiguity and prevents additional text such as numbering, bullets, or explanations. LLMs might be prompted as follows to rank all the features:

> *Rank all the features in order of their importance for predicting the target variable, from most important to least...*

We expect the model to return a list of feature names. For instance,



`[attribute_3, attribute_1, attribute_2, ..., attribute_7]`



## 6   Results & Discussion

The following section presents our findings evaluating 9 LLMs with STaDS. Throughout, we treat our metrics as *behavioral proxies* for different dimensions of understanding: comprehension fidelity, predictive competence, and global decision faithfulness.

| Model | $\sigma_{\text{LAO}}$ | Self-Faith | SelfAtt@$k$ |
|---|---|---|---|
| Gemma-1B | 0.17 | NaN | 0.00 |
| Gemini-2.5-Pro | 0.07 | 0.25 (0.38) | 1.00 |
| DeepSeek-Llama-8B | 0.24 | 0.24 (0.41) | 1.00 |
| Llama3-8B | 0.00 | −0.05 (0.89) | 0.73 |
| Qwen3-8B | 0.11 | −0.17 (0.67) | 0.44 |
| GPT-4.1-mini | 0.01 | −0.02 (0.96) | 1.00 |
| Llama3-3B | 0.11 | −0.34 (0.24) | 1.00 |
| Mistral-7B | 0.01 | −0.54 (0.08) | 0.73 |

Table 3: Decision faithfulness metrics between self-attribution rank ($\pi_{\text{self}}$) and LAO-attribution rank ($\pi_{\text{LAO}}$); Adult Income Dataset. NaN indicates $\pi_{\text{self}}$ empty.

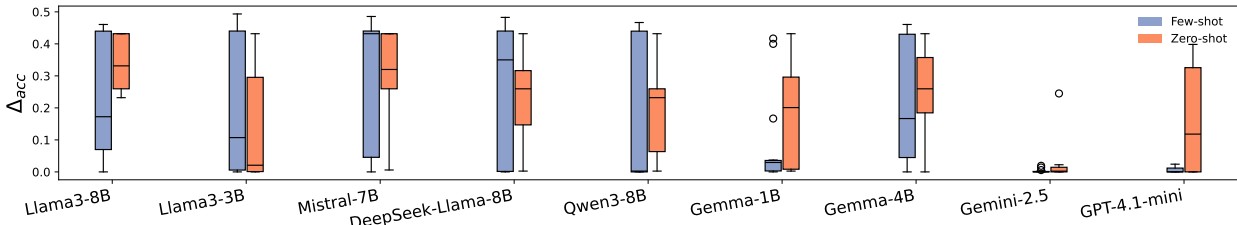

Figure 3: Box plots illustrate the distribution of $\Delta_{\text{acc}} = \text{Acc} - \text{PenAcc}$ for each model across all benchmark datasets. Blue and orange correspond to few-shot and zero-shot settings, respectively. Frontier models cluster near zero $\Delta_{\text{acc}}$, while several open-source checkpoints incur format penalties, especially in few-shot setting, indicating heightened prompt sensitivity.

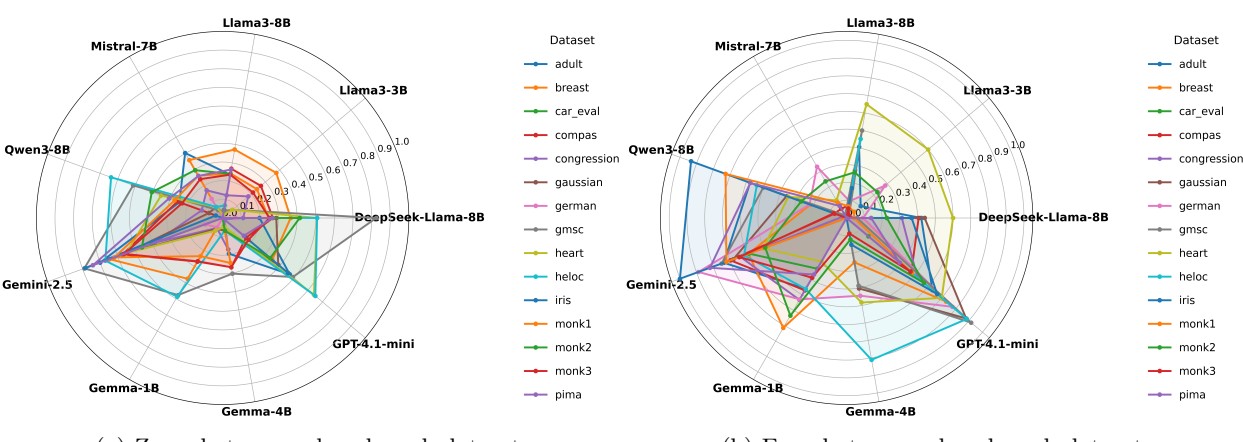

(a) Zero-shot across benchmark datasets      (b) Few-shot across benchmark datasets

Figure 4: Spider plots of Penalized Accuracy ($\alpha = 0.5, \beta = 0.5$) across models and datasets in (a) zero-shot and (b) few-shot settings. Each axis is a model; each colored trace is a dataset. Higher values indicate stronger accuracy and instruction-following. Few-shot generally inflates the polygons (with higher PenAcc) across datasets, with Gemini-2.5-Pro showing the most uniform gains.

## 6.1 Comprehension Fidelity via $\Delta_{\text{acc}}$ & SelfAtt@$k$

We first examine whether each model can handle **Comprehension Fidelity**, namely *reading the question, parsing the table, and adhering to the prescribed output format.* Fig. 3 visualizes the distribution of formatting violations across all 15 datasets in terms of the penalized accuracy difference ($\Delta_{\text{acc}}$), and Table 3 reports the Self-Attribution Recall (SelfAtt@$K$).

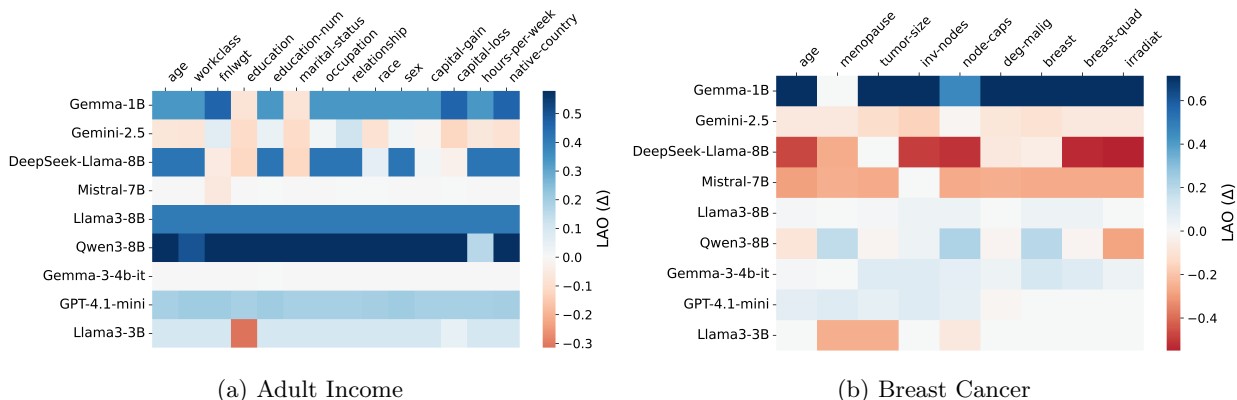

(a) Adult Income              (b) Breast Cancer

Figure 5: Heatmap of LAO performance ($\Delta_{\text{LAO}}$) for each feature (columns) and LLM (rows). Darker blue indicates a larger performance loss when the feature is removed (higher importance); red indicates a slight performance gain or negligible reliance. A few features dominate reliance for certain models (deep blue), while others spread reliance diffusely, consistent with their $\sigma_{\text{LAO}}$.

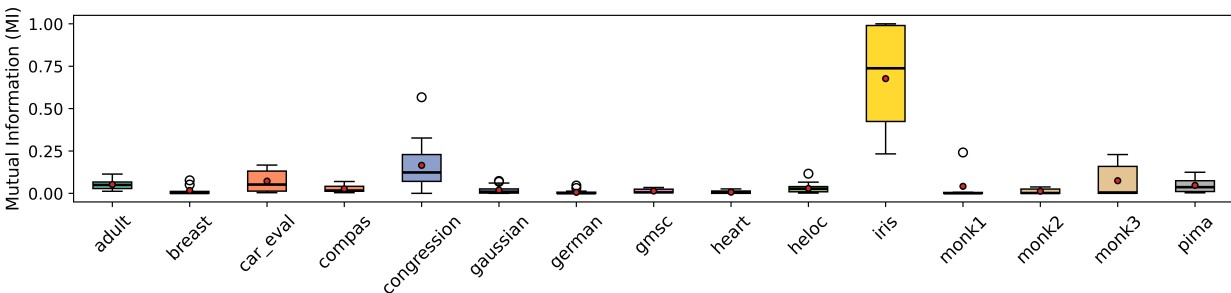

Figure 6: The boxplot summarizes the normalized distribution of Mutual NMI by dataset; Overall, NMI magnitudes are low, indicating that raw dataset co-occurrence cannot account for decision faithfulness.

**Some models struggle to interpret and follow instructions.** As shown in Fig. 3, state-of-the-art instruction-tuned models, particularly `Gemini-2.5-Pro`, exhibit near-zero $\Delta_{\text{acc}}$ across most datasets, whereas smaller open-source models such as `Llama3-8B` experience performance drops of approximately 5—40% due to basic formatting issues, such as exceeding the expected output length or generating unknown labels. Interestingly, `GPT-4.1` demonstrates that few-shot examples substantially enhance instruction-following capability, whereas smaller models (e.g., `DeepSeek-Llama-8B` and `Mistral-7B`) show little to no benefit, in some cases, even higher $\Delta_{\text{acc}}$ in few-shot settings, suggesting increased fragility to prompt variation rather than improved comprehension.

**Even short responses pose challenges for strict instruction adherence.** One might expect that long or verbose outputs are the primary cause of instruction violations, yet this assumption does not hold in our setting Tam et al. (2024). Even when responses are expected to be short, such as in the self-attribution ranking prompt, which should contain only feature names, many models still fail to comply with the prescribed output format. For instance, models such as `DeepSeek-Llama-8B` and `Llama3-3B` consistently return exactly $k$ feature names for datasets like Adult Income (Table 3), demonstrating good length fidelity. In contrast, models like `Llama3-8B` (SELFATT@$K =$ 0.73) and `Qwen3-8B` (0.44) often misidentify or omit relevant columns, indicating that instruction-following remains a persistent challenge independent of response length.

## 6.2 Predictive Competence via PenAcc

Following the instruction and table comprehension, this section mainly discusses whether models generate *accurate* labels. This reveals whether the model has failed to ground prior knowledge in the domain or apply it effectively in a tabular decision setting.

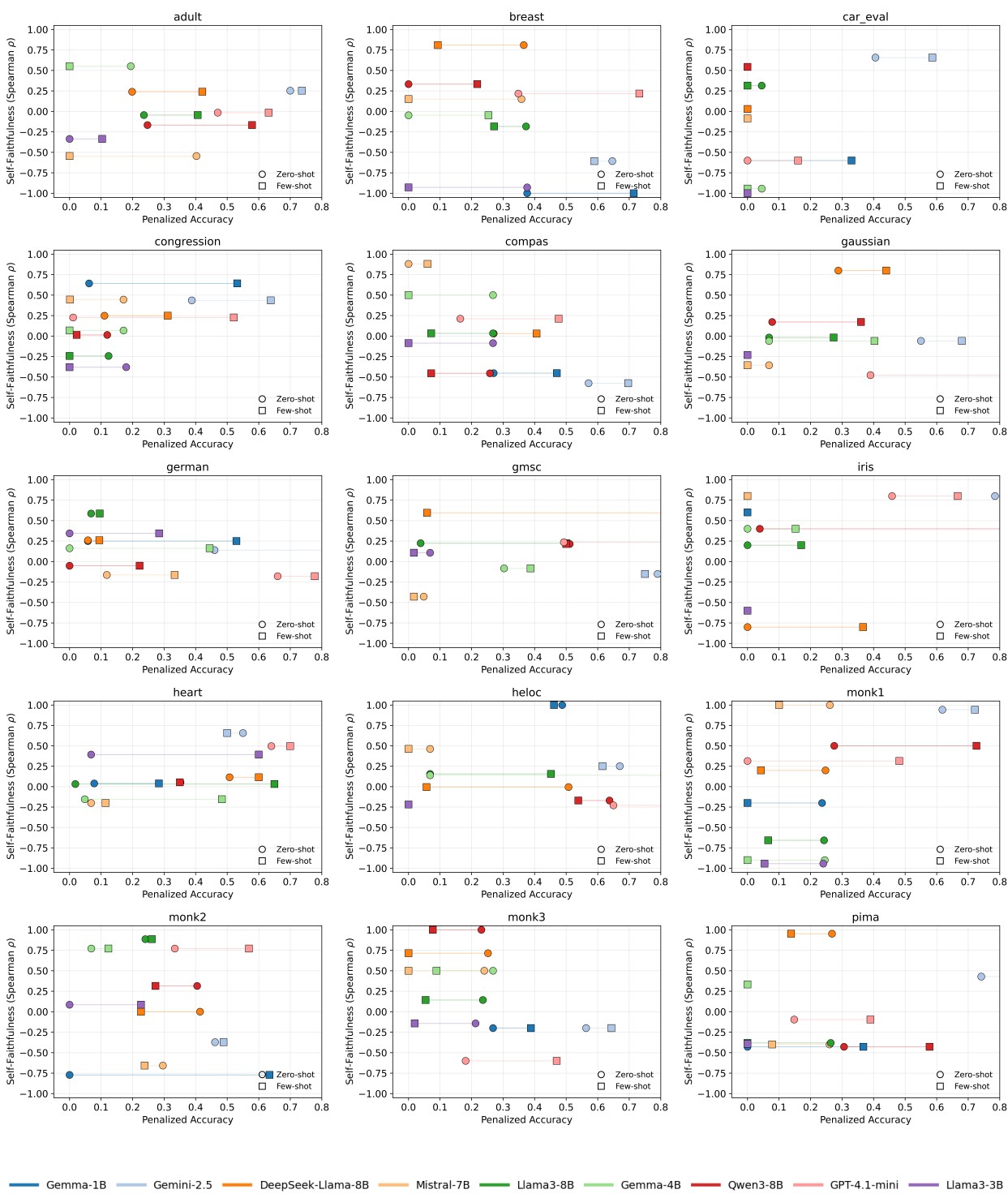

Figure 7: Penalized Accuracy vs. Self-Decision Faithfulness Across All Datasets. Each subplot shows penalized accuracy (x-axis) against self-faith (y-axis), where self-faith defined by (Spearman's $\rho$) between model-declared and behavioral (LAO) feature rankings. Zero-shot and few-shot results are shown with circles and squares, respectively. Models are colored constantly. Accuracy and faithfulness diverge for many model and dataset pairs (high-PenAcc with low $\rho$), i.e., accurate yet globally unfaithful.

**Mixed performance on short and long contexts.** While existing research commonly suggests that shorter contexts facilitate better model performance Liu et al. (2024), our results indicate complexity beyond mere input length. For instance, `Gemini-2.5-Pro` achieves high accuracy (0.81) on the COMPAS dataset (39K tokens) yet shows significantly lower performance ($\sim$0.4) on the shorter Car Evaluation dataset. This discrepancy, presented in Table 2 and Fig. 4, may arise from the multi-class nature of the Car Evaluation task, highlighting that *context length alone does not fully explain model performance variations*. However, the multi-class nature from Iris does not account for this across all models. This finding will be further explained in Sec. 6.3.

More broadly, performance analyses in (Fig. 4, panels (a) and (b)) demonstrate that models tend to achieve better results on intermediate-sized contexts, (e.g., Breast Cancer: 17K tokens; Iris: 7K tokens; Heart Disease: 12K tokens) and less consistently on very long contexts (COMPAS, PIMA). These observations suggest a complex interaction between context length, task complexity, and model capability, cautioning against interpreting token count alone as a reliable predictor of competence.

**Acting as a general professional across domains.** PenAcc jointly captures a model's ability to *follow formatting rules* and *produce the correct label*, so it naturally reflects whether an LLM can "act like a well-trained professional" across diverse structured tabular settings. Fig. 4 overlays PenAcc in radar form for every model–dataset pair, while Table 2 reports the best zero/few-shot scores.

`Gemini-2.5-Pro` dominates, topping 10/15 datasets and exhibiting almost identical polygons in the zero/few-shot plots, evidence of strong domain generalisation. Open-source checkpoints tell a more fragmented story: compact models such as `Llama3-3B` or `Gemma-1B` shine on small medical tables (e.g. Heart Disease) yet collapse on wide-schema sets like COMPAS; even larger `Llama3-8B` and `Qwen3-8B` sometimes fail simply because they mis-parse column headers or drift from the required output format rather than because they "do not know" the task.

**Zero-shot vs. few-shot: demonstrations matter.** We also compare zero-shot and few-shot performance to ask how much "being shown how to behave" helps a model act like that professional. Zero-shot in STaDS is effectively pure retrieval and schema inference: *the model must infer label semantics, column roles, and decision logic from the instruction and table alone.* Few-shot augments this with a handful of in-context demonstrations, i.e., explicit exemplars of how a domain expert would label similar rows. Across all 15 datasets, providing these demonstrations consistently improves the *best achievable* PenAcc (see Table 2), which is an upper-bound view (see detailed results in Appendix Tables 12 - 27). Averaged over all datasets, the best few-shot PenAcc exceeds the best zero-shot PenAcc by an absolute +0.15, corresponding to a 27% relative improvement. This pattern holds even for datasets where zero-shot performance is already strong, such as Iris improved from 0.79 to 1.00, and Give Me Some Credit improved from 0.83 to 0.92. In other words, demonstrations do not just rescue weaker models, and they sharpen already competent ones.

**Answer to RQ1.** Frontier LLMs exhibit *partial* professional-level competence out of the box: the strongest models can often produce valid, well-formatted predictions across diverse decision settings, but this behavior is not yet consistent across domains or model families. Demonstrations remain crucial, indicating that current models still require task-specific guidance rather than universally understanding tabular decision rules in a zero-shot setting.

### 6.3 Decision Faithfulness via Self-claimed, LAO, and Statistical Feature Importance Ranking

We now examine *global decision faithfulness*, using LAO Magnitude ($\sigma_{\text{LAO}}$), SELF-FAITH ($\rho$), and SELFATT@$k$ as behavioral proxies for the extent to which models' stated decision factors align with their actual behavior. Results are shown in Fig. 5, Fig. 7, Fig. 6, Table 4, and Appendix Figs. 13 and 14.

**Models don't always act like what they claim.** Across datasets, we observe clear discrepancies between models' self-declared feature importance and their behavioral reliance measured by LAO. For ADULT INCOME (Table 3), `DeepSeek-Llama-8B` exhibits both large LAO magnitude and a concentrated reliance on a small subset of features, while `Gemini-2.5-Pro` shows moderate sparsity but a positive SELF-FAITH ($\rho = 0.25$), indicating broad alignment between its claimed and actual decision factors. This echoes *complexity* scores used in recent XAI work (Nauta & Snoek, 2023). Conversely, models such as `Mistral-7B` and `Llama3-3B` exhibit negative SELF-FAITH ($-0.54$ and $-0.34$), meaning that the features they *say* matter diverge substantially from those that actually drive their predictions. Fig. 5 also supports these findings. For ADULT INCOME, `Gemini-2.5-Pro` and `DeepSeek-Llama-8B` predominantly depend on socio-economic attributes such as *education-num* and *occupation*, whereas `GPT-4.1-mini` and `Llama3-8B` spreads its reliance across almost every feature, producing a harder-to-interpret signature. A similar pattern emerges on BREAST CANCER in Fig. 5 (b). Results of other datasets are provided in Appendix.

Fig. 7 further illustrates this divergence by plotting PenAcc against $\rho$. We observe four regimes: (i) **Accurate & Faithful**: `Gemini-2.5-Pro` on Iris reaches PenAcc $\approx 1.0$ and $\rho \approx 0.9$, indicating concept understanding; (ii) **Inaccurate & Faithful** (e.g., `GPT-4.1-mini` on Iris), where models behave consistently with their stated priorities but still misclassify a non-trivial fraction of cases; (iii) **Accurate & Unfaithful**: `GPT-4.1-mini` on Breast Cancer predictions are strong but global alignment is weak ($\rho \approx 0.2$); and (iv) **Inaccurate & Unfaithful** (most smaller open-source models below 0.5 PenAcc and 0.4 $\rho$). These regimes demonstrate that predictive competence and global faithfulness can come apart: a model can produce many correct answers while relying on factors that differ systematically from those it highlights in its explanations.

**Understanding is Not Statistical Dependency.** To disentangle genuine reliance from dataset artifacts, we compare model attributions with feature–label dependencies measured by **Normalized Mutual Information** (NMI) [4]. We observe that most datasets exhibit low intrinsic dependence (mean NMI $< 0.08$ for 12/15 datasets), meaning accurate predictions cannot be attributed to simple, high correlations alone. High-NMI tasks (e.g., Iris or Congressional Voting) naturally support better in-context learning, and explains why models make accurate predictions in some cases (see Fig. 6). However, this trend does not generalize, datasets with similar dependence, such as Monk3, show no comparable gains.

Across domains, the features most strongly correlated with labels (e.g., `education` and `capital-gain` in Adult, or `petal-length` and `petal-width` in Iris) frequently appear in explanations and LAO rankings, but this reflects statistical covariation rather than, by itself, a guarantee of mechanistic reliance. In other words, models can detect and verbalize correlations, yet the extent to which these factors *actually* govern their decisions must be established behaviorally, via interventions such as LAO.

**Triangulated faithfulness: models explain like statisticians but act differently.** We "triangulate" understanding by correlating self-attribution ($\pi_{\text{self}}$), LAO-based behavioral reliance ($\pi_{\text{LAO}}$), and statistical dependence ($\pi_{\text{NMI}}$). Across datasets (Table 4, Appendix Table 29), we consistently find:

$$\rho(\pi_{\text{self}}, \pi_{\text{NMI}}) > \rho(\pi_{\text{LAO}}, \pi_{\text{NMI}}),$$

indicating that models' self-claimed attributions align more with *correlations present in the data* than with their own decision behavior. Only a small number of model–dataset pairs (e.g., `Gemma-4B` on Adult; `DeepSeek-Llama-8B` on GMSC and Monk1) achieve strong positive alignment between LAO and data dependencies ($> 0.5$, $p < 0.05$).

Overall, models often produce *plausible, correlation-driven rationales* while relying on different internal factors during inference. In this sense, LLMs frequently "explain like statisticians" but do not consistently "act like reliable domain experts." We also note that our LAO protocol focuses on single-feature ablations; it may underestimate reliance when models depend on higher-order feature interactions, so our faithfulness scores should be viewed as conservative, behavior-based lower bounds rather than exhaustive characterizations of understanding.

**Answer to RQ2.** Current LLMs frequently state rationales diverging from behavioral reliance ($\rho(\pi_{\text{self}}, \pi_{\text{LAO}})$) even when PenAcc is high. These self-attributions correlate more strongly with dataset regularities than with behavioral reliance, suggesting that explanations often reflect surface statistical patterns rather than stable, globally coherent decision policies. Under STaDS, current models therefore exhibit *partial* understanding: they can be competent and plausible, but not reliably faithful to the principles they articulate.

## 6.4 Conclusion

This work introduced the Structured Tabular Decision Simulations (STaDS) protocol, a principled framework for evaluating whether LLMs *understand* decision tasks rather than merely reproduce surface-level outputs. STaDS operationalizes understanding through behavioral proxies, most importantly, a model's ability not only to generate accurate predictions but also to identify and consistently rely on the correct decision factors. By integrating predictive competence, instruction adherence, and global attributional faithfulness, STaDS provides a holistic assessment of model behavior across diverse, domain-grounded tabular settings.

Our large-scale analysis of 9 LLMs over 15 tabular domains reveals a persistent gap between accuracy and understanding. While frontier models often achieve high predictive performance, they frequently violate global faithfulness, producing correct answers for the wrong reasons or inconsistently relying on relevant factors. Only a small subset of model–dataset pairs exhibit both accuracy and stable reliance on domain-relevant factors, indicating that expert-like generalization remains limited even in frontier models. These findings highlight the need to move beyond instance-level

---

[4]Complemented by Cramér's V for categorical or multi-class targets and Pearson's $r$/Spearman's $\rho$ for numerical targets.

Table 4: "Triangulated" Faithfulness Across Models and Datasets.

| Dataset | Model | $\rho(\pi_{\text{self}}, \pi_{\text{LAO}})$ | $\rho(\pi_{\text{self}}, \pi_{\text{NMI}})$ | $\rho(\pi_{\text{LAO}}, \pi_{\text{NMI}})$ |
|---|---|---|---|---|
| Adult Income | Gemma-4B | 0.552$^{\dagger}$ [0.098] | 0.394 [0.260] | 0.547* [0.043] |
| | Gemini-2.5-Pro | 0.253 [0.383] | 0.477$^{\dagger}$ [0.085] | 0.187 [0.523] |
| | DeepSeek-Llama-8B | 0.240 [0.409] | 0.301 [0.296] | -0.187 [0.523] |
| | GPT-4.1-mini | -0.015 [0.958] | 0.240 [0.409] | -0.618* [0.019] |
| Breast Cancer | DeepSeek-Llama-8B | 0.810* [0.015] | 0.586 [0.127] | 0.009 [0.982] |
| | GPT-4.1-mini | 0.217 [0.576] | 0.775* [0.014] | 0.000 [1.000] |
| Car Evaluation | Gemini-2.5-Pro | 0.657 [0.156] | 0.886* [0.019] | 0.314 [0.544] |
| | GPT-4.1-mini | -0.600 [0.208] | 0.657 [0.156] | -0.943* [0.005] |
| | Qwen3-8B | 0.543 [0.266] | 0.257 [0.623] | 0.429 [0.397] |
| COMPAS | Mistral-7B | 0.881* [0.004] | 0.095 [0.823] | 0.285 [0.425] |
| | Llama3-8B | 0.033 [0.932] | -0.417 [0.265] | 0.406 [0.244] |
| Iris | Gemini-2.5-Pro | 0.800 [0.200] | 1.000* [0.000] | 0.800 [0.200] |
| | GPT-4.1-mini | 0.800 [0.200] | 1.000* [0.000] | 0.800 [0.200] |
| Monk1 | DeepSeek-Llama-8B | 0.200 [0.704] | 0.676 [0.140] | 0.845* [0.034] |
| | Mistral-7B | 1.000* [0.000] | 0.500 [0.667] | -0.068 [0.899] |
| Pima Diabetes | DeepSeek-Llama-8B | 0.952* [0.000] | -0.286 [0.493] | -0.286 [0.493] |
| | GPT-4.1-mini | -0.095 [0.823] | 0.905* [0.002] | -0.333 [0.420] |
| | Qwen3-8B | -0.429 [0.289] | 0.857* [0.007] | -0.286 [0.493] |

*Notes.* Stars: * $p<.05$, $\dagger$ $p<.10$. Brackets show $p$-values.

prediction or reasoning traces, and instead focus on **global assessments of decision consistency and faithful factor reliance**. Future work should investigate how fine-tuning, causal or contrastive supervision, and interactive alignment with human experts can promote more robust and faithful decision policies. STaDS offers a reproducible and extensible foundation for such efforts, bridging the evaluation of accuracy, interpretability, and explainability toward a more comprehensive assessment of understanding in LLMs.

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

# Appendix

The appendix provides extended details of our experimental setup and additional results supporting the main text.

## A    Implementation Details.

We detail the full STaDS prompt template, decoding configuration, and evaluation setup to ensure reproducibility. Each prompt follows a deterministic structure composed of four blocks, as shown in Fig. 8 - 12.

All open-source models (Llama-3, Mistral-7B, Gemma, Qwen3, DeepSeek-R1) were evaluated on an $8\times$NVIDIA RTX 3090 cluster with temperature $= 0.2$ and top-p $= 1.0$. Commercial baselines (Gemini-2.5-Pro and GPT-4.1-mini) were accessed through their official APIs. Outputs were automatically cleaned using GPT-4-mini for consistent formatting. Official code will be published soon.

---

**System Prompt:**
Below is an instruction that describes a task, paired with an input table that provides further context. Write a response that appropriately completes the request.

**Instruction:**
  **Dataset**: **{Dataset name}**
  **Role**: You are a **{Role player}**
  **Task**: Your task is to perform **{binary/multi} classification** , predicting {task description}.
  **Target Encoding**: {0: {class_1}, 1: {class_2}}
  **Attribute Glossary**: **attribute_1**: attribute information,..., *** attribute _m*: attribute information.
  **Class Priors**: **0**: 0.51, **1**: 0.5.

  **Your Job**:
    • For every row where 'class=?', predict its integer-encoded target relying solely on your pre-trained knowledge.
    • Return one integer per row, in the exact same order as the rows appear.
    • The number of predictions must equal the number of rows with 'class=?'.

  **Important:**
    • Do NOT include any code, code blocks, or explanations of code in your answer.
    • Do NOT use or mention sklearn, pandas, or any code-based methods.
    • DO NOT output text as target labels, only output the integer-encoded target.

**Input:**
  Row 1: age=2, menopause=2, tumor-size=0, inv-nodes=0, breast=1, breast-quad=2, class=?
                              ⋮
  Row 5: age=4, menopause=0, tumor-size=4, inv-nodes=0, breast=1, breast-quad=2, class=?
  Row 7: age=8, menopause=0, tumor-size=4, inv-nodes=0, breast=1, breast-quad=2, class=?
                              ⋮
  Row 9: age=4, menopause=0, tumor-size=4, inv-nodes=0, breast=1, breast-quad=2, class=?

**Question:**
  Predict the integer-encoded class for each row where 'class=?'. There are exactly {num prediction} rows with 'class=?' in the input table. Do not output more or fewer predictions than {num prediction}.

---

Figure 8: **Zero-shot prompt composition.** This setup is designed to evaluate: (i) the model's ability of instruction/table comprehension, and (ii) its intrinsic, pre-trained knowledge in specific role, serving as zero-shot baselines.

## B    Extended Results

**Comprehension Fidelity & Predictive Competence**    Table 12 − 26 list detailed Acc, Macro-F1, PenAcc, Len-F1, UnkLbl%, and Set-Jacc for zero-shot and few-shot settings across all 15 datasets and 11 models. Table 27 summarizes penalized accuracy of best performing models.

**System Prompt:**
Below is an instruction that describes a task, paired with an input table that provides further context. Write a response that appropriately completes the request.

**Instruction:**
    **Dataset**: **{Dataset name}**
    **Role**: You are a **{Role player}**
    **Task**: Your task is to perform **{binary/multi} classification** , predicting {task description}.
    **Target Encoding**: {0: {class_1}, 1: {class_2}}
    **Attribute Glossary**: **attribute_1**: attribute information,..., *** attribute _m*: attribute information.
    **Class Priors**: **0**: 0.51, **1**: 0.5.

    **Your Job**:
      • For every row where 'class=?', predict its integer-encoded target relying solely on your pre-trained knowledge.
      • Return one integer per row, in the exact same order as the rows appear.
      • The number of predictions must equal the number of rows with 'class=?'.

    **Important:**
      • Do NOT include any code, code blocks, or explanations of code in your answer.
      • Do NOT use or mention sklearn, pandas, or any code-based methods.
      • DO NOT output text as target labels, only output the integer-encoded target.

**Input:**
    Row 1: age=2, menopause=2, tumor-size=0, inv-nodes=0, breast=1, breast-quad=2, class=0
                             ⋮
    Row 5: age=4, menopause=0, tumor-size=4, inv-nodes=0, breast=1, breast-quad=2, class=1
    Row 7: age=4, menopause=0, tumor-size=4, inv-nodes=0, breast=1, breast-quad=2, class=?
                             ⋮
    Row 9: age=4, menopause=0, tumor-size=4, inv-nodes=0, breast=1, breast-quad=2, class=?

**Question:**
    Predict the integer-encoded class for each row where 'class=?'. There are exactly {num prediction} rows with 'class=?' in the input table. Do not output more or fewer predictions than {num prediction}.

- - - - - - - - - - - - - - - - - - - - - - - - - - - - - - - - - -

**Response:**
    DeepSeek-R1-Llama-8B:
        <think>Alright, so I'm trying to figure out how to predict whether a <think>
        output:[1, 4, 5, 6, 7, 8, 9, 10]
    Gemma-1B:
        output:[0, 0, 0, 0, 0, 0, 0, 0,..., 0]
    Gemma-4B:
        output:[1, 1, 1, 1, 1, 1, 1, 1,..., 1]
    Qwen-8B:
        <think>Alright, so I'm trying to figure out how to predict whether a <think>
        output:
        Row 1: age=2, ... inv-nodes=0, breast=1, breast-quad=2, class=0
        Row 2: age=8, ...,inv-nodes=0, breast=1, breast-quad=2, class=0
                          ⋮
        Row 8: age=8, ...,inv-nodes=0, breast=1, breast-quad=2, class=0
    Llama3.2-8B:
        output:[0, 0, 0, 0, 0, 0, 0, 0,..., 0]
    GPT-4.1-mini:
        output:[1, 1, 0, 1, 0, 0, 1, 0]
    Gemini-2.5-Pro:
        output:[0, 1, 0, 1, 0, 0, 0, 1]

Figure 9: **Few-shot prompt composition.** The setup is designed to evaluate (i) the model's capacity for in-context generalization beyond zero-shot baselines, and (ii) its ability to jointly parse instructions and structured tabular inputs. The bottom panel presents model outputs for the same prediction prompt on the example dataset (Breast Cancer) under the few-shot configuration.

System Prompt:
Below is an instruction that describes a task, paired with an input table that provides further context. Write a response that appropriately completes the request.

Instruction:
   **Dataset**: **{Dataset name}**
   **Role**: You are a **{Role player}**
   **Task**: Your task is to perform **{binary/multi} classification** , predicting {task description}.
   **Target Encoding**: {0: {class_1}, 1: {class_2}}
   **Attribute Glossary**: **attribute_1**: attribute information,..., *** attribute _m*: attribute information.
   **Class Priors**: **0**: 0.51, **1**: 0.5.

   **Your Job**:
      • Return a **single comma-separated list** of the most important feature names (attributes) that influence the target variable according to your general understanding of the dataset.
      • Only include **original feature names** as listed in the Attribute Glossary.
      • Do **not** include reasoning, explanations, or duplicate features.
      • The number of features should be appropriate for your global understanding of the task.

   **Important:**
      • Do NOT include any code, code blocks, or explanations of code in your answer.
      • Do NOT use or mention sklearn, pandas, or any code-based methods.
      • DO NOT output text as target labels, only output the integer-encoded target.

Input:
   Row 1: age=2, menopause=2, tumor-size=0, inv-nodes=0, breast=1, breast-quad=2, class=0
                              ⋮
   Row 5: age=4, menopause=0, tumor-size=4, inv-nodes=0, breast=1, breast-quad=2, class=1
   Row 7: age=4, menopause=0, tumor-size=4, inv-nodes=0, breast=1, breast-quad=2, class=1
                              ⋮
   Row 9: age=4, menopause=0, tumor-size=4, inv-nodes=0, breast=1, breast-quad=2, class=0

Question:
   Based on your overall understanding of the dataset and the prediction task, rank all the features (**attributes**) in order of their importance for predicting the target variable (**class**), from most important to least important. Your response should be a single, comma-separated list of feature names, ordered by their relevance to the prediction task across the entire dataset.

Response:
   DeepSeek-R1-Llama-8B:
      <think>Alright, so I'm trying to figure out how to predict whether a <think>
      output: tumor-size, inv-nodes, deg-malig, age, menopause, node-caps, irradiat, breast-quad
   Gemma-1B:
      output: menopause, breast-quad, breast, breast-quad, breast, breast-quad, breast, breast-quad, breast, breast-quad, breast, breast-quad
   Gemma-4B:
      output: age, inv-nodes, tumor-size, deg-malig, node-caps, breast, breast-quad, irradiat
   Qwen-8B:
      <think>Okay, let's tackle this problem. So, the task is to determine the most important features for predicting whether a ... <think>
      output:  tumor-size,inv-nodes,deg-malig,node caps, irradiat, menopause, age, breast-quad
   Llama3.2-3B:
      ['tumor-size', 'inv-nodes', 'node-caps', 'deg-malig', 'breast-quad', 'breast', 'irradiat']
    GPT-4.1-mini:
       ['tumor-size', 'breast-quad', 'deg-malig', inv-nodes, 'breast', 'node-caps', 'irradiat']
    Gemini-2.5-Pro:
      ['tumor-size', 'inv-nodes', ' 'irradiat', 'deg-malig', 'breast-quad', 'breast', 'node-caps ']

Figure 10: **Self-attribution prompt composition.** This setup is designed to produce a ranking of $m$ features, indicating which attributes it believes were most influential for its decision. The bottom panel presents model outputs for the same attribution prompt on the example dataset (Breast Cancer).

**System Prompt:**
Below is an instruction that describes a task, paired with an input table that provides further context. Write a response that appropriately completes the request.

**Instruction:**
    **Dataset**: **{Dataset name}**
    **Role**: You are a **{Role player}**
    **Task**: Your task is to perform **{binary/multi} classification** , predicting {task description}.
    **Target Encoding**: {0: {class_1}, 1: {class_2}}
    **Attribute Glossary**: **attribute_1**: attribute information,..., *** attribute _m*: attribute information.
    **Class Priors**: **0**: 0.51, **1**: 0.5.

    **Your Job**:
        • Return a **single comma-separated list** of the most important feature names (attributes) that influence the target variable according to your general understanding of the dataset.
        • Only include **original feature names** as listed in the Attribute Glossary.
        • Do **not** include reasoning, explanations, or duplicate features.
        • The number of features should be appropriate for your global understanding of the task.

    **Important:**
        • Do NOT include any code, code blocks, or explanations of code in your answer.
        • Do NOT use or mention sklearn, pandas, or any code-based methods.
        • DO NOT output text as target labels, only output the integer-encoded target.

**Input:**
    Row 1: age=2, menopause=2, tumor-size=0, inv-nodes=0, breast=1, breast-quad=2, class=0
                        ⋮
    Row 5: age=4, menopause=0, tumor-size=4, inv-nodes=0, breast=1, breast-quad=2, class=1
    Row 7: age=4, menopause=0, tumor-size=4, inv-nodes=0, breast=1, breast-quad=2, class=1
                        ⋮
    Row 9: age=4, menopause=0, tumor-size=4, inv-nodes=0, breast=1, breast-quad=2, class=0

**Question:**
    Based on your overall understanding of the dataset and the prediction task, rank all the features (**attributes**) in order of their importance for predicting the target variable (**class**), from most important to least important. Your response should be a single, comma-separated list of feature names, ordered by their relevance to the prediction task across the entire dataset.

- - - - - - - - - - - - - - - - - - - - - - -

**Response:**
    **DeepSeek-R1-Llama-8B:**
        <think>Alright, so I'm trying to figure out how to predict whether a <think>
        output: tumor-size, inv-nodes, deg-malig, age, menopause, node-caps, irradiat, breast-quad
    **Gemma-1B:**
        output: menopause, breast-quad, breast, breast-quad, breast, breast-quad, breast, breast-quad, breast, breast-quad, breast, breast-quad
    **Gemma-4B:**
        output: age, inv-nodes, tumor-size, deg-malig, node-caps, breast, breast-quad, irradiat
    **Qwen-8B:**
        <think>Okay, let's tackle this problem. So, the task is to determine the most important features for predicting whether a ... <think>
        output:  tumor-size,inv-nodes,deg-malig,node caps, irradiat, menopause, age, breast-quad
    **Llama3.2-3B:**
        ['tumor-size', 'inv-nodes', 'node-caps', 'deg-malig', 'breast-quad', 'breast', 'irradiat']
    **GPT-4.1-mini:**
        ['tumor-size', 'breast-quad', 'deg-malig', inv-nodes', 'breast', 'node-caps', 'irradiat']
    **Gemini-2.5-Pro:**
        ['tumor-size', 'inv-nodes', ' 'irradiat', 'deg-malig', 'breast-quad', 'breast', 'node-caps ']

Figure 11: **LAO prompt composition.** This setup is designed to re-evaluate the model under identical prompting while ablating that feature from every row. The bottom panel presents model outputs for the same attribution prompt on the example dataset (Breast Cancer).

System Prompt:
Below is the output from a language model that has made predictions for a tabular dataset.
The output is expected to be a list of features, but it may not be formatted correctly or may contain additional text.
Your task is to extract the feature rankings from this output. The output may contain extra explanations or formatting, which you should ignore. You must return a complete Python list of feature names (e.g., ['sex', 'age', ...])

User: Here is the text: {output text from LLMs}.

System Prompt:
Below is the output from a language model that has made predictions for a tabular dataset.
The output is expected to be a list of predictions, but it may not be formatted correctly or may contain additional text. Your task is to extract the integer predictions from this output. The output may contain extra explanations or formatting, which you should ignore. You must return a complete Python list of integers (e.g., [0, 1, 0, ...]), where each integer corresponds to a prediction for a row in the dataset.

User: Here is the text: {output text from LLMs}.

Figure 12: **Post-processing assistant prompts** for extracting structured outputs from LLM generations. *Top panel*: Prompt for extracting a list of integer predictions from noisy or verbose LLM responses. *Bottom panel*: Prompt for extracting ranked feature names from attribution outputs, isolating the comma-separated list from surrounding text.

| Model | $\sigma_{\mathrm{LAO}}$ | SELF-FAITH | SELFATT@$k$ |
|---|---|---|---|
| Gemma-1B | 0.17 | NaN | 0.00 |
| Gemini-2.5-Pro | 0.07 | 0.25 (0.38) | 1.00 |
| DeepSeek-Llama-8B | 0.24 | 0.24 (0.41) | 1.00 |
| Llama3-8B | 0.00 | $-0.05$ (0.89) | 0.73 |
| Qwen3-8B | 0.11 | $-0.17$ (0.67) | 0.44 |
| GPT-4.1-mini | 0.01 | $-0.02$ (0.96) | 1.00 |
| Llama3-3B | 0.11 | $-0.34$ (0.24) | 1.00 |
| Mistral-7B | 0.01 | $-0.54$ (0.08) | 0.73 |

Table 5: Decision faithfulness metrics between self-attribution rank ($\pi_{\mathrm{self}}$) and LAO-attribution rank ($\pi_{\mathrm{LAO}}$); Adult Income Dataset. NaN indicates $\pi_{\mathrm{self}}$ empty.

**Decision Faithfulness** Table 5 - 11 list $\Delta_{\mathrm{LAO}}$, SELF-FAITH, and SELFATT@K for every model–dataset pair. Fig. 13, 14, and 15 visualize heatmaps of LAO performance ($\Delta_{\mathrm{LAO}}$) across all datasets.

Average of feature–target statistical dependency metrics across all benchmark datasets is provided in Table 28. "Triangulated" faithfulness across all models and datasets are provided in Tabel 29.

Table 29: "Triangulated" Faithfulness Across All Models and Datasets.

| Dataset | Model | $\rho(\pi_{\mathrm{self}}, \pi_{\mathrm{LAO}})$ | $\rho(\pi_{\mathrm{self}}, \pi_{\mathrm{NMI}})$ | $\rho(\pi_{\mathrm{LAO}}, \pi_{\mathrm{NMI}})$ |
|---|---|---|---|---|
| **Adult Income** | | | | |
| | Gemma-4B | $0.552^{\dagger}$ [0.098] | 0.394 [0.260] | $0.547^{*}$ [0.043] |
| | Gemma-1B | — [—] | — [—] | -0.165 [0.573] |
| | Gemini-2.5-Pro | 0.253 [0.383] | $0.477^{\dagger}$ [0.085] | 0.187 [0.523] |
| | DeepSeek-Llama-8B | 0.240 [0.409] | 0.301 [0.296] | -0.187 [0.523] |
| | Mistral-7B | $-0.545^{\dagger}$ [0.083] | -0.509 [0.110] | 0.235 [0.418] |
| | Llama3-8B | -0.045 [0.894] | 0.436 [0.180] | -0.182 [0.533] |
| | Qwen3-8B | -0.167 [0.668] | 0.350 [0.356] | $0.481^{\dagger}$ [0.081] |
| | GPT-4.1-mini | -0.015 [0.958] | 0.240 [0.409] | $-0.618^{*}$ [0.019] |
| | Llama3-3B | -0.336 [0.240] | 0.121 [0.681] | -0.415 [0.140] |

*Continued on next page*

| Dataset | Model | $\rho(\pi_{\text{self}}, \pi_{\text{LAO}})$ | $\rho(\pi_{\text{self}}, \pi_{\text{NMI}})$ | $\rho(\pi_{\text{LAO}}, \pi_{\text{NMI}})$ |
|---------|-------|------------------|------------------|------------------|
| **Breast Cancer** | | | | |
| | DeepSeek-Llama-8B | 0.810* [0.015] | 0.586 [0.127] | 0.009 [0.982] |
| | Llama3-3B | -0.929* [0.003] | 0.259 [0.574] | 0.183 [0.638] |
| | GPT-4.1-mini | 0.217 [0.576] | 0.775* [0.014] | 0.000 [1.000] |
| | Mistral-7B | 0.150 [0.700] | 0.366 [0.333] | 0.583$^\dagger$ [0.099] |
| | Gemini-2.5-Pro | -0.607 [0.148] | 0.595 [0.159] | -0.409 [0.274] |
| | Llama3-8B | -0.183 [0.637] | 0.522 [0.149] | -0.148 [0.704] |
| | Gemma-1B | -1.000* [0.000] | — [—] | 0.574 [0.106] |
| | Qwen3-8B | 0.333 [0.420] | 0.634$^\dagger$ [0.091] | -0.392 [0.297] |
| | Gemma-4B | -0.048 [0.911] | 0.366 [0.373] | 0.078 [0.841] |
| **Car Evaluation** | | | | |
| | Llama3-3B | -1.000 [—] | 1.000 [—] | 0.257 [0.623] |
| | Gemini-2.5-Pro | 0.657 [0.156] | 0.886* [0.019] | 0.314 [0.544] |
| | GPT-4.1-mini | -0.600 [0.208] | 0.657 [0.156] | -0.943* [0.005] |
| | DeepSeek-Llama-8B | 0.029 [0.957] | 0.143 [0.787] | -0.143 [0.787] |
| | Llama3-8B | 0.314 [0.544] | 0.657 [0.156] | 0.429 [0.397] |
| | Mistral-7B | -0.086 [0.872] | 0.086 [0.872] | -0.543 [0.266] |
| | Gemma-4B | -0.943* [0.005] | -0.086 [0.872] | -0.143 [0.787] |
| | Gemma-1B | -0.600 [0.208] | -0.314 [0.544] | -0.086 [0.872] |
| | Qwen3-8B | 0.543 [0.266] | 0.257 [0.623] | 0.429 [0.397] |
| **COMPAS** | | | | |
| | Gemini-2.5-Pro | -0.576 [0.082] | 0.030 [0.934] | -0.212 [0.556] |
| | Gemma-4B | 0.500 [0.207] | -0.167 [0.693] | -0.200 [0.580] |
| | Gemma-1B | -0.452 [0.260] | -0.119 [0.779] | -0.515 [0.128] |
| | Llama3-8B | 0.033 [0.932] | -0.417 [0.265] | 0.406 [0.244] |
| | Mistral-7B | 0.881* [0.004] | 0.095 [0.823] | 0.285 [0.425] |
| | Qwen3-8B | -0.455 [0.187] | 0.103 [0.777] | -0.248 [0.489] |
| | GPT-4.1-mini | 0.212 [0.556] | 0.394 [0.260] | 0.248 [0.489] |
| | Llama3-3B | -0.086 [0.872] | -0.543 [0.266] | 0.103 [0.777] |
| | DeepSeek-Llama-8B | 0.030 [0.934] | 0.636* [0.048] | 0.188 [0.603] |
| **Congressional Voting** | | | | |
| | DeepSeek-Llama-8B | 0.248 [0.392] | -0.095 [0.748] | -0.453 [0.078] |
| | Llama3-8B | -0.243 [0.383] | -0.013 [0.965] | 0.477$^\dagger$ [0.062] |
| | Mistral-7B | 0.445 [0.128] | -0.058 [0.851] | 0.319 [0.228] |
| | GPT-4.1-mini | 0.226 [0.399] | 0.350 [0.184] | -0.037 [0.892] |
| | Gemma-4B | 0.068 [0.810] | 0.579* [0.024] | 0.306 [0.249] |
| | Gemma-1B | 0.643 [0.119] | -0.487 [0.268] | -0.311 [0.242] |
| | Qwen3-8B | 0.014 [0.960] | 0.465 [0.081] | 0.255 [0.341] |
| | Llama3-3B | -0.379 [0.147] | 0.041 [0.880] | 0.284 [0.286] |
| | Gemini-2.5-Pro | 0.435 [0.092] | 0.748* [0.001] | 0.383 [0.144] |
| **Gaussian Synthetic** | | | | |
| | Gemma-1B | — [—] | — [—] | -0.061 [0.798] |
| | Gemma-4B | -0.060 [0.801] | 0.645* [0.002] | -0.301 [0.197] |
| | DeepSeek-Llama-8B | 0.800 [0.200] | 0.800 [0.200] | 0.271 [0.248] |
| | Llama3-3B | -0.232 [0.326] | 0.645* [0.002] | -0.057 [0.810] |
| | GPT-4.1-mini | -0.477* [0.034] | 0.192 [0.418] | -0.101 [0.671] |
| | Mistral-7B | -0.356 [0.123] | 0.575* [0.008] | -0.525* [0.017] |
| | Qwen3-8B | 0.171 [0.470] | -0.193 [0.414] | -0.031 [0.897] |
| | Gemini-2.5-Pro | -0.059 [0.806] | 0.380 [0.098] | 0.225 [0.340] |
| | Llama3-8B | -0.018 [0.940] | 0.645* [0.002] | -0.306 [0.190] |
| **German Credit** | | | | |
| | Llama3-3B | 0.344 [0.137] | 0.021 [0.929] | 0.268 [0.253] |
| | DeepSeek-Llama-8B | 0.260 [0.283] | -0.108 [0.659] | -0.430 [0.059] |
| | Llama3-8B | 0.586* [0.008] | -0.033 [0.893] | 0.092 [0.699] |
| | Mistral-7B | -0.164 [0.631] | 0.569 [0.068] | 0.021 [0.929] |
| | Qwen3-8B | -0.051 [0.836] | -0.066 [0.787] | -0.066 [0.783] |
| | GPT-4.1-mini | -0.179 [0.450] | -0.053 [0.825] | -0.095 [0.689] |
| | Gemma-1B | 0.250 [0.369] | 0.113 [0.689] | -0.148 [0.533] |
| | Gemma-4B | 0.162 [0.521] | 0.125 [0.622] | 0.097 [0.684] |
| | Gemini-2.5-Pro | 0.139 [0.701] | -0.020 [0.955] | 0.389 [0.266] |
| **Give Me Some Credit** | | | | |
| | Mistral-7B | -0.429 [0.289] | 0.452 [0.260] | -0.394 [0.260] |
| | GPT-4.1-mini | 0.236 [0.511] | 0.709* [0.022] | 0.261 [0.467] |
| | Gemma-4B | -0.083 [0.831] | 0.400 [0.286] | -0.139 [0.701] |
| | Gemini-2.5-Pro | -0.152 [0.676] | 0.758* [0.011] | 0.127 [0.726] |
| | Llama3-8B | 0.224 [0.533] | 0.224 [0.533] | -0.564 [0.090] |
| | Gemma-1B | — [—] | — [—] | 0.685* [0.029] |
| | DeepSeek-Llama-8B | 0.595 [0.120] | 0.333 [0.420] | 0.673* [0.033] |
| | Qwen3-8B | 0.214 [0.610] | 0.857* [0.007] | -0.030 [0.934] |
| | Llama3-3B | 0.107 [0.819] | 0.893* [0.007] | 0.406 [0.244] |
| **Heart Disease** | | | | |

*Continued on next page*

| Dataset | Model | $\rho(\pi_{\text{self}}, \pi_{\text{LAO}})$ | $\rho(\pi_{\text{self}}, \pi_{\text{NMI}})$ | $\rho(\pi_{\text{LAO}}, \pi_{\text{NMI}})$ |
|---|---|---|---|---|
| | Mistral-7B | -0.200 [0.475] | 0.170 [0.545] | -0.435 [0.105] |
| | Gemma-1B | 0.039 [0.889] | -0.129 [0.647] | 0.275 [0.322] |
| | Gemma-4B | -0.154 [0.633] | 0.303 [0.339] | 0.144 [0.609] |
| | Gemini-2.5-Pro | 0.657* [0.008] | 0.572* [0.026] | 0.061 [0.829] |
| | Llama3-8B | 0.032 [0.909] | 0.380 [0.162] | -0.046 [0.870] |
| | GPT-4.1-mini | 0.496 [0.060] | 0.321 [0.244] | -0.100 [0.724] |
| | Llama3-3B | 0.393 [0.147] | -0.129 [0.647] | -0.184 [0.511] |
| | DeepSeek-Llama-8B | 0.115 [0.707] | 0.328 [0.274] | -0.266 [0.339] |
| | Qwen3-8B | 0.054 [0.850] | 0.245 [0.378] | 0.172 [0.541] |
| **HELOC** | | | | |
| | Gemma-4B | 0.139 [0.536] | 0.194 [0.388] | 0.024 [0.914] |
| | Gemma-1B | 1.000 [—] | 1.000 [—] | 0.196 [0.371] |
| | GPT-4.1-mini | -0.229 [0.293] | 0.173 [0.430] | 0.060 [0.785] |
| | DeepSeek-Llama-8B | -0.005 [0.984] | 0.265 [0.287] | 0.123 [0.578] |
| | Gemini-2.5-Pro | 0.251 [0.248] | 0.696* [0.000] | 0.275 [0.205] |
| | Llama3-3B | -0.218 [0.317] | 0.133 [0.544] | -0.009 [0.968] |
| | Llama3-8B | 0.156 [0.564] | -0.097 [0.721] | 0.199 [0.364] |
| | Qwen3-8B | -0.170 [0.438] | 0.042 [0.847] | 0.410 [0.052] |
| | Mistral-7B | 0.462* [0.030] | 0.263 [0.238] | 0.190 [0.386] |
| **Iris** | | | | |
| | Llama3-8B | 0.200 [0.800] | 0.400 [0.600] | 0.800 [0.200] |
| | DeepSeek-Llama-8B | -0.800 [0.200] | -0.600 [0.400] | 0.000 [1.000] |
| | Mistral-7B | 0.800 [0.200] | -0.600 [0.400] | -0.800 [0.200] |
| | Gemini-2.5-Pro | 0.800 [0.200] | 1.000* [0.000] | 0.800 [0.200] |
| | Gemma-1B | 0.600 [0.400] | -0.600 [0.400] | -1.000* [0.000] |
| | Gemma-4B | 0.400 [0.600] | -0.600 [0.400] | 0.400 [0.600] |
| | Qwen3-8B | 0.400 [0.600] | 1.000* [0.000] | 0.400 [0.600] |
| | GPT-4.1-mini | 0.800 [0.200] | 1.000* [0.000] | 0.800 [0.200] |
| | Llama3-3B | -0.600 [0.400] | -0.600 [0.400] | 1.000* [0.000] |
| **Monk1** | | | | |
| | DeepSeek-Llama-8B | 0.200 [0.704] | 0.676 [0.140] | 0.845* [0.034] |
| | Gemini-2.5-Pro | 0.943* [0.005] | 0.372 [0.468] | 0.541 [0.268] |
| | Llama3-3B | -0.943* [0.005] | -0.101 [0.848] | 0.338 [0.512] |
| | Mistral-7B | 1.000* [0.000] | 0.500 [0.667] | -0.068 [0.899] |
| | Llama3-8B | -0.657 [0.156] | 0.372 [0.468] | 0.034 [0.949] |
| | GPT-4.1-mini | 0.314 [0.544] | 0.372 [0.468] | -0.135 [0.798] |
| | Gemma-4B | -0.900* [0.037] | -0.112 [0.858] | -0.372 [0.468] |
| | Gemma-1B | -0.200 [0.704] | -0.101 [0.848] | -0.778 [0.069] |
| | Qwen3-8B | 0.500 [0.667] | -1.000* [0.000] | -0.169 [0.749] |
| **Monk2** | | | | |
| | Gemini-2.5-Pro | -0.371 [0.468] | 0.030 [0.954] | 0.152 [0.774] |
| | Llama3-8B | 0.886* [0.019] | 0.030 [0.954] | -0.030 [0.954] |
| | Gemma-1B | -0.771 [0.072] | 0.030 [0.954] | -0.638 [0.173] |
| | Gemma-4B | 0.771 [0.072] | 0.030 [0.954] | -0.152 [0.774] |
| | DeepSeek-Llama-8B | 0.000 [1.000] | -0.316 [0.684] | -0.395 [0.439] |
| | Mistral-7B | -0.657 [0.156] | 0.030 [0.954] | 0.516 [0.295] |
| | Qwen3-8B | 0.314 [0.544] | -0.091 [0.864] | -0.030 [0.954] |
| | GPT-4.1-mini | 0.771 [0.072] | 0.030 [0.954] | -0.213 [0.686] |
| | Llama3-3B | 0.086 [0.872] | 0.030 [0.954] | 0.030 [0.954] |
| **Monk3** | | | | |
| | Mistral-7B | 0.500 [0.667] | 0.500 [0.667] | -0.395 [0.439] |
| | Gemma-1B | -0.200 [0.704] | -0.395 [0.439] | 0.334 [0.518] |
| | Gemma-4B | 0.500 [0.391] | -0.447 [0.450] | 0.516 [0.295] |
| | Llama3-8B | 0.143 [0.787] | 0.395 [0.439] | -0.152 [0.774] |
| | GPT-4.1-mini | -0.600 [0.208] | 0.395 [0.439] | -0.395 [0.439] |
| | Qwen3-8B | 1.000* [0.000] | 0.500 [0.667] | 0.516 [0.295] |
| | Gemini-2.5-Pro | -0.200 [0.704] | 0.577 [0.231] | 0.334 [0.518] |
| | DeepSeek-Llama-8B | 0.714 [0.111] | 0.395 [0.439] | 0.395 [0.439] |
| | Llama3-3B | -0.143 [0.787] | -0.395 [0.439] | -0.516 [0.295] |
| **Pima Diabetes** | | | | |
| | Gemini-2.5-Pro | 0.429 [0.289] | 0.976* [0.000] | 0.571 [0.139] |
| | Gemma-1B | -0.429 [0.289] | 0.429 [0.289] | 0.238 [0.570] |
| | Gemma-4B | 0.333 [0.420] | 0.095 [0.823] | -0.571 [0.139] |
| | Qwen3-8B | -0.429 [0.289] | 0.857* [0.007] | -0.286 [0.493] |
| | GPT-4.1-mini | -0.095 [0.823] | 0.905* [0.002] | -0.333 [0.420] |
| | DeepSeek-Llama-8B | 0.952* [0.000] | -0.286 [0.493] | -0.286 [0.493] |
| | Llama3-3B | -0.393 [0.383] | -0.036 [0.939] | -0.286 [0.493] |
| | Llama3-8B | -0.381 [0.352] | -0.262 [0.531] | -0.286 [0.493] |
| | Mistral-7B | -0.400 [0.505] | -0.300 [0.624] | 0.048 [0.911] |

*Notes.* Dashes (—) denote undefined due to NaNs or degenerate ranks. Stars: * $p<.05$, † $p<.10$. Brackets show $p$-values.

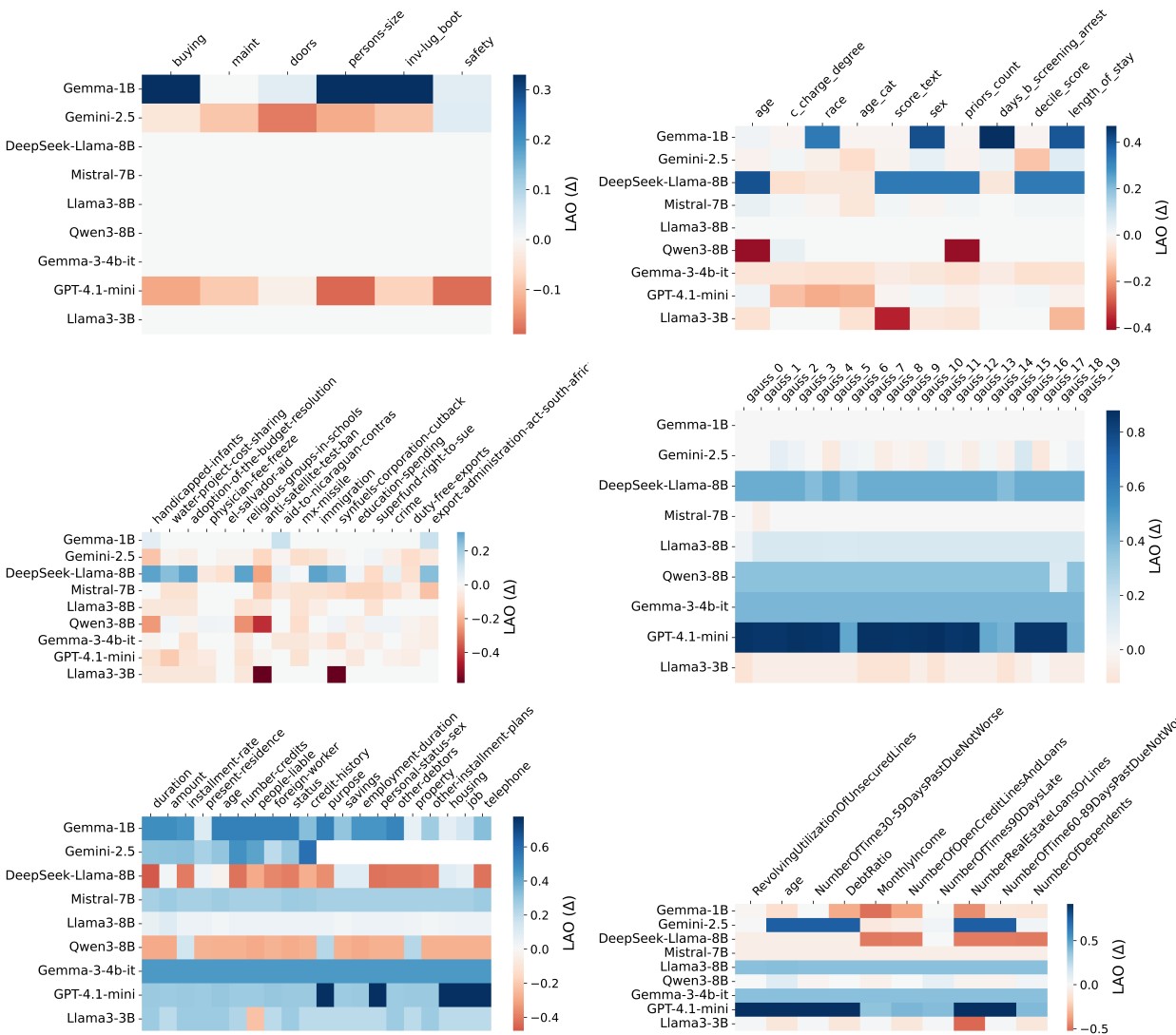

Figure 13: Heatmap of LAO performance ($\Delta_{\mathrm{LAO}}$) for each feature (columns) and LLM (rows). Darker blue indicates a larger performance loss when the feature is removed (higher importance); red indicates a slight performance gain or negligible reliance.

| Model | $\sigma_{\mathrm{LAO}}$ | SELF-FAITH | SELFATT@$k$ |
|---|---|---|---|
| Gemma-1B | 0.24 | $-1.00$ (0.00) | 0.33 |
| Gemini-2.5-Pro | 0.04 | $-0.61$ (0.15) | 0.78 |
| DeepSeek-Llama-8B | 0.23 | 0.81 (0.01) | 0.89 |
| Llama3-8B | 0.02 | $-0.18$ (0.64) | 1.00 |
| Qwen3-8B | 0.16 | 0.33 (0.42) | 0.89 |
| GPT-4.1-mini | 0.04 | 0.22 (0.58) | 1.00 |
| Llama3-3B | 0.11 | $-0.93$ (0.00) | 0.78 |
| Mistral-7B | 0.09 | 0.15 (0.70) | 1.00 |

Table 6: Decision faithfulness metrics between self-attribution rank ($\pi_{\mathrm{self}}$) and LAO-attribution rank ($\pi_{\mathrm{LAO}}$); Breast Cancer Dataset. NaN indicates $\pi_{\mathrm{self}}$ empty.

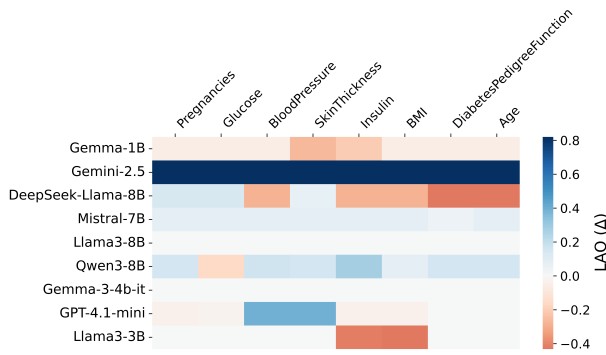

Figure 14: Heatmap of LAO performance ($\Delta_{\mathrm{LAO}}$) for each feature (columns) and LLM (rows). Darker blue indicates a larger performance loss when the feature is removed (higher importance); red indicates a slight performance gain or negligible reliance.

Figure 15: Heatmap of LAO performance ($\Delta_{\mathrm{LAO}}$) for each feature (columns) of `Car Evaluation` and LLM (rows). Darker blue indicates a larger performance loss when the feature is removed (higher importance); red indicates a slight performance gain or negligible reliance.

| Model | $\sigma_{\text{LAO}}$ | SELF-FAITH | SELFATT@$k$ |
|---|---|---|---|
| Gemma-1B | 0.17 | $-0.60$ (0.21) | 1.00 |
| Gemini-2.5-Pro | 0.07 | 0.66 (0.16) | 1.00 |
| DeepSeek-Llama-8B | 0.00 | 0.03 (0.96) | 1.00 |
| Llama3-8B | 0.00 | 0.31 (0.54) | 1.00 |
| Qwen3-8B | 0.00 | 0.54 (0.27) | 1.00 |
| GPT-4.1-mini | 0.07 | $-0.60$ (0.21) | 1.00 |
| Llama3-3B | 0.00 | $-1.00$ (NaN) | 0.33 |

Table 7: Decision faithfulness metrics between self-attribution rank ($\pi_{\text{self}}$) and LAO-attribution rank ($\pi_{\text{LAO}}$); Car Evaluation Dataset. NaN indicates $\pi_{\text{self}}$ empty.

| Model | $\sigma_{\text{LAO}}$ | SELF-FAITH | SELFATT@$k$ |
|---|---|---|---|
| Gemini-2.5-Pro | 0.06 | $-0.58$ (0.08) | 1.00 |
| Gemma-1B | 0.21 | $-0.45$ (0.26) | 0.80 |
| DeepSeek-Llama-8B | 0.21 | 0.03 (0.94) | 1.00 |
| Llama3-8B | 0.00 | 0.03 (0.93) | 0.90 |
| Qwen3-8B | 0.18 | $-0.45$ (0.19) | 1.00 |
| GPT-4.1-mini | 0.08 | 0.21 (0.56) | 1.00 |
| Llama3-3B | 0.11 | $-0.09$ (0.87) | 0.60 |

Table 8: Decision faithfulness metrics between self-attribution rank ($\pi_{\text{self}}$) and LAO-attribution rank ($\pi_{\text{LAO}}$); COMPAS Dataset. NaN indicates $\pi_{\text{self}}$ empty.

| Model | $\sigma_{\text{LAO}}$ | SELF-FAITH | SELFATT@$k$ |
|---|---|---|---|
| Gemma-1B | 0.04 | 0.64 (0.12) | 0.44 |
| Gemini-2.5-Pro | 0.05 | 0.44 (0.09) | 1.00 |
| DeepSeek-Llama-8B | 0.19 | 0.25 (0.39) | 0.88 |
| Llama3-8B | 0.04 | $-0.24$ (0.38) | 0.94 |
| Qwen3-8B | 0.13 | 0.01 (0.96) | 0.94 |
| GPT-4.1-mini | 0.04 | 0.23 (0.40) | 1.00 |
| Llama3-3B | 0.19 | $-0.38$ (0.15) | 1.00 |

Table 9: Decision faithfulness metrics between self-attribution rank ($\pi_{\text{self}}$) and LAO-attribution rank ($\pi_{\text{LAO}}$); Congression Voting Record Dataset. NaN indicates $\pi_{\text{self}}$ empty.

| Model | $\sigma_{\text{LAO}}$ | SELF-FAITH | SELFATT@$k$ |
|---|---|---|---|
| Gemma-1B | 0.00 | 0.60 (0.40) | 1.00 |
| Gemini-2.5-Pro | 0.03 | 0.80 (0.20) | 1.00 |
| DeepSeek-Llama-8B | 0.15 | $-0.80$ (0.20) | 1.00 |
| Llama3-8B | 0.10 | 0.20 (0.80) | 1.00 |
| Qwen3-8B | 0.06 | 0.40 (0.60) | 1.00 |
| GPT-4.1-mini | 0.10 | 0.80 (0.20) | 1.00 |
| Llama3-3B | 0.00 | $-0.60$ (0.40) | 1.00 |

Table 10: Decision faithfulness metrics between self-attribution rank ($\pi_{\text{self}}$) and LAO-attribution rank ($\pi_{\text{LAO}}$); Iris Dataset. NaN indicates $\pi_{\text{self}}$ empty.

| Model | $\sigma_{\text{LAO}}$ | SELF-FAITH | SELFATT@$k$ |
|---|---|---|---|
| Gemma-1B | 0.05 | $-0.20$ (0.70) | 1.00 |
| Gemini-2.5-Pro | 0.09 | 0.94 (0.00) | 1.00 |
| DeepSeek-Llama-8B | 0.18 | 0.20 (0.70) | 1.00 |
| Llama3-8B | 0.02 | $-0.66$ (0.16) | 1.00 |
| Qwen3-8B | 0.21 | 0.50 (0.67) | 0.50 |
| GPT-4.1-mini | 0.07 | 0.31 (0.54) | 1.00 |
| Llama3-3B | 0.00 | $-0.94$ (0.00) | 1.00 |

Table 11: Decision faithfulness metrics between self-attribution rank ($\pi_{\text{self}}$) and LAO-attribution rank ($\pi_{\text{LAO}}$); Monk-1 Dataset. NaN indicates $\pi_{\text{self}}$ empty.

| **Adult Income** — Zero-shot | | | | | | |
|---|---|---|---|---|---|---|
| Model | Acc | Macro-F1 | PenAcc | Len-F1 | UnkLbl% | Set-Jacc |
| Gemma-1B | 0.010 | 0.020 | 0.000 | 0.198 | 81.8 | 0.182 |
| Gemma-4B | 0.490 | 0.428 | 0.194 | 0.408 | 0.0 | 1.000 |
| Llama3-8B | 0.530 | 0.530 | 0.236 | 0.412 | 0.0 | 1.000 |
| Mistral-7B | 0.410 | 0.377 | 0.402 | 0.985 | 0.0 | 1.000 |
| GPT-4.1-mini | 0.470 | 0.320 | 0.470 | 1.000 | 0.0 | 1.000 |
| Llama3-3B | 0.010 | 0.020 | 0.000 | 0.020 | 0.0 | 0.500 |
| Gemini-2.5-Pro | 0.700 | 0.688 | 0.700 | 1.000 | 0.0 | 1.000 |
| Qwen3-8B | 0.380 | 0.433 | 0.247 | 0.734 | 0.0 | 1.000 |
| DeepSeek-Llama-8B | 0.500 | 0.333 | 0.198 | 0.397 | 0.0 | 0.500 |
| **Adult Income** — Few-shot | | | | | | |
| Model | Acc | Macro-F1 | PenAcc | Len-F1 | UnkLbl% | Set-Jacc |
| Gemma-4B | 0.053 | 0.083 | 0.000 | 0.100 | 0.0 | 0.500 |
| Gemma-1B | 0.474 | 0.321 | 0.461 | 0.974 | 0.0 | 1.000 |
| Qwen3-8B | 0.579 | 0.367 | 0.579 | 1.000 | 0.0 | 0.500 |
| Llama3-3B | 0.579 | 0.367 | 0.104 | 0.049 | 0.0 | 0.500 |
| Gemini-2.5-Pro | 0.737 | 0.708 | 0.737 | 1.000 | 0.0 | 1.000 |
| Llama3-8B | 0.579 | 0.367 | 0.407 | 0.655 | 0.0 | 0.500 |
| GPT-4.1-mini | 0.632 | 0.614 | 0.632 | 1.000 | 0.0 | 1.000 |
| DeepSeek-Llama-8B | 0.421 | 0.296 | 0.421 | 1.000 | 0.0 | 0.500 |
| Mistral-7B | 0.053 | 0.083 | 0.000 | 0.100 | 0.0 | 0.500 |

Table 12: Adult Income: STaDS metrics per model for zero-shot and few-shot. Acc = accuracy; PenAcc = penalised accuracy; Len-F1 = length F1; UnkLbl% = unknown label rate (%).

| Breast — Zero-shot | | | | | | |
|---|---|---|---|---|---|---|
| Model | Acc | Macro-F1 | PenAcc | Len-F1 | UnkLbl% | Set-Jacc |
| DeepSeek-Llama-8B | 0.697 | 0.433 | 0.365 | 0.337 | 0.0 | 1.000 |
| Gemini-2.5-Pro | 0.650 | 0.575 | 0.646 | 0.993 | 0.0 | 1.000 |
| Gemma-1B | 0.708 | 0.414 | 0.376 | 0.337 | 0.0 | 0.500 |
| Gemma-4B | 0.000 | 0.000 | 0.000 | 0.007 | 1.0 | 0.000 |
| Mistral-7B | 0.690 | 0.408 | 0.358 | 0.337 | 0.0 | 1.000 |
| GPT-4.1-mini | 0.729 | 0.524 | 0.348 | 0.238 | 0.0 | 1.000 |
| Llama3-8B | 0.704 | 0.413 | 0.373 | 0.337 | 0.0 | 1.000 |
| Qwen3-8B | 0.224 | 0.296 | 0.000 | 0.478 | 0.0 | 1.000 |
| Llama3-3B | 0.708 | 0.414 | 0.376 | 0.337 | 0.0 | 0.500 |

| Breast — Few-shot | | | | | | |
|---|---|---|---|---|---|---|
| Model | Acc | Macro-F1 | PenAcc | Len-F1 | UnkLbl% | Set-Jacc |
| Mistral-7B | 0.036 | 0.082 | 0.000 | 0.133 | 0.0 | 1.000 |
| GPT-4.1-mini | 0.732 | 0.525 | 0.732 | 1.000 | 0.0 | 1.000 |
| Qwen3-8B | 0.679 | 0.491 | 0.218 | 0.079 | 0.0 | 1.000 |
| Gemini-2.5-Pro | 0.589 | 0.548 | 0.589 | 1.000 | 0.0 | 1.000 |
| Llama3-8B | 0.732 | 0.525 | 0.272 | 0.079 | 0.0 | 1.000 |
| DeepSeek-Llama-8B | 0.554 | 0.495 | 0.093 | 0.079 | 0.0 | 1.000 |
| Gemma-4B | 0.714 | 0.470 | 0.254 | 0.079 | 0.0 | 1.000 |
| Gemma-1B | 0.714 | 0.417 | 0.714 | 1.000 | 0.0 | 0.500 |
| Llama3-3B | 0.107 | 0.120 | 0.000 | 0.303 | 0.0 | 0.500 |

Table 13: Breast: STaDS metrics per model for zero-shot and few-shot. Acc = accuracy; PenAcc = penalised accuracy; Len-F1 = length F1; UnkLbl% = unknown label rate (%).

| Car evaluation — Zero-shot | | | | | | |
|---|---|---|---|---|---|---|
| Model | Acc | Macro-F1 | PenAcc | Len-F1 | UnkLbl% | Set-Jacc |
| Gemini-2.5-Pro | 0.419 | 0.243 | 0.406 | 0.973 | 0.0 | 0.500 |
| Gemma-4B | 0.326 | 0.123 | 0.045 | 0.439 | 0.0 | 0.250 |
| Gemma-1B | 0.005 | 0.008 | 0.000 | 0.036 | 0.0 | 1.000 |
| GPT-4.1-mini | 0.326 | 0.129 | 0.000 | 0.316 | 0.0 | 0.750 |
| DeepSeek-Llama-8B | 0.263 | 0.219 | 0.000 | 0.439 | 0.0 | 1.000 |
| Mistral-7B | 0.279 | 0.215 | 0.000 | 0.439 | 0.0 | 0.750 |
| Llama3-3B | 0.000 | 0.000 | 0.000 | 0.000 | 0.0 | 0.000 |
| Qwen3-8B | 0.021 | 0.030 | 0.000 | 0.410 | 0.0 | 0.500 |
| Llama3-8B | 0.326 | 0.123 | 0.045 | 0.439 | 0.0 | 0.250 |

| Car evaluation — Few-shot | | | | | | |
|---|---|---|---|---|---|---|
| Model | Acc | Macro-F1 | PenAcc | Len-F1 | UnkLbl% | Set-Jacc |
| Mistral-7B | 0.333 | 0.125 | 0.000 | 0.104 | 0.0 | 0.250 |
| Gemma-1B | 0.333 | 0.126 | 0.330 | 0.993 | 0.0 | 0.250 |
| Gemma-4B | 0.293 | 0.113 | 0.000 | 0.104 | 0.0 | 0.500 |
| DeepSeek-Llama-8B | 0.333 | 0.125 | 0.000 | 0.104 | 0.0 | 0.250 |
| Llama3-8B | 0.013 | 0.019 | 0.000 | 0.026 | 0.0 | 0.250 |
| Gemini-2.5-Pro | 0.600 | 0.616 | 0.586 | 0.973 | 0.0 | 1.000 |
| Qwen3-8B | 0.307 | 0.117 | 0.000 | 0.104 | 0.0 | 0.250 |
| Llama3-3B | 0.000 | 0.000 | 0.000 | 0.000 | 0.0 | 0.000 |
| GPT-4.1-mini | 0.173 | 0.079 | 0.160 | 0.974 | 0.0 | 0.750 |

Table 14: Car evaluation: STaDS metrics per model for zero-shot and few-shot. Acc = accuracy; PenAcc = penalised accuracy; Len-F1 = length F1; UnkLbl% = unknown label rate (%).

| **COMPAS** — Zero-shot | | | | | | |
|---|---|---|---|---|---|---|
| Model | Acc | Macro-F1 | PenAcc | Len-F1 | UnkLbl% | Set-Jacc |
| Gemma-1B | 0.502 | 0.338 | 0.270 | 0.536 | 0.0 | 1.000 |
| Gemma-4B | 0.500 | 0.495 | 0.268 | 0.536 | 0.0 | 1.000 |
| Qwen3-8B | 0.490 | 0.329 | 0.258 | 0.536 | 0.0 | 1.000 |
| Gemini-2.5-Pro | 0.816 | 0.810 | 0.571 | 0.510 | 0.0 | 1.000 |
| GPT-4.1-mini | 0.468 | 0.319 | 0.164 | 0.392 | 0.0 | 1.000 |
| Llama3-8B | 0.500 | 0.333 | 0.268 | 0.536 | 0.0 | 0.500 |
| Mistral-7B | 0.006 | 0.012 | 0.000 | 0.016 | 0.0 | 1.000 |
| Llama3-3B | 0.500 | 0.333 | 0.268 | 0.536 | 0.0 | 0.500 |
| DeepSeek-Llama-8B | 0.502 | 0.338 | 0.270 | 0.536 | 0.0 | 1.000 |
| **COMPAS** — Few-shot | | | | | | |
| Model | Acc | Macro-F1 | PenAcc | Len-F1 | UnkLbl% | Set-Jacc |
| Llama3-8B | 0.511 | 0.338 | 0.072 | 0.121 | 0.0 | 0.500 |
| Mistral-7B | 0.500 | 0.333 | 0.061 | 0.121 | 0.0 | 1.000 |
| Gemma-4B | 0.011 | 0.022 | 0.000 | 0.022 | 0.0 | 0.500 |
| GPT-4.1-mini | 0.500 | 0.355 | 0.476 | 0.952 | 0.0 | 1.000 |
| Gemma-1B | 0.500 | 0.333 | 0.471 | 0.941 | 0.0 | 1.000 |
| Qwen3-8B | 0.511 | 0.338 | 0.072 | 0.121 | 0.0 | 0.500 |
| Llama3-3B | 0.011 | 0.022 | 0.000 | 0.108 | 0.0 | 1.000 |
| Gemini-2.5-Pro | 0.716 | 0.714 | 0.697 | 0.962 | 0.0 | 1.000 |
| DeepSeek-Llama-8B | 0.409 | 0.321 | 0.406 | 0.994 | 0.0 | 1.000 |

Table 15: COMPAS: STaDS metrics per model for zero-shot and few-shot. Acc = accuracy; PenAcc = penalised accuracy; Len-F1 = length F1; UnkLbl% = unknown label rate (%).

| **Congression Vote** — Zero-shot | | | | | | |
|---|---|---|---|---|---|---|
| Model | Acc | Macro-F1 | PenAcc | Len-F1 | UnkLbl% | Set-Jacc |
| Mistral-7B | 0.526 | 0.524 | 0.171 | 0.291 | 0.0 | 1.000 |
| Gemini-2.5-Pro | 0.397 | 0.395 | 0.388 | 0.982 | 0.0 | 1.000 |
| Llama3-3B | 0.534 | 0.348 | 0.180 | 0.291 | 0.0 | 0.500 |
| Llama3-8B | 0.478 | 0.478 | 0.124 | 0.291 | 0.0 | 1.000 |
| Qwen3-8B | 0.474 | 0.360 | 0.119 | 0.291 | 0.0 | 1.000 |
| GPT-4.1-mini | 0.409 | 0.401 | 0.011 | 0.204 | 0.0 | 1.000 |
| DeepSeek-Llama-8B | 0.466 | 0.325 | 0.111 | 0.291 | 0.0 | 1.000 |
| Gemma-1B | 0.263 | 0.274 | 0.062 | 0.598 | 0.0 | 0.500 |
| Gemma-4B | 0.526 | 0.525 | 0.171 | 0.291 | 0.0 | 1.000 |
| **Congression Vote** — Few-shot | | | | | | |
| Model | Acc | Macro-F1 | PenAcc | Len-F1 | UnkLbl% | Set-Jacc |
| Llama3-3B | 0.064 | 0.086 | 0.000 | 0.351 | 0.0 | 0.500 |
| GPT-4.1-mini | 0.532 | 0.451 | 0.521 | 0.978 | 0.0 | 1.000 |
| Gemini-2.5-Pro | 0.638 | 0.636 | 0.638 | 1.000 | 0.0 | 1.000 |
| Qwen3-8B | 0.489 | 0.478 | 0.023 | 0.067 | 0.0 | 1.000 |
| DeepSeek-Llama-8B | 0.489 | 0.390 | 0.311 | 0.644 | 0.0 | 1.000 |
| Mistral-7B | 0.468 | 0.467 | 0.001 | 0.067 | 0.0 | 1.000 |
| Llama3-8B | 0.404 | 0.402 | 0.000 | 0.067 | 0.0 | 1.000 |
| Gemma-4B | 0.447 | 0.447 | 0.000 | 0.067 | 0.0 | 1.000 |
| Gemma-1B | 0.532 | 0.347 | 0.532 | 1.000 | 0.0 | 0.500 |

Table 16: Congression Vote: STaDS metrics per model for zero-shot and few-shot. Acc = accuracy; PenAcc = penalised accuracy; Len-F1 = length F1; UnkLbl% = unknown label rate (%).

| Synthetic Gaussian — Zero-shot | | | | | | |
|---|---|---|---|---|---|---|
| Model | Acc | Macro-F1 | PenAcc | Len-F1 | UnkLbl% | Set-Jacc |
| Gemma-1B | 0.500 | 0.500 | 0.068 | 0.137 | 0.0 | 1.000 |
| Gemma-4B | 0.500 | 0.487 | 0.068 | 0.137 | 0.0 | 1.000 |
| GPT-4.1-mini | 0.390 | 0.281 | 0.390 | 1.000 | 0.0 | 1.000 |
| Qwen3-8B | 0.510 | 0.355 | 0.078 | 0.137 | 0.0 | 1.000 |
| Llama3-8B | 0.500 | 0.500 | 0.068 | 0.137 | 0.0 | 1.000 |
| Gemini-2.5-Pro | 0.550 | 0.448 | 0.550 | 1.000 | 0.0 | 1.000 |
| DeepSeek-Llama-8B | 0.350 | 0.398 | 0.288 | 0.876 | 0.0 | 1.000 |
| Mistral-7B | 0.500 | 0.500 | 0.068 | 0.137 | 0.0 | 1.000 |
| Llama3-3B | 0.000 | 0.000 | 0.000 | 0.000 | 0.0 | 0.000 |
| Synthetic Gaussian — Few-shot | | | | | | |
| Model | Acc | Macro-F1 | PenAcc | Len-F1 | UnkLbl% | Set-Jacc |
| Llama3-8B | 0.440 | 0.306 | 0.273 | 0.667 | 0.0 | 1.000 |
| Gemma-4B | 0.440 | 0.306 | 0.403 | 0.926 | 0.0 | 0.500 |
| Gemma-1B | 0.400 | 0.384 | 0.000 | 0.036 | 0.0 | 1.000 |
| Mistral-7B | 0.000 | 0.000 | 0.000 | 0.958 | 95.7 | 0.042 |
| GPT-4.1-mini | 0.880 | 0.873 | 0.880 | 1.000 | 0.0 | 1.000 |
| Gemini-2.5-Pro | 0.680 | 0.603 | 0.680 | 1.000 | 0.0 | 1.000 |
| Qwen3-8B | 0.360 | 0.359 | 0.360 | 1.000 | 0.0 | 1.000 |
| Llama3-3B | 0.000 | 0.000 | 0.000 | 0.000 | 0.0 | 0.000 |
| DeepSeek-Llama-8B | 0.440 | 0.306 | 0.440 | 1.000 | 0.0 | 0.500 |

Table 17: Synthetic Gaussian: STaDS metrics per model for zero-shot and few-shot. Acc = accuracy; PenAcc = penalised accuracy; Len-F1 = length F1; UnkLbl% = unknown label rate (%).

| German Credit Risk — Zero-shot | | | | | | |
|---|---|---|---|---|---|---|
| Model | Acc | Macro-F1 | PenAcc | Len-F1 | UnkLbl% | Set-Jacc |
| GPT-4.1-mini | 0.660 | 0.616 | 0.660 | 1.000 | 0.00 | 1.000 |
| Mistral-7B | 0.550 | 0.436 | 0.118 | 0.137 | 0.00 | 1.000 |
| Llama3-8B | 0.500 | 0.500 | 0.068 | 0.137 | 0.00 | 1.000 |
| DeepSeek-Llama-8B | 0.490 | 0.329 | 0.058 | 0.137 | 0.00 | 1.000 |
| Gemma-1B | 0.490 | 0.490 | 0.058 | 0.137 | 0.00 | 1.000 |
| Gemini-2.5-Pro | 0.460 | 0.457 | 0.460 | 1.000 | 0.00 | 1.000 |
| Qwen3-8B | 0.020 | 0.038 | 0.000 | 0.387 | 0.00 | 1.000 |
| Llama3-3B | 0.000 | 0.000 | 0.000 | 0.000 | 0.00 | 0.000 |
| Gemma-4B | 0.000 | 0.000 | 0.000 | 0.020 | 1.00 | 0.000 |
| German Credit Risk — Few-shot | | | | | | |
| Model | Acc | Macro-F1 | PenAcc | Len-F1 | UnkLbl% | Set-Jacc |
| Mistral-7B | 0.333 | 0.325 | 0.333 | 1.000 | 0.0 | 1.000 |
| DeepSeek-Llama-8B | 0.444 | 0.308 | 0.094 | 0.300 | 0.0 | 1.000 |
| Llama3-8B | 0.222 | 0.222 | 0.096 | 0.947 | 20.0 | 0.333 |
| Llama3-3B | 0.778 | 0.438 | 0.284 | 0.013 | 0.0 | 0.500 |
| GPT-4.1-mini | 0.778 | 0.679 | 0.778 | 1.000 | 0.0 | 1.000 |
| Gemini-2.5-Pro | 0.889 | 0.862 | 0.889 | 1.000 | 0.0 | 1.000 |
| Gemma-4B | 0.444 | 0.444 | 0.444 | 1.000 | 0.0 | 1.000 |
| Qwen3-8B | 0.222 | 0.182 | 0.222 | 1.000 | 0.0 | 0.500 |
| Gemma-1B | 0.556 | 0.357 | 0.529 | 0.947 | 0.0 | 1.000 |

Table 18: German Credit Risk: STaDS metrics per model for zero-shot and few-shot. Acc = accuracy; PenAcc = penalised accuracy; Len-F1 = length F1; UnkLbl% = unknown label rate (%).

| **Give Me Some Credit** — Zero-shot | | | | | | |
|---|---|---|---|---|---|---|
| Model | Acc | Macro-F1 | PenAcc | Len-F1 | UnkLbl% | Set-Jacc |
| GPT-4.1-mini | 0.550 | 0.533 | 0.492 | 0.885 | 0.0 | 1.000 |
| Llama3-8B | 0.470 | 0.320 | 0.038 | 0.137 | 0.0 | 1.000 |
| DeepSeek-Llama-8B | 0.830 | 0.832 | 0.825 | 0.990 | 0.0 | 1.000 |
| Mistral-7B | 0.480 | 0.324 | 0.048 | 0.137 | 0.0 | 1.000 |
| Llama3-3B | 0.500 | 0.333 | 0.068 | 0.137 | 0.0 | 0.500 |
| Gemini-2.5-Pro | 0.790 | 0.785 | 0.790 | 1.000 | 0.0 | 1.000 |
| Qwen3-8B | 0.570 | 0.646 | 0.511 | 0.883 | 0.0 | 1.000 |
| Gemma-4B | 0.440 | 0.436 | 0.303 | 0.726 | 0.0 | 1.000 |
| Gemma-1B | 0.480 | 0.327 | 0.477 | 0.995 | 0.0 | 1.000 |
| **Give Me Some Credit** — Few-shot | | | | | | |
| Model | Acc | Macro-F1 | PenAcc | Len-F1 | UnkLbl% | Set-Jacc |
| Qwen3-8B | 0.500 | 0.500 | 0.500 | 1.000 | 0.0 | 1.000 |
| Llama3-3B | 0.500 | 0.333 | 0.017 | 0.035 | 0.0 | 0.500 |
| DeepSeek-Llama-8B | 0.542 | 0.420 | 0.059 | 0.035 | 0.0 | 1.000 |
| Gemini-2.5-Pro | 0.750 | 0.743 | 0.750 | 1.000 | 0.0 | 1.000 |
| Llama3-8B | 0.500 | 0.333 | 0.500 | 1.000 | 0.0 | 0.500 |
| Mistral-7B | 0.500 | 0.333 | 0.017 | 0.035 | 0.0 | 0.500 |
| GPT-4.1-mini | 0.917 | 0.916 | 0.917 | 1.000 | 0.0 | 1.000 |
| Gemma-1B | 0.417 | 0.294 | 0.000 | 0.035 | 0.0 | 1.000 |
| Gemma-4B | 0.500 | 0.438 | 0.387 | 0.774 | 0.0 | 1.000 |

Table 19: Give Me Some Credit: STaDS metrics per model for zero-shot and few-shot. Acc = accuracy; PenAcc = penalised accuracy; Len-F1 = length F1; UnkLbl% = unknown label rate (%).

| **Heart Disease** — Zero-shot | | | | | | |
|---|---|---|---|---|---|---|
| Model | Acc | Macro-F1 | PenAcc | Len-F1 | UnkLbl% | Set-Jacc |
| Qwen3-8B | 0.420 | 0.484 | 0.352 | 0.864 | 0.0 | 1.000 |
| GPT-4.1-mini | 0.640 | 0.614 | 0.640 | 1.000 | 0.0 | 1.000 |
| Llama3-8B | 0.450 | 0.449 | 0.018 | 0.137 | 0.0 | 1.000 |
| Gemma-1B | 0.510 | 0.510 | 0.078 | 0.137 | 0.0 | 1.000 |
| Gemma-4B | 0.480 | 0.448 | 0.048 | 0.137 | 0.0 | 1.000 |
| Mistral-7B | 0.500 | 0.487 | 0.068 | 0.137 | 0.0 | 1.000 |
| DeepSeek-Llama-8B | 0.510 | 0.372 | 0.507 | 0.995 | 0.0 | 1.000 |
| Llama3-3B | 0.500 | 0.333 | 0.068 | 0.137 | 0.0 | 0.500 |
| Gemini-2.5-Pro | 0.550 | 0.544 | 0.550 | 1.000 | 0.0 | 1.000 |
| **Heart Disease** — Few-shot | | | | | | |
| Model | Acc | Macro-F1 | PenAcc | Len-F1 | UnkLbl% | Set-Jacc |
| Gemma-4B | 0.650 | 0.642 | 0.483 | 0.667 | 0.0 | 1.000 |
| Gemma-1B | 0.450 | 0.429 | 0.283 | 0.667 | 0.0 | 1.000 |
| GPT-4.1-mini | 0.700 | 0.697 | 0.700 | 1.000 | 0.0 | 1.000 |
| DeepSeek-Llama-8B | 0.600 | 0.375 | 0.600 | 1.000 | 0.0 | 0.500 |
| Llama3-8B | 0.650 | 0.642 | 0.650 | 1.000 | 0.0 | 1.000 |
| Llama3-3B | 0.600 | 0.375 | 0.600 | 1.000 | 0.0 | 0.500 |
| Gemini-2.5-Pro | 0.500 | 0.495 | 0.500 | 1.000 | 0.0 | 1.000 |
| Qwen3-8B | 0.350 | 0.307 | 0.350 | 1.000 | 0.0 | 1.000 |
| Mistral-7B | 0.600 | 0.375 | 0.114 | 0.029 | 0.0 | 0.500 |

Table 20: Heart Disease: STaDS metrics per model for zero-shot and few-shot. Acc = accuracy; PenAcc = penalised accuracy; Len-F1 = length F1; UnkLbl% = unknown label rate (%).

| HELOC — Zero-shot | | | | | | |
|---|---|---|---|---|---|---|
| Model | Acc | Macro-F1 | PenAcc | Len-F1 | UnkLbl% | Set-Jacc |
| Llama3-3B | 0.000 | 0.000 | 0.000 | 0.000 | 0.0 | 0.000 |
| Llama3-8B | 0.500 | 0.333 | 0.068 | 0.137 | 0.0 | 0.500 |
| GPT-4.1-mini | 0.650 | 0.601 | 0.650 | 1.000 | 0.0 | 1.000 |
| DeepSeek-Llama-8B | 0.510 | 0.372 | 0.507 | 0.995 | 0.0 | 1.000 |
| Mistral-7B | 0.500 | 0.500 | 0.068 | 0.137 | 0.0 | 1.000 |
| Qwen3-8B | 0.640 | 0.596 | 0.637 | 0.995 | 0.0 | 1.000 |
| Gemma-4B | 0.500 | 0.500 | 0.068 | 0.137 | 0.0 | 1.000 |
| Gemma-1B | 0.490 | 0.331 | 0.487 | 0.995 | 0.0 | 1.000 |
| Gemini-2.5-Pro | 0.670 | 0.670 | 0.670 | 1.000 | 0.0 | 1.000 |
| HELOC — Few-shot | | | | | | |
| Model | Acc | Macro-F1 | PenAcc | Len-F1 | UnkLbl% | Set-Jacc |
| Llama3-3B | 0.038 | 0.077 | 0.000 | 0.074 | 0.0 | 0.500 |
| Gemma-1B | 0.462 | 0.316 | 0.462 | 1.000 | 0.0 | 0.500 |
| Gemma-4B | 0.885 | 0.883 | 0.811 | 0.852 | 0.0 | 1.000 |
| Qwen3-8B | 0.538 | 0.513 | 0.538 | 1.000 | 0.0 | 1.000 |
| Llama3-8B | 0.462 | 0.324 | 0.452 | 0.980 | 0.0 | 0.500 |
| Gemini-2.5-Pro | 0.615 | 0.613 | 0.615 | 1.000 | 0.0 | 1.000 |
| GPT-4.1-mini | 0.885 | 0.880 | 0.885 | 1.000 | 0.0 | 1.000 |
| Mistral-7B | 0.038 | 0.077 | 0.000 | 0.143 | 0.0 | 1.000 |

Table 21: HELOC: STaDS metrics per model for zero-shot and few-shot. Acc = accuracy; PenAcc = penalised accuracy; Len-F1 = length F1; UnkLbl% = unknown label rate (%).

| Iris — Zero-shot | | | | | | |
|---|---|---|---|---|---|---|
| Model | Acc | Macro-F1 | PenAcc | Len-F1 | UnkLbl% | Set-Jacc |
| Llama3-3B | 0.007 | 0.013 | 0.000 | 0.039 | 0.0 | 1.000 |
| GPT-4.1-mini | 0.487 | 0.516 | 0.458 | 0.944 | 0.0 | 1.000 |
| Llama3-8B | 0.373 | 0.373 | 0.000 | 0.198 | 0.0 | 1.000 |
| Qwen3-8B | 0.440 | 0.402 | 0.039 | 0.198 | 0.0 | 1.000 |
| Mistral-7B | 0.320 | 0.266 | 0.000 | 0.198 | 0.0 | 1.000 |
| Gemma-1B | 0.007 | 0.013 | 0.000 | 0.997 | 98.0 | 0.020 |
| Gemma-4B | 0.360 | 0.360 | 0.000 | 0.198 | 0.0 | 1.000 |
| Gemini-2.5-Pro | 0.787 | 0.787 | 0.785 | 0.997 | 0.0 | 1.000 |
| DeepSeek-Llama-8B | 0.380 | 0.376 | 0.000 | 0.198 | 0.0 | 1.000 |
| Iris — Few-shot | | | | | | |
| Model | Acc | Macro-F1 | PenAcc | Len-F1 | UnkLbl% | Set-Jacc |
| Llama3-3B | 0.333 | 0.167 | 0.000 | 0.043 | 0.0 | 0.333 |
| Gemma-4B | 0.333 | 0.334 | 0.152 | 0.638 | 0.0 | 1.000 |
| Gemma-1B | 0.033 | 0.061 | 0.000 | 0.984 | 90.3 | 0.097 |
| Gemini-2.5-Pro | 1.000 | 1.000 | 1.000 | 1.000 | 0.0 | 1.000 |
| GPT-4.1-mini | 0.667 | 0.662 | 0.667 | 1.000 | 0.0 | 1.000 |
| Qwen3-8B | 0.933 | 0.933 | 0.933 | 1.000 | 0.0 | 1.000 |
| DeepSeek-Llama-8B | 0.367 | 0.330 | 0.367 | 1.000 | 0.0 | 1.000 |
| Llama3-8B | 0.300 | 0.295 | 0.170 | 0.741 | 0.0 | 1.000 |
| Mistral-7B | 0.367 | 0.355 | 0.000 | 0.043 | 0.0 | 1.000 |

Table 22: Iris: STaDS metrics per model for zero-shot and few-shot. Acc = accuracy; PenAcc = penalised accuracy; Len-F1 = length F1; UnkLbl% = unknown label rate (%).

| Monk 1 — Zero-shot | | | | | | |
|---|---|---|---|---|---|---|
| Model | Acc | Macro-F1 | PenAcc | Len-F1 | UnkLbl% | Set-Jacc |
| Qwen3-8B | 0.535 | 0.427 | 0.275 | 0.481 | 0.0 | 1.000 |
| Llama3-8B | 0.502 | 0.502 | 0.243 | 0.481 | 0.0 | 1.000 |
| Gemma-4B | 0.505 | 0.505 | 0.245 | 0.481 | 0.0 | 1.000 |
| Mistral-7B | 0.521 | 0.507 | 0.261 | 0.481 | 0.0 | 1.000 |
| Gemma-1B | 0.498 | 0.334 | 0.237 | 0.481 | 30.0 | 0.333 |
| Gemini-2.5-Pro | 0.620 | 0.613 | 0.618 | 0.995 | 0.0 | 1.000 |
| GPT-4.1-mini | 0.118 | 0.193 | 0.000 | 0.348 | 0.0 | 1.000 |
| DeepSeek-Llama-8B | 0.507 | 0.352 | 0.247 | 0.481 | 0.0 | 1.000 |
| Llama3-3B | 0.500 | 0.333 | 0.240 | 0.481 | 0.0 | 0.500 |
| Monk 1 — Few-shot | | | | | | |
| Model | Acc | Macro-F1 | PenAcc | Len-F1 | UnkLbl% | Set-Jacc |
| Gemma-1B | 0.034 | 0.059 | 0.000 | 0.168 | 0.0 | 1.000 |
| Gemma-4B | 0.011 | 0.023 | 0.000 | 0.023 | 0.0 | 0.500 |
| GPT-4.1-mini | 0.506 | 0.495 | 0.482 | 0.952 | 0.0 | 1.000 |
| Llama3-8B | 0.506 | 0.449 | 0.066 | 0.120 | 0.0 | 1.000 |
| Mistral-7B | 0.540 | 0.512 | 0.100 | 0.120 | 0.0 | 1.000 |
| Gemini-2.5-Pro | 0.724 | 0.724 | 0.721 | 0.994 | 0.0 | 1.000 |
| Llama3-3B | 0.494 | 0.331 | 0.054 | 0.120 | 0.0 | 1.000 |
| DeepSeek-Llama-8B | 0.483 | 0.476 | 0.043 | 0.120 | 0.0 | 1.000 |
| Qwen3-8B | 0.759 | 0.758 | 0.726 | 0.935 | 0.0 | 1.000 |

Table 23: Monk 1: STaDS metrics per model for zero-shot and few-shot. Acc = accuracy; PenAcc = penalised accuracy; Len-F1 = length F1; UnkLbl% = unknown label rate (%).

| Monk 2 — Zero-shot | | | | | | |
|---|---|---|---|---|---|---|
| Model | Acc | Macro-F1 | PenAcc | Len-F1 | UnkLbl% | Set-Jacc |
| Gemma-1B | 0.174 | 0.187 | 0.000 | 0.481 | 40.0 | 0.200 |
| Gemma-4B | 0.329 | 0.247 | 0.069 | 0.481 | 0.000 | 0.500 |
| Llama3-8B | 0.500 | 0.485 | 0.240 | 0.481 | 0.000 | 1.000 |
| DeepSeek-Llama-8B | 0.674 | 0.409 | 0.414 | 0.481 | 0.000 | 1.000 |
| Qwen3-8B | 0.664 | 0.476 | 0.405 | 0.481 | 0.000 | 1.000 |
| Mistral-7B | 0.556 | 0.498 | 0.296 | 0.481 | 0.000 | 1.000 |
| Llama3-3B | 0.021 | 0.030 | 0.000 | 0.045 | 0.000 | 0.500 |
| GPT-4.1-mini | 0.660 | 0.397 | 0.334 | 0.348 | 0.000 | 1.000 |
| Gemini-2.5-Pro | 0.484 | 0.475 | 0.461 | 0.955 | 0.000 | 1.000 |
| Monk 2 — Few-shot | | | | | | |
| Model | Acc | Macro-F1 | PenAcc | Len-F1 | UnkLbl% | Set-Jacc |
| Gemma-1B | 0.667 | 0.400 | 0.634 | 0.935 | 0.00 | 0.500 |
| Gemma-4B | 0.563 | 0.422 | 0.123 | 0.120 | 0.00 | 1.000 |
| Llama3-8B | 0.701 | 0.525 | 0.261 | 0.120 | 0.00 | 1.000 |
| DeepSeek-Llama-8B | 0.667 | 0.552 | 0.227 | 0.120 | 0.00 | 1.000 |
| Qwen3-8B | 0.713 | 0.601 | 0.273 | 0.120 | 0.00 | 1.000 |
| Mistral-7B | 0.678 | 0.436 | 0.238 | 0.120 | 0.00 | 1.000 |
| Llama3-3B | 0.667 | 0.400 | 0.227 | 0.120 | 0.00 | 0.500 |
| GPT-4.1-mini | 0.575 | 0.450 | 0.569 | 0.988 | 0.00 | 1.000 |
| Gemini-2.5-Pro | 0.494 | 0.456 | 0.489 | 0.989 | 0.00 | 1.000 |

Table 24: Monk 2: STaDS metrics per model for zero-shot and few-shot. Acc = accuracy; PenAcc = penalised accuracy; Len-F1 = length F1; UnkLbl% = unknown label rate (%).

| Monk 3 — Zero-shot | | | | | | |
|---|---|---|---|---|---|---|
| Model | Acc | Macro-F1 | PenAcc | Len-F1 | UnkLbl% | Set-Jacc |
| DeepSeek-Llama-8B | 0.512 | 0.511 | 0.252 | 0.481 | 0.00 | 1.000 |
| Llama3-3B | 0.472 | 0.321 | 0.213 | 0.481 | 0.00 | 0.500 |
| Gemini-2.5-Pro | 0.579 | 0.596 | 0.563 | 0.969 | 0.00 | 1.000 |
| GPT-4.1-mini | 0.507 | 0.423 | 0.181 | 0.348 | 0.00 | 1.000 |
| Llama3-8B | 0.495 | 0.495 | 0.236 | 0.481 | 0.00 | 1.000 |
| Gemma-4B | 0.528 | 0.345 | 0.268 | 0.481 | 0.00 | 0.500 |
| Gemma-1B | 0.528 | 0.345 | 0.268 | 0.481 | 0.00 | 0.500 |
| Mistral-7B | 0.500 | 0.474 | 0.240 | 0.481 | 0.00 | 1.000 |
| Qwen3-8B | 0.491 | 0.362 | 0.231 | 0.481 | 0.00 | 1.000 |
| Monk 3 — Few-shot | | | | | | |
| Model | Acc | Macro-F1 | PenAcc | Len-F1 | UnkLbl% | Set-Jacc |
| Llama3-8B | 0.494 | 0.494 | 0.054 | 0.120 | 0.00 | 1.000 |
| Gemini-2.5-Pro | 0.644 | 0.642 | 0.644 | 1.000 | 0.00 | 1.000 |
| Qwen3-8B | 0.517 | 0.516 | 0.077 | 0.120 | 0.00 | 1.000 |
| Llama3-3B | 0.460 | 0.315 | 0.020 | 0.120 | 0.00 | 1.000 |
| Mistral-7B | 0.437 | 0.335 | 0.000 | 0.120 | 0.00 | 1.000 |
| DeepSeek-Llama-8B | 0.437 | 0.432 | 0.000 | 0.120 | 0.00 | 1.000 |
| Gemma-1B | 0.425 | 0.351 | 0.388 | 0.926 | 0.00 | 1.000 |
| Gemma-4B | 0.529 | 0.346 | 0.089 | 0.120 | 0.00 | 0.500 |
| GPT-4.1-mini | 0.494 | 0.492 | 0.470 | 0.952 | 0.00 | 1.000 |

Table 25: Monk 3: STaDS metrics per model for zero-shot and few-shot. Acc = accuracy; PenAcc = penalised accuracy; Len-F1 = length F1; UnkLbl% = unknown label rate (%).

| Pima Indians Diabetes — Zero-shot | | | | | | |
|---|---|---|---|---|---|---|
| Model | Acc | Macro-F1 | PenAcc | Len-F1 | UnkLbl% | Set-Jacc |
| Gemini-2.5-Pro | 0.758 | 0.784 | 0.741 | 0.967 | 0.00 | 1.000 |
| Qwen3-8B | 0.538 | 0.531 | 0.306 | 0.536 | 0.00 | 1.000 |
| Llama3-8B | 0.496 | 0.332 | 0.264 | 0.536 | 0.00 | 1.000 |
| Mistral-7B | 0.492 | 0.330 | 0.260 | 0.536 | 0.00 | 1.000 |
| DeepSeek-Llama-8B | 0.500 | 0.333 | 0.268 | 0.536 | 0.00 | 0.500 |
| GPT-4.1-mini | 0.452 | 0.311 | 0.148 | 0.392 | 0.00 | 1.000 |
| Gemma-1B | 0.196 | 0.282 | 0.000 | 0.331 | 0.00 | 1.000 |
| Gemma-4B | 0.020 | 0.038 | 0.000 | 0.039 | 0.00 | 0.500 |
| Llama3-3B | 0.002 | 0.004 | 0.000 | 0.008 | 0.00 | 1.000 |
| Pima Indians Diabetes — Few-shot | | | | | | |
| Model | Acc | Macro-F1 | PenAcc | Len-F1 | UnkLbl% | Set-Jacc |
| Gemini-2.5-Pro | 0.820 | 0.814 | 0.820 | 1.000 | 0.00 | 1.000 |
| Qwen3-8B | 0.580 | 0.408 | 0.577 | 0.995 | 0.00 | 1.000 |
| DeepSeek-Llama-8B | 0.570 | 0.363 | 0.138 | 0.137 | 0.00 | 0.500 |
| Mistral-7B | 0.510 | 0.508 | 0.078 | 0.137 | 0.00 | 1.000 |
| Gemma-4B | 0.420 | 0.296 | 0.000 | 0.137 | 0.00 | 1.000 |
| Llama3-8B | 0.430 | 0.301 | 0.000 | 0.137 | 0.00 | 0.500 |
| GPT-4.1-mini | 0.390 | 0.281 | 0.390 | 1.000 | 0.00 | 1.000 |
| Gemma-1B | 0.370 | 0.272 | 0.367 | 0.995 | 0.00 | 1.000 |
| Llama3-3B | 0.000 | 0.000 | 0.000 | 0.000 | 0.00 | 0.000 |

Table 26: Pima Indians Diabetes: STaDS metrics per model for zero-shot and few-shot. Acc = accuracy; PenAcc = penalised accuracy; Len-F1 = length F1; UnkLbl% = unknown label rate (%).

| Dataset | Zero-shot (by PA) | | Best Z | Few-shot (by PA) | | Best F |
| | P-Acc | (Acc) | Model | P-Acc | (Acc) | Model |
|---|---|---|---|---|---|---|
| Adult Income | 0.700 | 0.700 | Gemini-2.5-Pro | 0.737 | 0.737 | Gemini-2.5-Pro |
| Breast Cancer | 0.646 | 0.650 | Gemini-2.5-Pro | 0.732 | 0.732 | GPT-4.1-mini |
| Car Evaluation | 0.406 | 0.419 | Gemini-2.5-Pro | 0.586 | 0.600 | Gemini-2.5-Pro |
| COMPAS | 0.571 | 0.816 | Gemini-2.5-Pro | 0.697 | 0.716 | Gemini-2.5-Pro |
| Congression Voting | 0.388 | 0.397 | Gemini-2.5-Pro | 0.638 | 0.638 | Gemini-2.5-Pro |
| Gaussian Synthetic | 0.550 | 0.550 | Gemini-2.5-Pro | 0.880 | 0.880 | GPT-4.1-mini |
| German Credit | 0.118 | 0.550 | Mistral-7B-Instruct-v0.3 | 0.889 | 0.889 | Gemini-2.5-Pro |
| Give Me Some Credit | 0.825 | 0.830 | DeepSeek-Llama-8B | 0.917 | 0.917 | GPT-4.1-mini |
| Heart Disease | 0.640 | 0.640 | GPT-4.1-mini | 0.700 | 0.700 | GPT-4.1-mini |
| HELOC | 0.670 | 0.670 | Gemini-2.5-Pro | 0.885 | 0.885 | GPT-4.1-mini |
| Iris | 0.785 | 0.787 | Gemini-2.5-Pro | 1.000 | 1.000 | Gemini-2.5-Pro |
| Monk1 | 0.618 | 0.620 | Gemini-2.5-Pro | 0.726 | 0.759 | Qwen3-8B |
| Monk2 | 0.461 | 0.484 | Gemini-2.5-Pro | 0.634 | 0.667 | Gemma-1B |
| Monk3 | 0.563 | 0.579 | Gemini-2.5-Pro | 0.644 | 0.644 | Gemini-2.5-Pro |
| Pima Diabetes | 0.741 | 0.758 | Gemini-2.5-Pro | 0.820 | 0.820 | Gemini-2.5-Pro |

Table 27: Penalised accuracy (P-Acc) summary. Higher is better; penalisation reduces scores for overlong outputs and invalid labels.($\alpha$=0.5, $\beta$=0.5).

| Dataset | Mean Cramér's V | Mean NMI | Mean Pearson $r$ | Mean Spearman $\rho$ | Top-3 by NMI |
|---|---|---|---|---|---|
| Adult Income | 0.308 | 0.053 | 0.143 | 0.164 | relationship, marital-status, capital-gain |
| Breast Cancer | 0.162 | 0.017 | – | – | inv-nodes, deg-malig, irradiat |
| Car Evaluation | 0.196 | 0.072 | – | – | persons, safety, buying |
| COMPAS | 0.203 | 0.027 | 0.071 | 0.086 | decile_score, score_text, priors_count |
| Congressional Voting | 0.503 | 0.165 | – | – | physician-fee-freeze, el-salvador-aid, education-spending |
| Gaussian Synthetic | – | 0.019 | −0.019 | −0.022 | gauss_1, gauss_2, gauss_6 |
| German Credit | 0.025 | 0.006 | −0.025 | −0.023 | other-installment-plans, installment-rate, number-credits |
| Give Me Some Credit | 0.205 | 0.014 | −0.031 | 0.017 | RevolvingUtilizationOfUnsecuredLines, NumberOfTimes90DaysLate, NumberOfTime30-59DaysPastDueNotWorse |
| Heart Disease | 0.083 | 0.008 | 0.122 | 0.104 | age, prevalentHyp, sysBP |
| HELOC | 0.189 | 0.030 | 0.035 | 0.036 | ExternalRiskEstimate, NetFractionRevolvingBurden, PercentTradesWBalance |
| Iris | 0.633 | 0.677 | 0.866 | 0.867 | petal_length, petal_width, sepal_length |
| Monk1 | 0.095 | 0.041 | – | – | a5, a2, a1 |
| Monk2 | 0.021 | 0.012 | – | – | a4, a1, a6 |
| Monk3 | 0.213 | 0.075 | – | – | a5, a2, a6 |
| Pima Diabetes | 0.252 | 0.047 | 0.206 | 0.224 | Glucose, BMI, Age |

Table 28: Average of feature–target statistical dependency metrics across all benchmark datasets. Values are averaged over all features within each dataset. Dashes indicate non-applicable metrics (e.g., Pearson/Spearman for categorical targets). Top-ranked features by NMI highlight dominant statistical dependencies, which serve as proxies for co-occurrence rather than causal relationships.

# C    Additional Analyses for Reviewer Concerns

This appendix reports additional analyses that further test the robustness and scope of STaDS. Specifically, we examine sensitivity to alternative perturbation operators, robustness of the penalized-accuracy metric, the effect of output post-processing, behavior on a domain-specialized medical model, and higher-order feature reliance through correlated group ablations. Together, these analyses clarify the methodological choices in the main paper and probe several plausible alternative explanations for the observed accuracy–faithfulness gap.

## C.1    Alternative perturbation operators versus deletion-based LAO

**Reviewer concern.**    Reviewers questioned whether complete feature removal is the most appropriate intervention, and further asked whether the results are sensitive to alternative perturbation mechanisms, including replacement-based baselines and positional perturbations such as row-order randomization and feature-order shuffling.

**Setup.**    We compared deletion-based LAO against four alternative perturbation operators on two representative datasets, *Breast Cancer* and Iris, using Qwen3-8B and GPT-4.1-mini. In all cases, we perturbed one feature at a time across the full table and then re-ran the same prompting and evaluation pipeline as in the main experiments. The goal was to test whether the inferred feature-reliance ranking is stable under different intervention semantics.

Concretely, we considered the following operators. **(1) Drop / deletion (LAO):** the target column is removed entirely from the table, yielding a missing-field counterfactual in which the attribute is no longer available to the model. **(2) Constant replacement:** the target column is replaced by a fixed missing-value placeholder for all rows, so that the column schema remains visible but its content is uniformly uninformative. **(3) Mean replacement:** for numeric features, every entry in the target column is replaced by the dataset mean of that column, producing a deterministic low-variance baseline that preserves scale while suppressing instance-specific information. **(4) Marginal sampling:** the target feature is resampled from its empirical marginal distribution and rewritten across rows, following the same intuition as SHAP-style marginal perturbation, namely to break the original instance-specific association while keeping sampled values in-distribution at the univariate level. **(5) Permutation:** the target column is permuted across rows, so that the original set of observed values is preserved exactly but reassigned to different instances; this destroys the row-wise alignment between that feature and the remaining columns while leaving the marginal distribution unchanged.

For each operator, we computed the change in predictive performance after perturbation and induced a feature ranking from the resulting degradation profile. We then compared each operator-specific ranking with the deletion-based LAO ranking using Spearman correlation. In addition, we report the proportion of features for which perturbation yields $\Delta > 0$, i.e., cases where the perturbed table leads to *better* predictive performance than the unperturbed table. This additional statistic is useful because, if an operator cleanly captures behavioral reliance, one would not expect perturbing an important feature to systematically improve model performance.

**Results and discussion.**    Table 30 shows that the agreement between alternative perturbation operators and deletion-based LAO varies substantially across datasets and models. Several settings exhibit weak or even negative agreement, including Breast–Qwen3-8B under constant replacement ($\rho = +0.53$), mean replacement ($\rho = -0.29$), and marginal sampling ($\rho = -0.69$), and Iris–GPT-4.1-mini under permutation ($\rho = -0.40$) and marginal sampling ($\rho = -0.80$). These sign reversals indicate that the induced importance ordering can change materially depending on the perturbation mechanism.

Table 31 further shows that alternative operators frequently produce $\Delta > 0$ cases, that is, perturbations that *increase* predictive performance. This occurs at non-trivial rates across most settings, including 66.7%–88.9% of features for Breast–Qwen3-8B under replacement/permutation operators, 75.0% for multiple operators on Iris–Qwen3-8B, and 100% for mean replacement on Iris–GPT-4.1-mini. Such behavior is difficult to reconcile with a pure feature-reliance interpretation, because replacing or permuting a supposedly important feature should not systematically improve performance if the operator were measuring reliance cleanly. Figure 16 visualizes this instability at the feature level. The heatmaps show that both the magnitude and the sign of the feature-level performance change depend strongly on the perturbation operator, while the rank-bump plots show that features can move substantially in the induced importance ordering across operators.

These results support our use of deletion-based LAO as the primary behavioral reliance measure in STaDS. Our claim is not that deletion is universally superior to all other perturbation operators, but rather that it is the most transparent intervention for the decision-simulation setting studied here: the attribute is simply absent from the table presented to the model. By contrast, replacement and imputation operators introduce additional assumptions about plausible substitutes, marginal distributions, or correlation structure, and in our experiments they can substantially alter the

Table 30: Rank agreement between alternative perturbation operators and deletion-based LAO. We report Spearman correlation $\rho$ between the feature ranking induced by each operator and the deletion-based LAO ranking. Large variation, including negative values, indicates that the inferred reliance ordering can be strongly operator-dependent.

| Dataset | Model | Constant | Mean | Permutation | Marginal |
|---------|-------|----------|------|-------------|----------|
| Breast | Qwen3-8B | +0.53 | -0.29 | +0.22 | -0.69 |
| Breast | GPT-4.1-mini | +0.05 | +0.17 | +0.41 | +0.42 |
| Iris | Qwen3-8B | -0.40 | +0.20 | +0.00 | +0.60 |
| Iris | GPT-4.1-mini | +0.95 | +0.45 | -0.40 | -0.80 |

Table 31: Fraction of features with positive perturbation effect ($\Delta > 0$), i.e., cases where perturbing a feature *improves* predictive performance. For replacement/permutation/marginal operators, $\Delta = \text{Acc}_{\text{perturbed}} - \text{Acc}_{\text{base}}$. For deletion-based LAO, we report the analogous quantity under feature removal. A high prevalence of $\Delta > 0$ suggests that the operator may introduce artifacts rather than isolate clean feature reliance.

| Dataset | Model | Drop (LAO) | Constant | Mean | Permutation | Marginal |
|---------|-------|------------|----------|------|-------------|----------|
| Breast | Qwen3-8B | 55.6% | 66.7% | 88.9% | 66.7% | 55.6% |
| Breast | GPT-4.1-mini | 44.4% | 11.1% | 33.3% | 44.4% | 55.6% |
| Iris | Qwen3-8B | 25.0% | 75.0% | 75.0% | 50.0% | 75.0% |
| Iris | GPT-4.1-mini | 50.0% | 0.0% | 100.0% | 25.0% | 50.0% |

induced ranking and sometimes even improve predictive performance. We therefore interpret these comparisons as evidence that perturbation choice is itself a major source of attribution variability, while deletion provides the clearest missing-field counterfactual for our setting.

## C.2 Sensitivity of Penalized Accuracy to penalty weights

**Reviewer concern.** Reviewers asked whether the PenAcc weights are arbitrary and whether the conclusions depend heavily on the choice of penalty trade-off between output-length violations and unknown-label violations.

**Setup.** Recall that Penalized Accuracy is defined as

$$\text{PenAcc} = \text{Acc} - \big(\alpha(1 - \text{Len-F1}) + \beta\,\text{UnkLbl}\%\big), \tag{4}$$

where $\alpha, \beta > 0$ are penalty weights. To test robustness, we varied $(\alpha, \beta)$ along the simplex $\alpha + \beta = 1$ and recomputed both the base PenAcc and the LAO-induced degradation under each setting. We report this sensitivity analysis on Iris and Breast Cancer as representative case studies.

**Results and discussion.** Figure 18 shows that the mean degradation induced by feature removal changes smoothly as the weighting shifts from length-fidelity violations to unknown-label violations. We do not observe abrupt reversals or qualitatively unstable behavior across the explored range.

These results indicate that the main LAO-based conclusions are not an artifact of a single hand-picked penalty configuration. The precise PenAcc value naturally shifts with the intended emphasis on different classes of format failure, but the degradation trends remain stable. This is consistent with the role of PenAcc as an end-to-end task-quality measure that jointly penalizes wrong labels, missing labels, and malformed outputs, rather than as a brittle metric tuned to a specific domain. In the main paper we use equal weighting as a neutral default, not as a theoretically privileged choice.

## C.3 Runtime and computational cost

**Reviewer concern.** Reviewers requested approximate evaluation cost to assess the reproducibility and practical overhead of the proposed protocol.

**Results.** Table 32 reports wall-clock runtime for one single-feature LAO pass per feature. Runtime is dominated by model inference latency. GPT-4.1-mini is relatively inexpensive, requiring 3.78s per feature on Iris and 9.97s

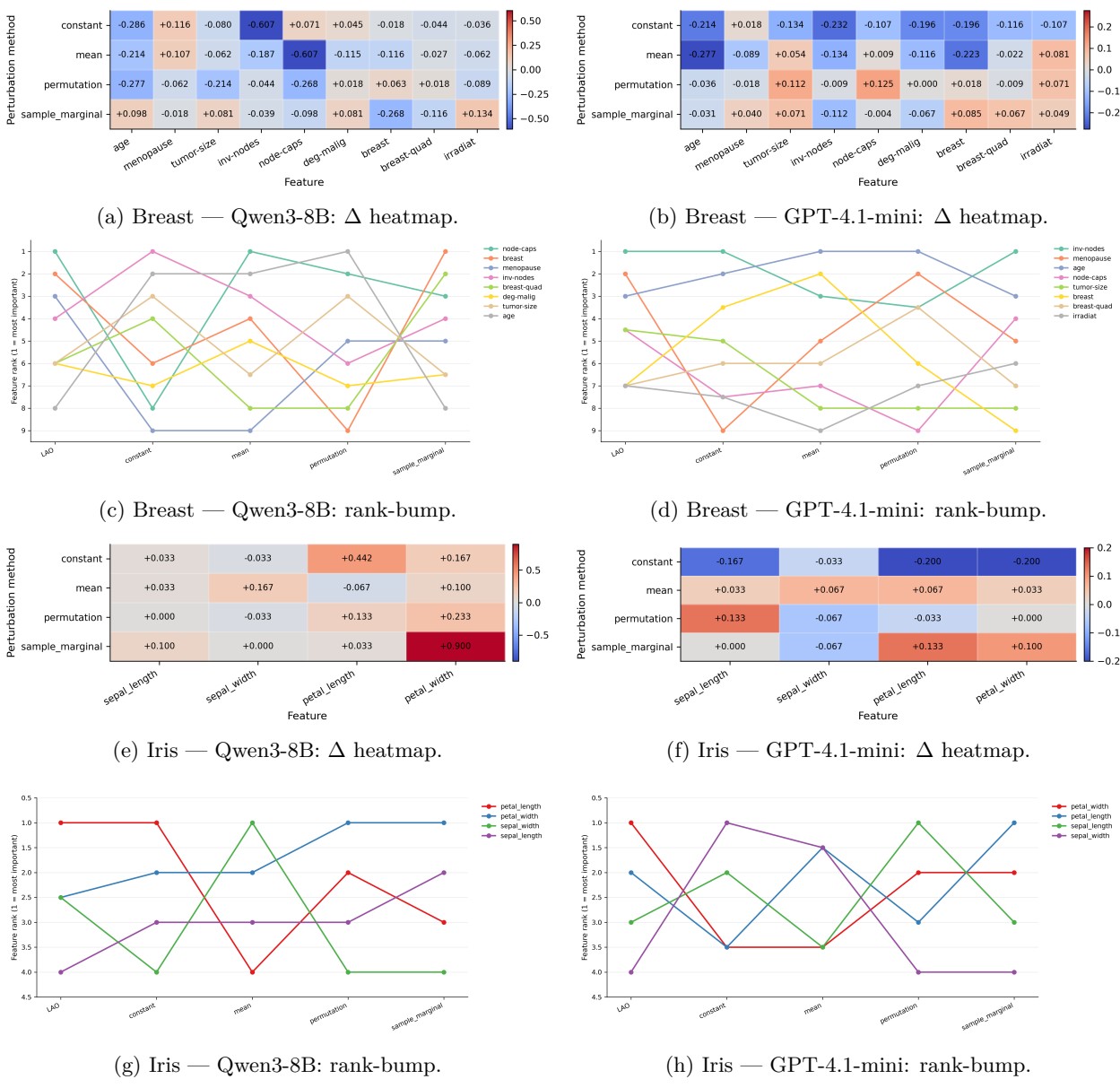

Figure 16: Perturbation-operator sensitivity of feature reliance estimates. Top: Breast Cancer; bottom: Iris. Heatmaps show the feature-level performance change Δ under each perturbation operator. Rank-bump plots show how the induced feature-importance ordering shifts across operators. The substantial movement in both sign and rank demonstrates that alternative perturbation mechanisms can produce materially different conclusions about which features a model relies on.

on Breast Cancer. Qwen3-8B is slower, requiring 102.45s per feature on Iris and 257.23s on Breast Cancer. The domain-specialized MedGemma-4B is slower still, requiring 461.28s per feature on Breast Cancer and 884.15s on Pima.

Table 33 reports runtime for correlated group ablations. Because each feature group is removed jointly in a single evaluation pass, group ablations are broadly comparable to a single-feature run rather than additive in the number of features in the group. The cost profile is practical for API models and moderate-scale local models, but more expensive for domain-specialized local models such as MedGemma-4B. These measurements provide a practical estimate of the computational cost of extending STaDS to additional datasets, models, or ablation settings.

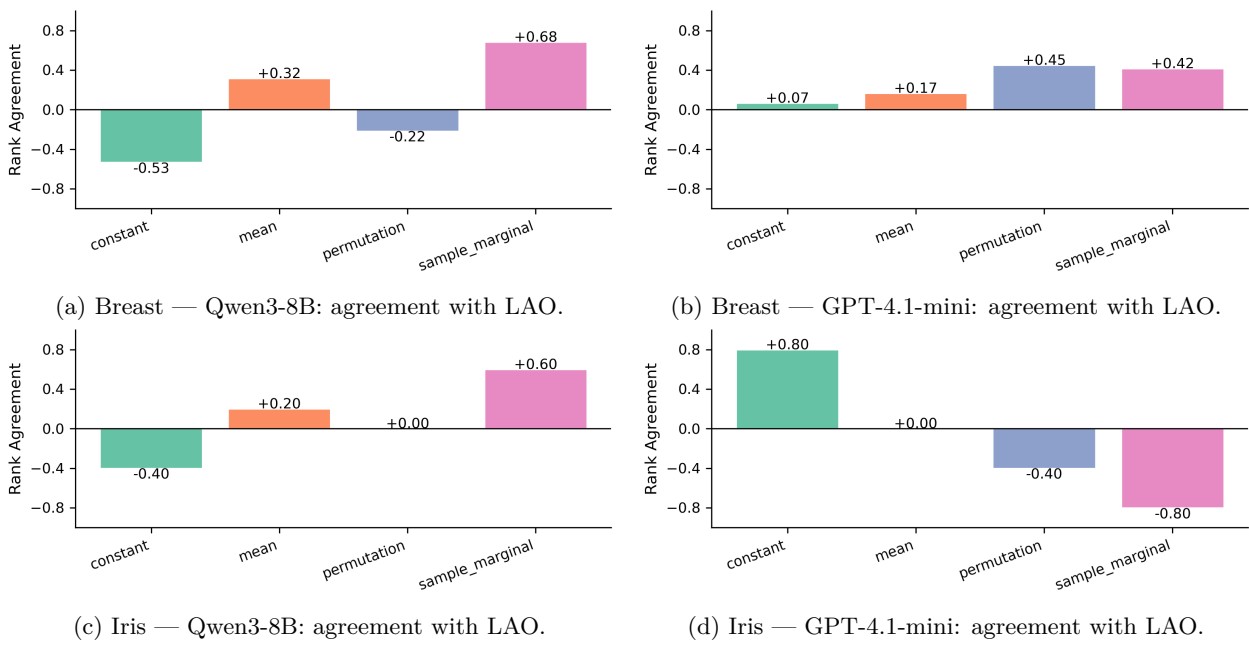

(a) Breast — Qwen3-8B: agreement with LAO.

(b) Breast — GPT-4.1-mini: agreement with LAO.

(c) Iris — Qwen3-8B: agreement with LAO.

(d) Iris — GPT-4.1-mini: agreement with LAO.

Figure 17: Rank agreement between each alternative perturbation operator and deletion-based LAO. Bars report Spearman correlation between the operator-induced ranking and the LAO ranking. The broad spread of values, including negative correlations, reinforces that estimated feature importance is highly operator-dependent.

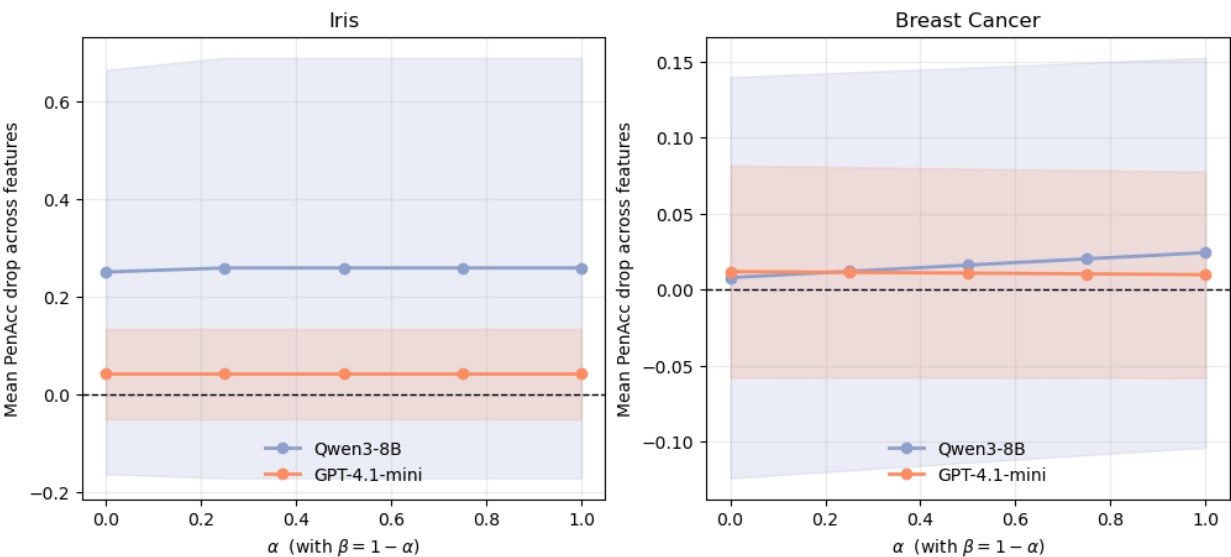

Figure 18: Sensitivity of the mean LAO-induced PenAcc drop to the penalty trade-off $\alpha$ (with $\beta = 1 - \alpha$). Across both datasets, the average degradation varies smoothly rather than erratically, indicating that the qualitative conclusions of the LAO analysis are stable to the choice of PenAcc penalty weights.

### C.4 Audit of output post-processing

**Reviewer concern.** Reviewers asked whether the GPT-4.1-mini post-processing step used to clean raw outputs could alter predictions, mask instruction-following failures, or otherwise confound the reported results.

Table 32: Runtime for single-feature LAO evaluation. Reported times correspond to one ablation pass per feature, aggregated across all features within the dataset. Mean and median are computed across runs to reflect typical per-feature wall-clock cost.

| Dataset | Model | # Runs | Mean (s) | Median (s) |
|---|---|---|---|---|
| breast | GPT-4.1-mini | 10 | 9.97 | 9.87 |
| iris | GPT-4.1-mini | 5 | 3.78 | 4.27 |
| breast | Qwen3-8B | 10 | 257.23 | 263.57 |
| iris | Qwen3-8B | 5 | 102.45 | 75.22 |
| breast | MedGemma-4B | 10 | 461.28 | 458.60 |
| pima | MedGemma-4B | 9 | 884.15 | 883.32 |

Table 33: Runtime for correlated group ablations. Each group ablation requires a single evaluation pass for the jointly removed feature set. We report these values as illustrative, as each group configuration was executed once.

| Dataset | Model | Group size | Total / run (s) |
|---|---|---|---|
| breast | GPT-4.1-mini | 3 cols | 6.62 |
| breast | GPT-4.1-mini | 2 cols | 6.12 |
| breast | Qwen3-8B | 3 cols | 184.98 |
| breast | Qwen3-8B | 2 cols | 242.33 |
| iris | GPT-4.1-mini | 3 cols | 4.34 |
| iris | GPT-4.1-mini | 2 cols | 12.43 |
| iris | Qwen3-8B | 3 cols | 136.94 |
| iris | Qwen3-8B | 2 cols | 121.78 |

**Setup.** We conducted a post-auditing analysis that re-checked accuracy after output cleaning under three settings: no ablation, single-column ablation, and multi-column ablation.

**Results.** Table 34 shows that post-auditing preserves accuracy in essentially all evaluated settings. All base runs remain at 1.00 post-audit accuracy. For single-column ablations, all settings remain at 1.00 except Breast–Qwen3-8B, which drops slightly to 0.94. All evaluated multi-column settings remain at 1.00.

These results suggest that the post-processing step does not materially alter the substantive conclusions of the evaluation. In nearly all cases it acts as a formatting normalizer rather than a corrective predictor. We therefore do not find evidence that the reported STaDS conclusions are driven by post-processing artifacts, although local formatting cleanup remains useful for robust metric extraction. We further examined the small accuracy drop for Breast–Qwen3-8B under single-column ablation. In these cases, the main issue appears to be inconsistency between the generated reasoning trace and the final predicted label, rather than systematic correction by the post-processing model.

### C.5 Domain-specialized medical model: MedGemma

**Reviewer concern.** Reviewers asked whether the observed gap between predictive success and faithfulness persists for domain-specialized expert models, especially in high-stakes settings such as medicine.

**Setup.** To address this point, we evaluated `google/medgemma-4b` on two healthcare datasets, Breast Cancer and Pima Diabetes, and compared it against the general-domain Gemma3-4B baseline to distinguish the effect of domain specialization from backbone-family effects. For each model, we measured behavioral reliance via LAO and compared it against the model's self-reported feature ranking.

**Results.** Figure 20 shows that domain specialization does not automatically resolve the accuracy–faithfulness gap. However, on *Pima Diabetes*, Gemma3-4B exhibits a nearly flat LAO profile despite a non-flat self-reported ranking, suggesting that the model can articulate a preference ordering without showing corresponding behavioral dependence. This illustrates a particularly clear mismatch case for Gemma3-4B under our protocol, where the model produces a non-flat self-reported ranking, yet its measured LAO profile on Pima is nearly flat. MedGemma's self-reported

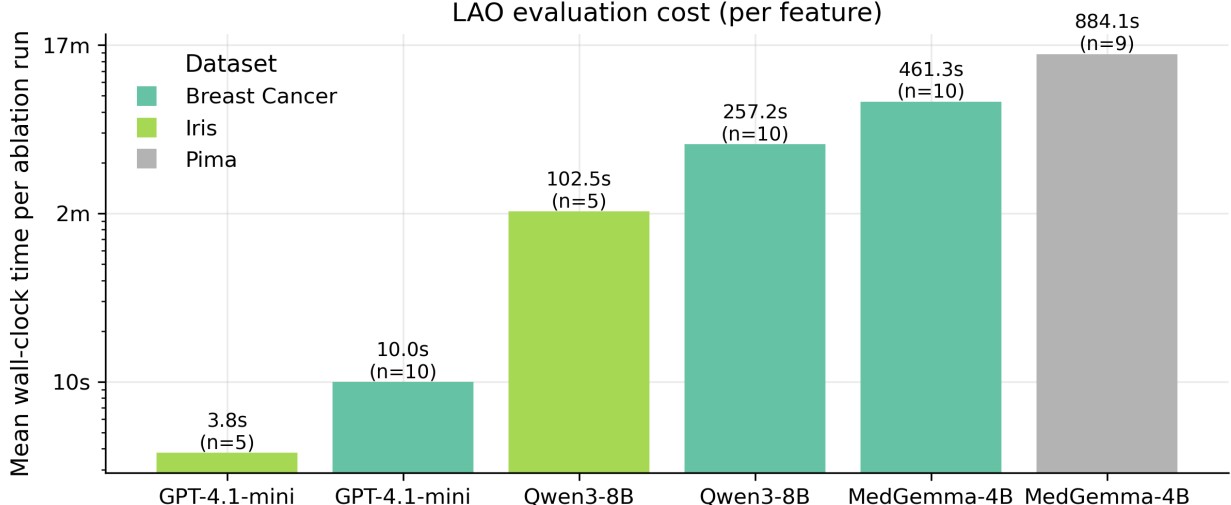

Figure 19: Per-feature evaluation cost of LAO across datasets and models. GPT-4.1-mini is the most efficient among the tested models, Qwen3-8B is substantially slower, and MedGemma-4B incurs the largest wall-clock cost. These results quantify the practical overhead of feature-level STaDS faithfulness evaluation.

Table 34: Post-auditing accuracy after output cleaning. We report the average accuracy after applying the output-cleaning step under no-ablation, single-feature ablation, and multi-column ablation settings. The near-perfect preservation of accuracy indicates that post-processing acts primarily as formatting normalization rather than substantive prediction correction.

| Dataset | Model | Ablation setting | Post-audit Acc (avg) |
|---------|-------|------------------|----------------------|
| breast | GPT-4.1-mini | 0 col | 1.00 |
| breast | Qwen3-8B | 0 col | 1.00 |
| iris | GPT-4.1-mini | 0 col | 1.00 |
| iris | Qwen3-8B | 0 col | 1.00 |
| breast | GPT-4.1-mini | 1 col | 1.00 |
| breast | Qwen3-8B | 1 col | 0.94 |
| iris | GPT-4.1-mini | 1 col | 1.00 |
| iris | Qwen3-8B | 1 col | 1.00 |
| pima | MedGemma-4B | 1 col | 1.00 |
| iris | GPT-4.1-mini | multi-cols | 1.00 |
| breast | Qwen3-8B | multi-cols | 1.00 |
| breast | GPT-4.1-mini | multi-cols | 1.00 |
| iris | Qwen3-8B | multi-cols | 1.00 |

rationales also appear more clinically plausible in qualitative terms, but they are still not strongly aligned with the measured LAO ranking. On *Breast Cancer*, both models again exhibit mismatches between self-attribution and behavioral reliance. MedGemma's self-reported rationales are more clinically plausible, but they are still not strongly aligned with the measured LAO ranking. Tables 35, 36, and 37 summarize these mismatches.

The main conclusion is that domain specialization appears to improve plausibility more readily than self-faithfulness in this setting. That is, expert-domain models may produce more clinically grounded explanations, while their stated decision factors can still diverge from the features that measurably affect their predictions under intervention.

## C.6 Correlated group ablations and higher-order reliance

**Reviewer concern.** Reviewers noted that single-feature LAO may underestimate reliance when models exploit correlated predictors jointly, and asked whether group-based ablations reveal meaningful interaction effects.

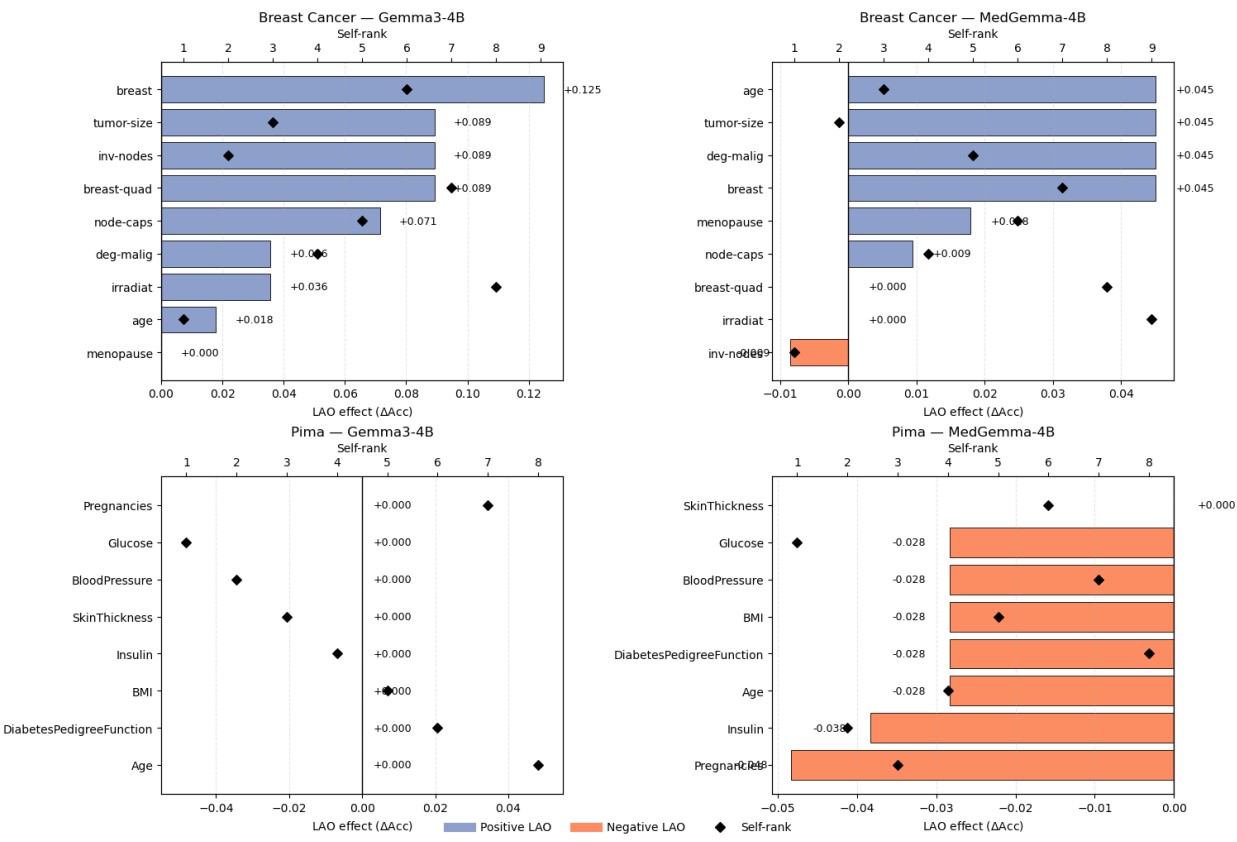

Figure 20: Direct comparison between general-domain Gemma3-4B and domain-specialized MedGemma-4B on *Breast Cancer* and *Pima Diabetes*. Bars show the LAO effect of removing each feature, while black diamonds show the model's self-reported importance rank. On Pima, Gemma displays a flat behavioral profile despite a non-flat self-ranking, whereas MedGemma exhibits stronger behavioral sensitivity but still poor alignment with self-attribution. On Breast Cancer, both models show mismatches between stated and behavioral importance, although MedGemma's explanations are more clinically plausible.

**Setup.** We conducted group ablations on correlated feature subsets for two datasets. For Iris, we considered the pair `petal_length`–`petal_width` and the three-feature set `petal_length`–`petal_width`–`sepal_length`. For *Breast Cancer*, we considered the correlated set `inv-nodes`–`deg-malig`–`irradiat`. These are most statistically correlated features from Table 28.

We define the standardized performance drop after removing a feature set $S$ as

$$\Delta(S) = \mathrm{Acc}_{\mathrm{base}} - \mathrm{Acc}_{\setminus S}, \tag{5}$$

where larger positive values indicate stronger reliance on the removed feature set. To isolate non-additive group effects, we compute

$$I(S) = \Delta(S) - \sum_{j \in S} \Delta(\{j\}), \tag{6}$$

where $I(S) > 0$ indicates super-additive interaction, $I(S) \approx 0$ indicates near-additivity, and $I(S) < 0$ indicates redundancy or overlap.

**Results.** Table 38 and Figures 21–22 show that meaningful higher-order reliance exists in several settings, but is strongly model-dependent.

On Iris, Qwen3-8B shows clear super-additive behavior. Removing `petal_length` and `petal_width` individually yields moderate drops (0.133 and 0.100), but removing the pair yields a much larger drop (0.433), giving a positive pair interaction of 0.200. Removing the three-feature group yields an even larger drop (0.567), with a triple interaction

Table 35: Case-study summary of MedGemma on two healthcare datasets. We compare self-reported feature importance with behavioral reliance measured by LAO. Positive $\Delta$Acc indicates that removing the feature harms performance, whereas negative values indicate that removal improves performance.

| Dataset | Top self-attributed features | Strongest LAO features | Self–LAO agreement | Interpretation |
|---|---|---|---|---|
| Breast Cancer | inv-nodes, tumor-size, age, node-caps, deg-malig | age, tumor-size, deg-malig, breast | weak positive ($\rho \approx 0.14$) | MedGemma identifies several clinically plausible variables, but the self-reported ranking only weakly matches behavioral reliance. |
| Pima Diabetes | Glucose, Insulin, Pregnancies, Age, BMI | SkinThickness (near zero), with several negative effects for self-reported features | moderate negative ($\rho \approx -0.49$) | Stated rationale diverges substantially from behavioral reliance; removing several self-reported important features improves performance. |

Table 36: Breast Cancer: MedGemma LAO ranks versus self-attribution ranks. Lower LAO rank indicates stronger measured behavioral reliance.

| Feature | LAO rank | Self rank |
|---|---|---|
| breast | 1 | 6 |
| inv-nodes | 2 | 2 |
| tumor-size | 3 | 3 |
| breast-quad | 4 | 7 |
| node-caps | 5 | 5 |
| deg-malig | 6 | 4 |
| irradiat | 7 | 8 |
| age | 8 | 1 |
| menopause | 9 | – |

of 0.333. This suggests that the model relies jointly on the correlated petal features in a way that is not captured by single-feature ablations alone. In contrast, GPT-4.1-mini on Iris exhibits the opposite pattern. Although the individual removals produce drops of 0.033 and 0.167, removing the pair yields only 0.037, corresponding to a negative interaction of $-0.163$. The three-feature group likewise yields a near-zero drop (0.003), indicating redundancy rather than synergy.

On Breast Cancer, the pattern is again model-dependent. Qwen3-8B shows a modest positive pair interaction, while the triple interaction is also positive, although its interpretation is complicated by heterogeneous single-feature effects, including a strongly negative drop for one feature. GPT-4.1-mini exhibits clearer group effects: the pair removal yields a drop of 0.205 and the triple removal yields a drop of 0.256, corresponding to positive non-additive effects of 0.129 and 0.180, respectively. These results support the reviewer's concern that single-feature ablations can miss higher-order reliance, but they also show that interaction effects are not universal. Group ablations should therefore be viewed as a complementary stress test for joint dependence rather than as a replacement for single-feature LAO. In our experiments, higher-order effects are present in some model–dataset settings and absent or even negative in others, which itself is an informative property of the decision strategy being probed. More broadly, the same framework can be extended to probe higher-order interactions beyond pairs and triples, although the computational cost grows quickly with interaction order.

Table 37: Pima Diabetes: MedGemma LAO ranks versus self-attribution ranks. LAO deltas are near-zero within rounding, indicating weak measured counterfactual dependence under this protocol.

| Feature | LAO rank | Self rank |
|---|---|---|
| BloodPressure | 1 | 2 |
| SkinThickness | 2 | 3 |
| Pregnancies | 3 | 7 |
| Glucose | 4 | 1 |
| Age | 5 | 8 |
| DiabetesPedigreeFunction | 6 | 6 |
| BMI | 7 | 5 |
| Insulin | 8 | 4 |

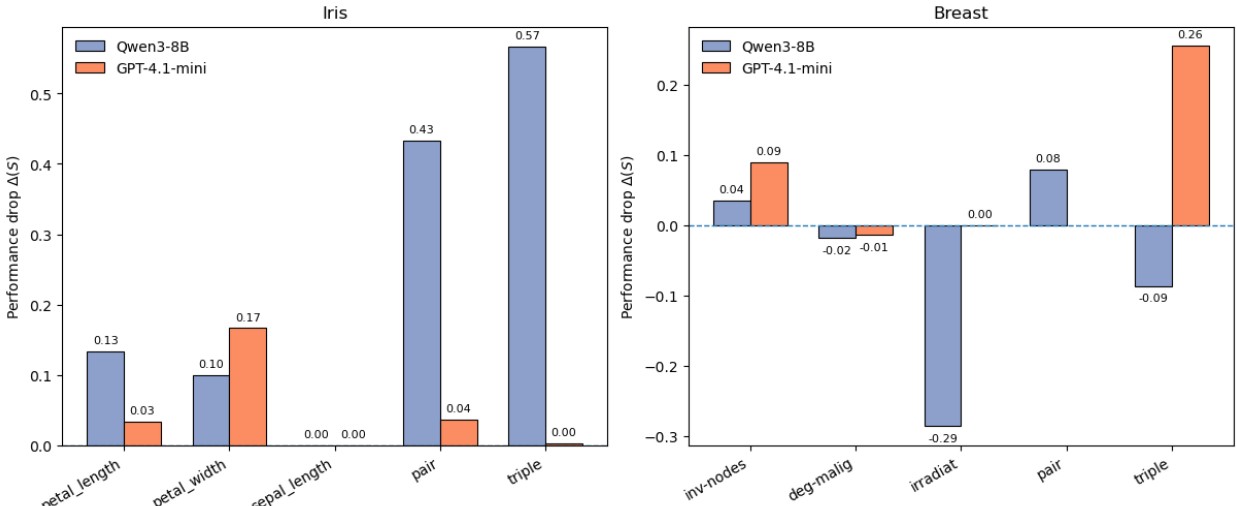

Figure 21: Standardized performance drops under single-feature and correlated group ablations. Larger values indicate stronger reliance on the removed feature set. Qwen3-8B on Iris shows a pronounced increase from single-feature to pair and triple ablations, consistent with joint reliance on correlated petal features, whereas GPT-4.1-mini on Iris exhibits much weaker group effects.

Table 38: Group ablations on correlated feature subsets. $\Delta(f)$ denotes the standardized performance drop after removing feature $f$, while $\Delta(f_1, f_2)$ and $\Delta(f_1, f_2, f_3)$ denote the drops under pair and triple removal. $I(\cdot)$ is the corresponding non-additive interaction effect relative to the sum of single-feature drops.

| Dataset | Model | $\Delta(f_1)$ | $\Delta(f_2)$ | $\Delta(f_3)$ | $\Delta(f_1, f_2)$ | $\Delta(f_1, f_2, f_3)$ | $I(f_1, f_2)$ / $I(f_1, f_2, f_3)$ |
|---|---|---|---|---|---|---|---|
| Iris | Qwen3-8B | 0.133 | 0.100 | 0.000 | 0.433 | 0.567 | 0.200 / 0.333 |
| Iris | GPT-4.1-mini | 0.033 | 0.167 | 0.000 | 0.037 | 0.003 | -0.163 / -0.197 |
| Breast | Qwen3-8B | 0.036 | -0.018 | -0.286 | 0.080 | -0.086 | 0.062 / 0.182 |
| Breast | GPT-4.1-mini | 0.090 | -0.013 | 0.000 | 0.205 | 0.256 | 0.129 / 0.180 |

For Iris, $f_1 = $ `petal_length`, $f_2 = $ `petal_width`, and $f_3 = $ `sepal_length`. For *Breast Cancer*, $f_1 = $ `inv-nodes`, $f_2 = $ `deg-malig`, and $f_3 = $ `irradiat`.

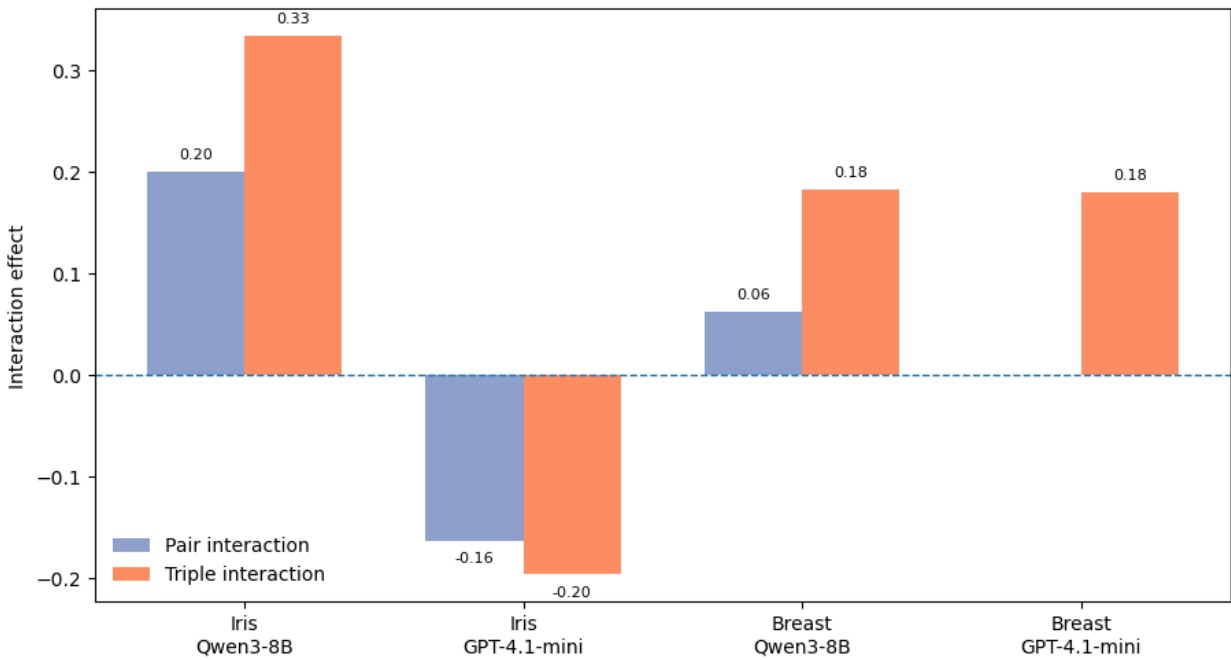

Figure 22: Non-additive interaction effects for correlated feature groups, computed as the observed group-ablation drop minus the sum of the corresponding single-feature drops. Positive values indicate super-additive reliance, whereas negative values indicate redundancy or overlap. The direction of the interaction differs across models, showing that higher-order feature reliance is itself model-dependent.

