# OpenReview forum: "Evaluating LLM Understanding via Structured Tabular Decision Simulations"
_TMLR — Rejected by TMLR_

### Review · Reviewer_2Zt2 · 2026-01-12

**Summary Of Contributions:**

This work proposes a benchmark based on tabular data. Basic idea is to investigate the structural data understanding by large language models, in which each instance is organized as a set of key-value features with the label in question for prediction, a kind of question-answering in a more structured form. Experiments are carried out on diverse tabular data and language models, showing that models perform differently.

**Audience:**

Yes

**Audience Explanation:**

This work might have a potential to measure the capabilities of large language models, which have not been investigated in prior studies, e.g., multi-hop QAs or long-context understanding.

**Broader Impact Concerns:**

None.

**Claims And Evidence:**

No

**Claims Explanation:**

* It is not clear whether the benchmark is investigating a new capability of LLMs when compared with other benchmarks, e.g., multi-hop QA and/or long-context benchmarks. This paper has several discussion on the understanding capability, but it is not clear whether the capability could be investigated uniquely by this dataset or not. In addition, the definition of "complexity" is not articulated clearly, and thus, it is not clear whether the experimental results could be fairly interpreted.
* It is not clear whether the benchmark is questioning the memorization ability or reasoning ability. If this work is focusing on reasoning without using memorized knowledge, then, it is better to isolate the benchmark. For instance, it is possible to isolate the impact of demonstration by feeding tabular data from different domain as few-shots.
* Some task settings are not clear and it is not clear whether the evaluation is fairly conducted. For example, the number of instances is specified by $N$, but the number of queries in a tabular data seems to be specified by $n_g$, potentially indicating $n_g < N$. Also, the answers are investigated only for the top $n_p < n_g$ without considering the alignment of gold queries and answered ones. In addition, the experiments does not take into account the positional biases.

**Requested Changes:**

* Discussion on the unique capabilities measured by the benchmark: I'd expect the more discussion by comparing with other benchmark data, e.g., multi-hop QA, to highlight the unique capabilities measured by the benchmark. Otherwise, it sounds like a suite of combining prior tabular data as a single dataset. In addition, it is better to define the complexity, e.g., whether a model has to predict from other entries in the same table, and how complex it is.
* it is better to run an experiment or analysis to isolate the effect of memorization. For example, it would be possible to use few-shot examples from different domain to isolate the effect of knowledge in context, to measure the reasoning capability from the tabular data.
* Improve the clarity:
  - It is not clear why $n_g < N$. In this case, only part of the data will be queried, instead of all. If this is the case, it is better to document how to select the subset for queries. Also, it is better to discuss the reason for choosing only subset.
  - The motivation of taking only the top most responses is not clear given that an LLM might simply skip an intermediate entries to respond. It is better to, at least, discuss any potential issues caused by this evaluation method.
  - It is not clear why the key=value format was used for the entire experiment. At least, this work needs further discussion on the impact of the format, or use, e.g., JSON format of dictionary, for feature representation.
* Impact of positions. LLMs are easily biased by positions, and it is better to run additional experiments by randomizing positions.

---

### Review · Reviewer_TdgX · 2026-01-16

**Summary Of Contributions:**

This paper introduces Structured Tabular Decision Simulations (STaDS), a novel evaluation framework for measuring LLMs’ understanding ability rather than merely surface-level predictive accuracy. The paper makes three primary contributions:
- A principled evaluation protocol (STaDS) that reframes real-world tabular datasets as expert decision simulations, enabling controlled and interpretable assessment of LLM behavior.
- A multidimensional operationalization of understanding, decomposed into: (1) Question and instruction comprehension, (2) Knowledge-based prediction competence, and (3) Reliance on the correct decision factors (global decision faithfulness).
- A large-scale empirical study across 9 frontier LLMs and 15 real-world tabular decision settings (healthcare, finance, public policy), showing that: (1) Many models fail to generalize robustly across domains, (2) Models can be accurate yet globally unfaithful, (3) Explanations frequently diverge from the true behavioral drivers of predictions.

Key strengths:
- Clear formalization of “understanding” into measurable behavioral dimensions.
- Rigorous experimental design with large-scale evaluation.
- Novel global-level faithfulness analysis via LAO attribution.

Key Weaknesses:
- All evaluated models are general-purpose LLMs. In many real-world high-stakes applications (e.g., medicine, finance, law), practitioners often rely on domain-specialized or fine-tuned expert models. It remains unclear whether the observed gaps between accuracy and faithfulness persist for expert models trained with domain supervision.
- Model performance may be influenced by how tabular data was represented as text during pre-training.

**Audience:**

Yes

**Audience Explanation:**

The paper is highly relevant to multiple segments of the TMLR audience:

- Researchers working on LLM evaluation, interpretability, and reasoning.
- The XAI community studying faithfulness and attribution reliability.
- Practitioners deploying LLMs in high-stakes domains such as healthcare, finance, and policy.
- Researchers studying in-context learning and generalization.

**Broader Impact Concerns:**

The work focuses on evaluation methodologies and does not present immediate ethical risks. Conversely, its focus on decision faithfulness helps identify potential hidden risks in automated decision-making systems.

**Claims And Evidence:**

Yes

**Claims Explanation:**

The claims are supported by a comprehensive and carefully designed experimental evaluation. The authors evaluate 9 state-of-the-art LLMs across 15 real-world tabular datasets spanning healthcare, finance, policy, and synthetic domains, totaling approximately 160k decision instances.

**Requested Changes:**

- Reporting approximate evaluation cost per dataset/model would help readers assess reproducibility.
- Adding a few concrete failure cases (tables + predictions + attribution mismatch) would improve interpretability.
- Evaluation of domain-specialized expert models.
- Given the use of well-known public datasets (e.g., Iris, Adult Income), a brief discussion on whether model performance is impacted by potential data leakage during pre-training would be beneficial. (Optional)

---

### Review · Reviewer_9siP · 2026-02-13

**Summary Of Contributions:**

This paper introduces STaDS - a protocol that intents to moves beyond traditional performance-only benchmarking to jointly evaluate understanding and explainability in large language models. Most existing LLM benchmarks focus exclusively on accuracy, with only limited explainability work using templated questions. In contrast, STaDS aims to provide a more comprehensive framework.
In this work, the authors made a key contribution of defining "understanding" along three behavioral dimensions: question and instruction comprehension, knowledge-based prediction, and reliance on the right decision factors.
Critically, the paper focuses on global faithfulness using tabular datasets that allows objective review of feature importance and faithful decision-making.
The paper makes 3 key contributions; (a) the STaDS protocol, (b) a suite of metrics covering comprehension fidelity, predicting competence, and decision faithfulness, and (c) a benchmark suite of 15 curated real-world tabular datasets spanning healthcare, finance, and public policy
The authors compared 9 state-of-the-art LLMs and found that models can be accurate yet globally unfaithful - these models can frequently achieve high predictive accuracy while demonstrating substantial mismatches between their stated rationales and actual decision factors.

Some of the key strengths of the papers are as follows
- The paper is well motivated and of high importance - they provide a well articulated reasoning to focus on global faithfulness rather than instance-level reasoning to uncover the models' true understanding
- The comprehensive metric suite covering multiple dimensions of understanding and the extensive empirical evaluation across diverse domains and models is well received and when published could benefit future research
- Finally, the findings around faithfulness gaps in several SOTA models, when substantiated further, is a key contribution to the general discourse

However, there are some major aspects where the paper needs improvements
- The paper suffers from poor presentations - frequently notations and abbreviations have been introduced before they have been defined. Sections seem to be lack a coherent voice and therefore makes the review more difficult
- There are several questions around the LAO approach and parameter choices that needs to be validated before the claims can be substantiated
- Finally, the paper lacks baseline comparisons such as missing human expert baselines or comparison to established explainability methods

**Additional Comments:**

See above

**Audience:**

Yes

**Audience Explanation:**

There is a clear audience for the paper. It extends the past work on LLM reasoning benchmarking and introduces new findings that can spur LLM methodological updates

**Claims And Evidence:**

No

**Claims Explanation:**

One of the key claims -  "Current LLMs struggle with consistent accuracy across diverse domains Substantiation: " - is well substantiated via the extensive suite of experiments.

However, the second and more interesting claim - that models can be accurate yet globally unfaithful - needs further deliberation. At a high level, the empricial evidence presented in the paper such as in Figure 7 effectively illustrates four distinct behavioral regimes, from models that are both accurate and faithful (rare) to those that are neither (common), with concerning examples of high accuracy paired with low faithfulness. The finding that ρ(πself, πNMI) > ρ(πLAO, πNMI) across datasets is particularly striking, suggesting that models' self-attributions align more strongly with statistical correlations in the data than with their actual behavioral reliance on features is also interesting.But the Leave-Any-Out (LAO) approach deserves deeper consideration - by completely removing features rather than replacing them with baseline values, the authors fundamentally change the model's functional form. This represents a significant departure from standard explainability methods like SHAP, which use reference values to maintain model structure. The logical fallacy here could undermine the validity of the Decision Faithfulness metrics, which form a cornerstone of the paper's contributions.

The final claim around using tabular simulations as the evaluation substrate is generally reasonable, with Section 3.2 providing five distinct advantages. The instance-level structure does prevent exploitation of dataset-level artifacts, and the explicit feature naming enables clear global-level faithfulness assessment. However, this also pulls into view a narrow focus on classification tasks. The paper briefly mentions extending to regression as "future work" without adequately justifying how that is feasible within this framework. Further it is not evident if the benchmarking for a tabular dataset can also be then extrapolated to open-domain problems.

**Requested Changes:**

Please see below a list of questions and proposed alternations to the paper

- First, critically the presentation needs to be improved substantially. In its current state, this significantly impedes comprehension. Concepts and notation are introduced before being defined, creating unnecessary confusion. For example, the understanding definition references parameters introduced much later. Metrics like UnkLbl% are mentioned well before Equation 2/3 where they're formally defined. The parameter $\gamma$ appears in understanding definitions before introduction
- Next, the authors should further deliberate on the feature ablation methodology. The complete removal of features changes the model's functional form, which differs fundamentally from standard explainability approaches.Some questions to consider
   - Did you consider baseline replacement (mean, median, or SHAP-style reference values)?
   - If so, how do the results change when using baseline replacement instead of removal?
   - How would you compare the feature removal technique as a measure of robustness to out-of-distribution inputs compared to feature importance?
- The penalized accuracy hyper-parameters were glossed over. The choice of $\alpha, \beta$ appears to be arbitrary. What are the theoretical bounds for these parameters? How sensitive are results to different parameter choices? are there justifications for length violations and unknown labels to receive equal weight across all domains?


Some other questions for the authors
- While evaluating "Instruction Comprehension", the paper focuses on whether models understand task structure and output format. However, there's no evaluation of alignment between stated decision strategies and actual behavior. Have you explored prompting models to explain their intended approach before prediction, then checking whether behavior matches the stated strategy? does this alignment matter and should this be the primary criteria as motivated during the understanding definitions?
- During the post-processing, the paper used GPT-4-mini to clean outputs. However, no analytical details have been provided on potential confounding via these steps. Have you looked at how often does post-processing change predictions? Could this introduce evaluation errors and fail to catch some instruction-following failures? Do some models require more post-processing intervention than others?
- Finally, even if the LAO approach is justified, the single-feature ablations may underestimate reliance when models use higher-order feature interactions. While the paper acknowledge this but to substantiate the claims of the paper, did you explore group-based feature ablations for correlated features? How prevalent are interaction effects in these datasets?

---

> ### Author Response · Authors · 2026-02-17
>
> We sincerely thank the reviewer for the careful reading and for recognizing the motivation and importance of this work, particularly our focus on global decision faithfulness and the value of STaDS to the community. We also greatly appreciate the clear and constructive suggestions. Below, we respond point by point.
>
> ---
>
> **Q1.The paper is difficult to follow because abbreviations/notations appear before being defined (e.g., $\gamma$ and UnkLbl% are mentioned before their formal definitions).**
>
> **Response.** We appreciate this observation and agree that the ordering of definitions is important for readability and comprehension. In the revision, we will carefully reorganize the presentation so that all symbols and abbreviations are defined at first use (or removed if unnecessary). In addition, we will revise the transitions between sections to improve the overall flow and coherence of the manuscript. A notation table will be added in the appendix for easy reference.
>
> ---
>
> **Q2. The authors should further deliberate on the feature ablation methodology. The complete removal of features changes the model's functional form, which differs fundamentally from standard explainability approaches.**
> Some questions to consider:
> - Did you consider baseline replacement (mean, median, or SHAP-style reference values)?
> - If so, how do the results change when using baseline replacement instead of removal?
> - How would you compare the feature removal technique as a measure of robustness to out-of-distribution inputs compared to feature importance?
>
> **Response.** We thank the reviewer for this insightful concern regarding our feature ablation methodology. We agree that the semantics of "removal" vs. "replacement" deserve a more formal treatment, particularly when transitioning from XAI to LLM evaluation.
>
> - In conventional tabular explainability, where a predictor $f:R^d$⁣ -> ⁣$R$ expects a fixed-dimensional input, "dropping" a feature is ill-defined because the model's architecture requires a value for every dimension. Consequently, standard methods (e.g., SHAP or PFI) use baseline replacement (mean/median/reference) to simulate "absence" while keeping the input manifold fixed. However, LLMs operate on a discrete, variable-length sequence space $\mathcal{V}$. In this setting, the "functional form" is inherently flexible. Prompt-level deletion is not a structural change to the model, but a **behavioral intervention on the context**. Because LLMs are pre-trained on diverse, incomplete, and noisy text, they are natively capable of processing varying input structures. Thus, deletion is a valid method to measure concept necessity within the natural language manifold.
>
> - We view deletion as a "Maximum Information Loss" baseline, providing a cleaner intervention than permutation in an LLM context: Permutation (e.g., swapping "Gender: Male" for "Gender: Female") only measures sensitivity to a specific value. The model's attention mechanism still processes the category of "Gender," potentially activating associated latent representations or gender-coded weights. Only by removing the field entirely can we ask: "Can the model perform this task without any exposure to this concept?" Deletion ensures that no residual semantic signal or "placeholder" artifacts can be leveraged by the model's self-attention layers. In this sense, deletion serves as a metric for measuring the total contribution (structural + semantic) of a feature to the model's reasoning process.
>
> - We agree that deletion could conflate **feature importance** with **structural robustness** (i.e., sensitivity to OOD prompt formats). To disentangle these, we will incorporate a "Neutral Placeholder" baseline in our revised manuscript (e.g., replacing values with "N/A"). If the performance drop fromNeutral Replacement matches the drop from "Deletion," we can conclude the effect is purely driven by semantic reliance. If Deletion causes a significantly larger drop, it indicates a degree of structural fragility.
>
> - To address the reviewer’s request for a baseline comparison, we will also add a comparative analysis on a subset of our data using Neutral Placeholder Replacement (the LLM equivalent of mean/median imputation) and SHAP-style reference values (marginalizing over a set of "null" prompts). We believe this will demonstrate that our findings are robust to the specific choice of ablation method.

---

> > ### Author Response · Authors · 2026-02-17
> >
> > **Q3. The hyperparameters appear arbitrary.**
> >
> > **Response.** We thank the reviewer for raising this important point.
> > - The hyperparameters $\alpha$ and $\beta$ are not intended as universal constants, but as task-dependent weights that calibrate how strongly we penalize two explicit contract violations: incorrect output length and invalid labels. Both correspond to instruction-comprehension failures rather than superficial formatting artifacts. We require PenAcc to remain within [0,1]. Since both violation terms lie in [0,1], the sufficient constraint is $\alpha,\beta \ge 0$ and $\alpha+\beta \le 1$, which ensures that penalties cannot outweigh correctness. We adopt $\alpha=\beta=0.5$ as a neutral, domain-agnostic default under this budget so that (i) PenAcc remains on the same scale as accuracy and (ii) neither violation type dominates the metric. We will revise the Comprehension Fidelity Metrics section to make these bounds and design choices explicit.
> >
> > - Importantly, these penalties are not fixed by design. Different application contexts may justify different trade-offs, for example, safety-critical domains may prioritize valid labels ($\beta>\alpha$), whereas structured reporting tasks may emphasize completeness ($\alpha>\beta$). Our current choice reflects a balanced baseline rather than an optimized setting.
> > In the revision, we will add a sensitivity analysis over a carefully selected grid of $(\alpha,\beta)$ values to demonstrate this perspective across domains.
> >
> > ---
> >
> > **Q4. Alignment between stated strategy and actual behavior.**
> >
> > **Response.**  We thank the reviewer for raising this point. We agree that “strategy–behavior alignment” closely matches a common definition of faithfulness in prior work [1–3]. However, we intentionally do not include it under Instruction Comprehension, since comprehension is only one component of our broader understanding evaluation. Eliciting and validating strategy statements would require extra prompting and verification, increasing cost and introducing additional prompting variance. Instead, STaDS prioritizes scalable, model-agnostic, end-to-end behavioral indicators, comprehension fidelity, predictive competence, and global decision faithfulness, measured consistently under a unified protocol. We will clarify this design choice in the revision.
> >
> > References:
> >
> > [1]Chain-of-Thought Reasoning In The Wild Is Not Always Faithful
> >
> > [2]Language Models Don’t Always Say What They Think: Unfaithful Explanations in Chain-of-Thought Prompting
> >
> > [3]Measuring Faithfulness in Chain-of-Thought Reasoning
> >
> > ---
> >
> > **Q5. Post-processing with GPT-4-mini**
> >
> > **Response.** We thank the reviewer for this important question.
> > - GPT-4-mini is used strictly for deterministic format normalization (e.g., removing extraneous text, extracting predicted labels, and aligning outputs to the required key–value structure). It does not modify semantic content, relabel predictions, or impute missing outputs. Nevertheless, we agree that additional auditing is valuable to rule out potential confounding effects.
> >
> > - In the revision, we will use a regex-based parser to extract predicted labels directly from the raw outputs, then compare them with the GPT-4-mini–processed outputs. The agreement will be reported, and this will allow us to measure how often post-processing changes predictions, if at all. We expect agreement to be near-perfect because the post-processing step is purely structural; however, we agree that making this auditing explicit will strengthen the methodological rigor.
> >
> > ---
> >
> > **Q6. Single-feature ablations may underestimate reliance.**
> >
> > **Response.** We appreciate this observation.
> > - Our framework naturally extends to group-level ablations, since the ablation operator can be applied to any subset of features by removing correlated or semantically related feature groups together and measuring the corresponding performance drop.
> > - Given the computational cost of exhaustive interaction analysis across multiple datasets, we will focus on structured group ablations in the revision. Specifically, we will use identified highly correlated feature clusters and perform joint ablations on the most positively and negatively correlated groups. We will compare the marginal effects of grouped removal against the sum of individual removals to quantify non-additive interaction strength.
> >
> > ---
> >
> > **Q7. Regression extension.**
> >
> > **Response.** We agree that the current presentation emphasizes classification for clarity, as a pilot study, but the framework extends naturally to regression. For regression tasks, accuracy can be replaced with a standard error-based metric (e.g., MSE), normalized to [0,1] for comparability across domains. The conceptual interpretation remains the same, where performance drop under controlled perturbations reflects reliance and faithfulness, independent of label discreteness.
> > We will expand the manuscript to explicitly formalize this regression extension and provide a concrete example.

---

### Decision · Action_Editor_WJ7U · 2026-04-22

**Recommendation:** Reject

**Audience:**

Yes

**Audience Explanation:**

There was consensus that the problem addressed in this paper is of interest to the community. Reviewer 9siP considers the paper "well motivated and of high importance", Reviewer TdgX lists as strengths the "Clear formalization of understanding into measurable behavioral dimensions".

**Claims And Evidence:**

No

**Claims Explanation:**

The concerns raised in the review were not fully addressed in the rebuttal. Reviewer 9siP stated that the paper "suffers from poor presentations" and that "frequently notations and abbreviations have been introduced before they have been defined", and raises a substantive methodological worry that "by completely removing features rather than replacing them with baseline values, the authors fundamentally change the model's functional form" and that "the logical fallacy here could undermine the validity of the Decision Faithfulness metrics". Reviewer 2Zt2 writes that "it is not clear whether the benchmark is investigating a new capability of LLMs when compared with other benchmarks, e.g., multi-hop QA and/or long-context benchmarks", that "the definition of "complexity" is not articulated clearly", and that some task settings are unclear, noting that "the experiments does not take into account the positional biases".

Reviewer 2Zt2 recommended rejection. Although Reviewer 9siP summarized their recommendation as Leaning Accept, the comment states "On the whole this is an interesting work but the limitations needs to be acknowledged and clarified well". As such, I believe that this paper needs a significant revision before acceptance.

**Resubmission Of Major Revision:**

The authors may consider submitting a major revision at a later time.